# Gq activity- and β-arrestin-1 scaffolding-mediated ADGRG2/CFTR coupling are required for male fertility

Dao-Lai Zhang[1,2,3†], Yu-Jing Sun[1,2†], Ming-Liang Ma[1,2†], Yi-jing Wang[1,2†], Hui Lin[1,2], Rui-Rui Li[1,2], Zong-Lai Liang[1,2], Yuan Gao[1,2], Zhao Yang[1,2], Dong-Fang He[1,2], Amy Lin[4], Hui Mo[1,2], Yu-Jing Lu[1,2], Meng-Jing Li[1,2], Wei Kong[5], Ka Young Chung[6], Fan Yi[7], Jian-Yuan Li[8], Ying-Ying Qin[9], Jingxin Li[2], Alex R B Thomsen[4], Alem W Kahsai[4], Zi-Jiang Chen[9], Zhi-Gang Xu[10], Mingyao Liu[11,12], Dali Li[11]*, Xiao Yu[2]*, Jin-Peng Sun[1,4]*

[1]Key Laboratory Experimental Teratology of the Ministry of Education, Department of Biochemistry and Molecular Biology, Shandong University School of Medicine, Jinan, China; [2]Department of Physiology, Shandong University School of Medicine, Jinan, China; [3]School of Pharmacy, Binzhou Medical University, Yantai, China; [4]Department of Biochemistry, School of Medicine, Duke University, Durham, United States; [5]Key Laboratory of Molecular Cardiovascular Science, Department of Physiology and Pathophysiology , School of Basic Medical Sciences, Peking University, Beijing, China; [6]School of Pharmacy, Sungkyunkwan University, Suwon, Korea; [7]Department of Pharmacology, Shandong University School of Medicine, Jinan, China; [8]Key Laboratory of Male Reproductive Health, National Research Institute for Family Planning, National Health and Family Planning Commission, Beijing, China; [9]National Research Center for Assisted Reproductive Technology and Reproductive Genetics, Shandong University, Jinan, China; [10]Shandong Provincial Key Laboratory of Animal Cells and Developmental Biology, Shandong University School of Life Sciences, Jinan, China; [11]Shanghai Key Laboratory of Regulatory Biology, School of Life Sciences,  Institute of Biomedical Sciences, East China Normal University, Shanghai, China; [12]Department of Molecular and Cellular Medicine, Institute of Biosciences and Technology, Texas A&M University Health Science Center, Houston, United States

*For correspondence:
dlli@bio.ecnu.edu.cn (DL);
yuxiao@sdu.edu.cn (XY);
sunjinpeng@sdu.edu.cn (J-PS)

†These authors contributed equally to this work

Competing interests: The authors declare that no competing interests exist.

**Abstract** Luminal fluid reabsorption plays a fundamental role in male fertility. We demonstrated that the ubiquitous GPCR signaling proteins Gq and β-arrestin-1 are essential for fluid reabsorption because they mediate coupling between an orphan receptor ADGRG2 (GPR64) and the ion channel CFTR. A reduction in protein level or deficiency of ADGRG2, Gq or β-arrestin-1 in a mouse model led to an imbalance in pH homeostasis in the efferent ductules due to decreased constitutive CFTR currents. Efferent ductule dysfunction was rescued by the specific activation of another GPCR, AGTR2. Further mechanistic analysis revealed that β-arrestin-1 acts as a scaffold for ADGRG2/CFTR complex formation in apical membranes, whereas specific residues of ADGRG2 confer coupling specificity for different G protein subtypes, this specificity is critical for male fertility. Therefore, manipulation of the signaling components of the ADGRG2-Gq/β-arrestin-1/CFTR complex by small molecules may be an effective therapeutic strategy for male infertility.
DOI: https://doi.org/10.7554/eLife.33432.001

## Introduction

Male infertility is transforming from a personal issue to a public health problem because approximately 15% of reproductive-age couples are infertile, and male infertility accounts for approximately 50% of this sterility (*Hamada et al., 2012*; *Jodar et al., 2015*). The unique structure of the male reproductive system increases the difficulty of determining the working mechanisms. Among male reproductive system, the efferent ductules of the male testis play important roles during sperm transportation and maturation by reabsorbing the fluid of the rete testis and maintaining the homeostasis of water and ion metabolism (*Hess et al., 1997*). Whereas a dysfunction of the efferent ductule reabsorption capacity caused by a developmental defect that produces improper signaling results in epididymal obstructions and abnormal spermiostasis, which ultimately lead to infertility in both humans and other mammals (*Hendry et al., 1990*; *Nistal et al., 1999*), manipulating the reabsorption function in the efferent ductules could be developed into a useful contraceptive method for males (*Gottwald et al., 2006*).

Receptors play key roles in the regulation of fluid reabsorption in tissues such as the proximal tubules and alveoli (*Haithcock et al., 1999*; *Thomson et al., 2006*). In contrast, only a few receptor functions in the efferent ductules have been characterized. Nuclear estrogen receptor α (ERα) must be activated for male reproductive tract development and reabsorption function maintenance to occur (*Hess et al., 1997*). However, the mechanism by which fluid reabsorption is regulated by cell surface receptors in the efferent ductules is only beginning to be appreciated (*Shum et al., 2008*). Knockout of an orphan G-protein-coupled receptor (GPCR), ADGRG2 (adhesion G-protein-coupled receptor G2), results in male infertility due to dysregulated fluid reabsorption in the efferent ductules, suggesting an active role for this cell surface receptor in regulating these processes (*Davies et al., 2004*). However, how ADGRG2 regulates water-ion homeostasis and fluid reabsorption remains elusive.

ADGRG2 belongs to the seven transmembrane receptor superfamily (*Hamann et al., 2015*), which regulates approximately 80% of signal transduction across the plasma membrane and accounts for 30% of current clinical prescription drug targets. Five different types of G proteins and arrestins act as signaling hubs downstream of these GPCRs, mediating most of their functions (*Alvarez-Curto et al., 2016*; *Cahill et al., 2017*; *Dong et al., 2017*; *Li et al., 2018*; *Liu et al., 2017*; *Nuber et al., 2016*; *Thomsen et al., 2016*; *Yang et al., 2015*). In the efferent ductules, it remains unclear how G proteins and their parallel signaling molecules, the arrestins, regulate reabsorption as well as fertility.

Here, we developed a new labeling method utilizing specific red fluorescent protein (RFP) expression driven by the ADGRG2 promoter, which enabled a detailed mechanistic study of efferent ductule functions. By exploiting $Adgrg2^{-/Y}$, $Gnaq^{+/-}$, $Arrb1^{-/-}$ and $Arrb2^{-/-}$ knockout mouse models, together with the combination of pharmacological interventions and electrophysiological approaches, we have identified the importance of the ubiquitous Gq protein and β-arrestin-1, which confer the ADGRG2 constitutive activity to a basic cystic fibrosis transmembrane conductance regulator (CFTR) current, in fluid reabsorption in the efferent ductules. Both specific Gq activity- and β-arrestin-1 scaffolding-mediated ADGRG2/CFTR coupling are required for male fertility and Cl⁻/ acid-base homeostasis in the efferent ductules. Our results not only reveal how fluid reabsorption in the male efferent ductules is precisely controlled by a specific subcellular signaling compartment encompassing ADGRG2, CFTR, β-arrestin-1 and Gq in non-ciliated cells but also provide a foundation for the development of new therapeutic approaches to control male fertility.

## Results

### Gq activity is required for fluid reabsorption and male fertility

Previous studies have found that knockout of the orphan receptor ADGRG2 causes infertility and fluid reabsorption dysfunction in the efferent ductules, indicating important roles for GPCR signaling in male reproductive functions. Downstream of GPCRs, there are 16 Gα proteins that mediate diverse GPCR functions (*DeVree et al., 2016*). However, the expression of these G protein subtypes and their functions in the efferent ductules have not been investigated. Here, we show that Gs is more enriched, while G11 and Gi3 have expression levels in the efferent ductules similar to those in brain tissue, whereas all other 11 tested G protein subtypes have detectable expression levels in the

efferent ductules (*Figure 1A*). ADGRG2 localizes in cells devoid of acetylated-tubulin staining, suggesting that it is specifically expressed in non-ciliated cells (*Figure 1B* and *Figure 1—figure supplement 1A–C*). We next used the promoter region of ADGRG2 to direct the expression of the fluorescent protein RFP, which enabled the specific labeling of ADGRG2-expressing non-ciliated cells in the efferent ductules (*Figure 1C* and *Figure 1—figure supplement 2A–C*). After fluorescence-activated cell sorting (FACS), quantitative RT-PCR (qRT-PCR) results indicated that ADGRG2-expressing non-ciliated cells have expression levels of Golf, Gi2, Gq, G11, and G13 that are higher than those in brain tissue and expression levels of Gs, G12 and Gz that are similar to those in brain tissue (*Figure 1D*).

We next investigated the contribution of different G protein subtype signaling pathways to fluid reabsorption in the efferent ductules using specific pharmacological interventions and knockout models. An ADGRG2 knockout mouse was produced by introducing an 11-nucleotide sequence into the first exon of the ADGRG2 gene (*Figure 2—figure supplement 1*), thereby creating a positive control for fluid reabsorption dysfunction in the efferent ductules (*Davies et al., 2004*). The wild-type (WT) mice did not show size alterations due to the normal reabsorption of luminal fluid, but the ligated efferent ductules derived from the ADGRG2 knockout mice displayed a 40% increase in luminal area after 72 hr of in vitro culture (*Figure 2A*). Application of the Gi inhibitor pertussis toxin (PTX) or the MEK-ERK signaling inhibitor U0126 did not have a significant effect on the efferent ductules (*Figure 2B and C*). In contrast, a 50% reduction in Gq protein levels in $Gnaq^{+/-}$ mice or the application of the protein kinase C(PKC) inhibitor Ro 31–8220 significantly impaired fluid reabsorption in the efferent ductules, which mimicked the phenotype of the ductules derived from $Adgrg2^{-/Y}$ mice (*Figure 2A and D–E* and *Figure 2—figure supplement 1F–G*). The contribution of Gs-PKA (protein kinase A) signaling to fluid reabsorption of the efferent ductules is confounded. While the application of the Gs inhibitor NF449 or the PKA inhibitors PKI14-22 or H89 to the efferent ductules derived from WT mice slightly increased the volume of the efferent ductules (*Figure 2F–H*), cAMP regulators, such as the adenyl cyclase activator forskolin (FSK) and the phosphodiesterase (PDE) inhibitor 3-isobutyl-1-methylxanthine(IBMX), increased the volume of the efferent ductules in an acute manner in both $Adgrg2^{-/Y}$ mice and WT littermates (*Figure 2—figure supplement 2A–B*). These results suggested that Gs-PKA signaling is finely tuned in the efferent ductules to maintain its fluid reabsorption function because both increasing and decreasing its activity caused detrimental effects. In conclusion, Gi and MEK-ERK signaling exerted no significant effects, whereas Gq-PKC signaling was required for efficient fluid reabsorption in the efferent ductules.

The efferent ductules of the $Gnaq^{+/-}$ animals consistently showed the accumulation of obstructed spermatozoa compared with those of WT mice, whereas the lumen of the initial segment and caput region in $Gnaq^{+/-}$ mice contained significantly reduced sperm levels (*Figure 3A–D*). Sperm numbers prepared from the caudal epididymis and the birth rate of the $Gnaq^{+/-}$ mice were also significantly decreased compared with their WT littermates (*Figure 3E–G*). Taken together, these data demonstrated that among different G protein subtypes, Gq activity is required for fluid reabsorption and male fertility.

## ADGRG2 and CFTR coupling in the efferent ductules and its function in fluid reabsorption

Membrane proteins, including bicarbonate and chloride transporters, sodium/potassium pumps and specific ion channels, are potential osmotic drivers for fluid secretion and reabsorption in the efferent ductules (*Estévez et al., 2001*; *Harvey, 1992*; *Liu et al., 2015*; *Park et al., 2001*; *Russell, 2000*; *Xiao et al., 2012*; *Xiao et al., 2011*; *Zhou et al., 2001*). Therefore, we examined the expression levels of these membrane proteins in the efferent ductules and ADGRG2 promoter-labeled ductule cells (*Figure 4A* and *Figure 4—figure supplement 1A*). Specifically, $Na^+$-$K^+$-$Cl^-$ cotransporter (NKCC), down-regulated in adenoma (DRA), CFTR, solute carrier family 26 member 9(SLC26a9), $Na^+$/$H^+$ exchanger 3(NHE3) and the L-type voltage dependent calcium channel Cav1.3 levels were readily measured in ADGRG2 promoter-labeled non-ciliated ductule cells; $Na^+$/$H^+$ exchanger 1 (NHE1), carbonic anhydrase II(CAII), Short transient receptor potential channel 3(TRPC3), chloride channel accessory 1(CLCA1) and Cav1.2 had lower but detectable expression levels, whereas anoctamin-1 (ANO1), V-ATPase and Cav2.2 demonstrated very little expression (*Figure 4A* and *Figure 4—figure supplement 1A*). Notably, we used the ADGRG2 promoter to label the non-ciliated cells, as the ADGRG2 receptor is specifically expressed on the apical membrane of these cells in

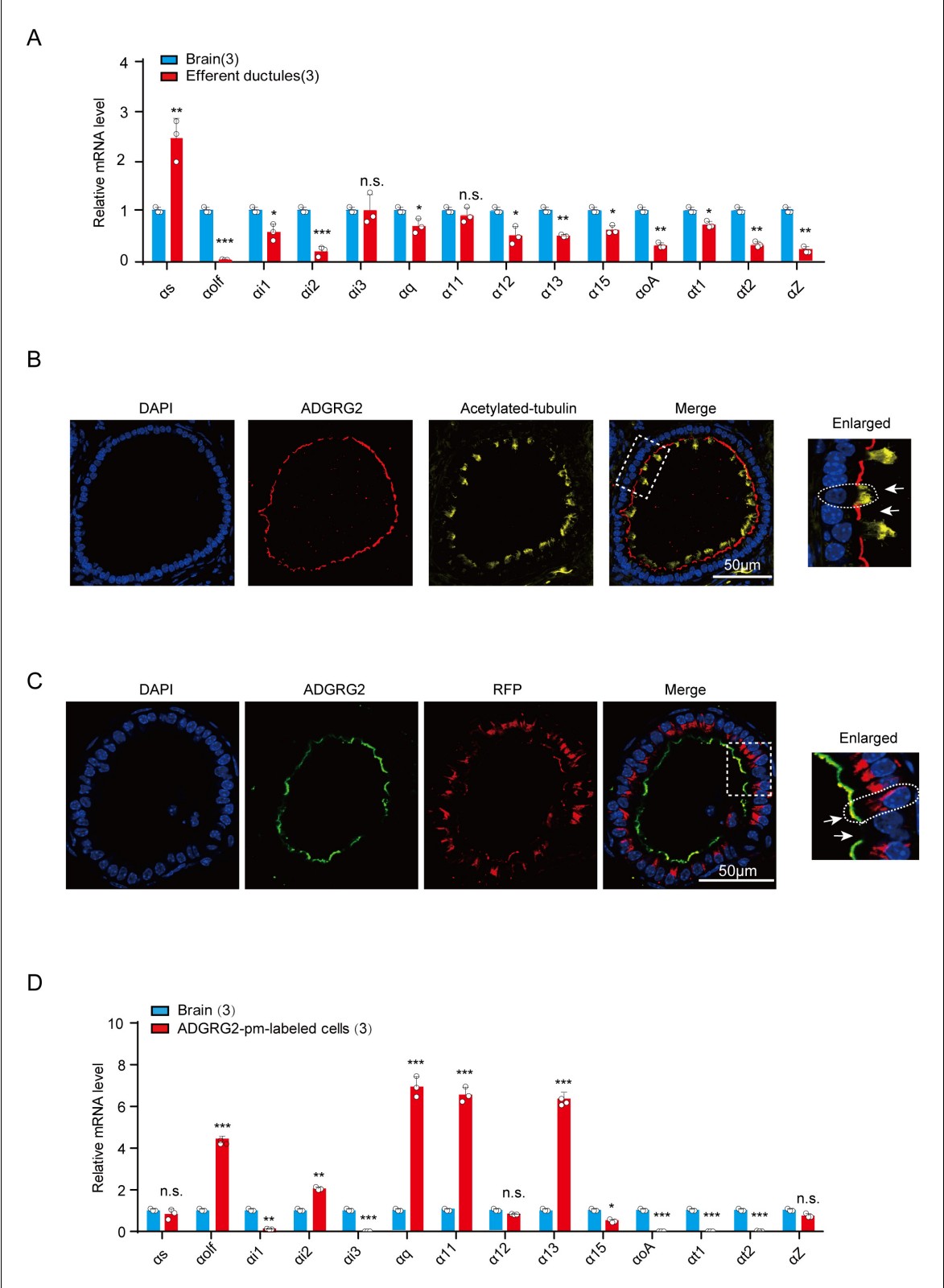

**Figure 1.** The expression of G protein subtypes in the efferent ductules and ADGRG2 promoter-labeled non-ciliated cells. (**A**) qRT-PCR analysis of mRNA transcription profiles of G proteins in brain tissues and the efferent ductules of WT (n = 3) male mice. Expression levels were normalized to GAPDH levels. *p<0.05, **p<0.01, ***p<0.001, efferent ductules compared with brain tissue. (**B**) Co-localization analysis of ADGRG2 (red fluorescence) and acetylated-tubulin (green fluorescence) in the efferent ductules of WT mice. Scale bars, 50 μm. (**C**) Co-localization of ADGRG2 (green fluorescence)

*Figure 1 continued on next page*

Figure 1 continued

and RFP (red fluorescence) in the same cells of male murine efferent ductules infected with the ADGRG2 promoter RFP adenovirus in WT mice. Scale bars, 50 μm. (D) qRT-PCR analysis of mRNA transcription profiles of G protein subtypes in brain tissues and isolated ADGRG2 promoter-labeled non-ciliated cells derived from the efferent ductules of WT (n = 3) male mice. Expression levels were normalized to GAPDH levels. *p<0.05, **p<0.01, ***p<0.001, ADGRG2 promoter-labeled efferent ductule cells compared with brain tissues. n.s., no significant difference. At least three independent biological replicates were performed for *Figure 1A and D*.

DOI: https://doi.org/10.7554/eLife.33432.002

The following figure supplements are available for figure 1:

**Figure supplement 1.** ADGRG2 is specifically expressed in non-ciliated cells.

DOI: https://doi.org/10.7554/eLife.33432.003

**Figure supplement 2.** The construction of the mouse ADGRG2-promoter-RFP used in the labeling of ADGRG2-expressed cells.

DOI: https://doi.org/10.7554/eLife.33432.004

efferent ductules (*Figure 1B–C* and *Figure 1—figure supplements 1–2*). A higher expression level of a particular membrane protein, such as CFTR, in the ADGRG2 promoter-labeled cells indicated that these membrane proteins are enriched in non-ciliated cells in efferent ductules but does not indicate that the expression of these proteins is dependent on ADGRG2. For example, the CFTR expression level in ADGRG2 promoter-labeled efferent ductule cells derived from *Adgrg2$^{-/Y}$* mice did not differ significantly from that in the cells derived from their WT littermates (*Figure 4—figure supplement 1D*).

We next used a panel of pharmacological blockers to examine whether the inappropriate regulation of these membrane protein functions was involved in the ADGRG2- or Gq-mediated regulation of fluid reabsorption in the efferent ductules. Importantly, application of the NKCC blocker bumetanide, the ANO1 inhibitor Ani9, the calcium-dependent chloride channel (CaCC) inhibitor niflumic acid (NFA), TRP channel inhibitors including ruthenium red, SKF96365 and LaCl$_3$, the L-type calcium channel blocker nicardipine or chelating extracellular calcium with EGTA showed no significant effects on fluid reabsorption in the efferent ductules in ligation experiments (*Figure 4B–H* and *Figure 4—figure supplement 1B*). Application of 4,4'-diisothiocyano-2,2'-stilbenedisulfonic acid (DIDS) to block the chloride-bicarbonate exchanger exerted a small effect only after 60 hr (*Figure 4I*), and the application of amiloride to inhibit sodium/hydrogen antiporter NHE1 activity exerted an acute effect on fluid reabsorption (*Figure 4—figure supplement 1C*), an outcome different from that observed in *Adgrg2$^{-/Y}$* or *Gnaq$^{+/-}$* mice (*Figure 2D*). In contrast, blocking CFTR activity either with GlyH-101 or CFTRinh-172 had significant effects on fluid reabsorption in the efferent ductules and pheno-copied the *Adgrg2$^{-/Y}$* mice (*Figure 4J–K*). Collectively, the phenotype caused by inactivating ADGRG2 and administering a CFTR channel blocker in WT mice suggested that CFTR and ADGRG2 may be functionally connected to the regulation of fluid reabsorption.

CFTR is the key regulator of pH homeostasis and chloride in the reproductive and renal systems and has important functions in fluid reabsorption (*Chen et al., 2012*). Therefore, we measured the pH value of the efferent ductules. The pH homeostasis was impaired in *Adgrg2$^{-/Y}$* mice, with a pH value of 7.6 for the inner solution in the efferent ductules, compared to a pH of 7.2 in WT littermates (*Figure 5A* and *Figure 5—figure supplement 2*). This dysfunction was not caused by decreased CFTR expression because the mRNA levels of CFTR in the *Adgrg2$^{-/Y}$* mice were not reduced compared with those of their WT littermates (*Figure 5B* and *Figure 5—figure supplement 1*). Moreover, application of the CFTR inhibitor CFTRinh-172 increased the pH value of the efferent ductules in WT mice by approximately 0.3 but did not have a significant effect in *Adgrg2$^{-/Y}$* mice, suggesting that CFTR dysfunction in *Adgrg2$^{-/Y}$* mice influences pH homeostasis (*Figure 5A–B*). Importantly, the pH imbalance in *Adgrg2$^{-/Y}$* mice was rescued by bicarb-free media or application of the carbonic anhydrase inhibitor acetazolamide (*Figure 5—figure supplement 2B–C*).

In particular, unambiguous co-localization of ADGRG2 and CFTR on the apical membrane was detected (*Figure 5C–G* and *Figure 5—figure supplement 3*) and ADGRG2 was associated with CFTR in co-immunoprecipitation assays (*Figure 5H* and *Figure 5—figure supplement 4*). Taken together, these results suggest a complex formation and functional coupling of ADGRG2 and CFTR in the non-ciliated cells of the efferent ductules.

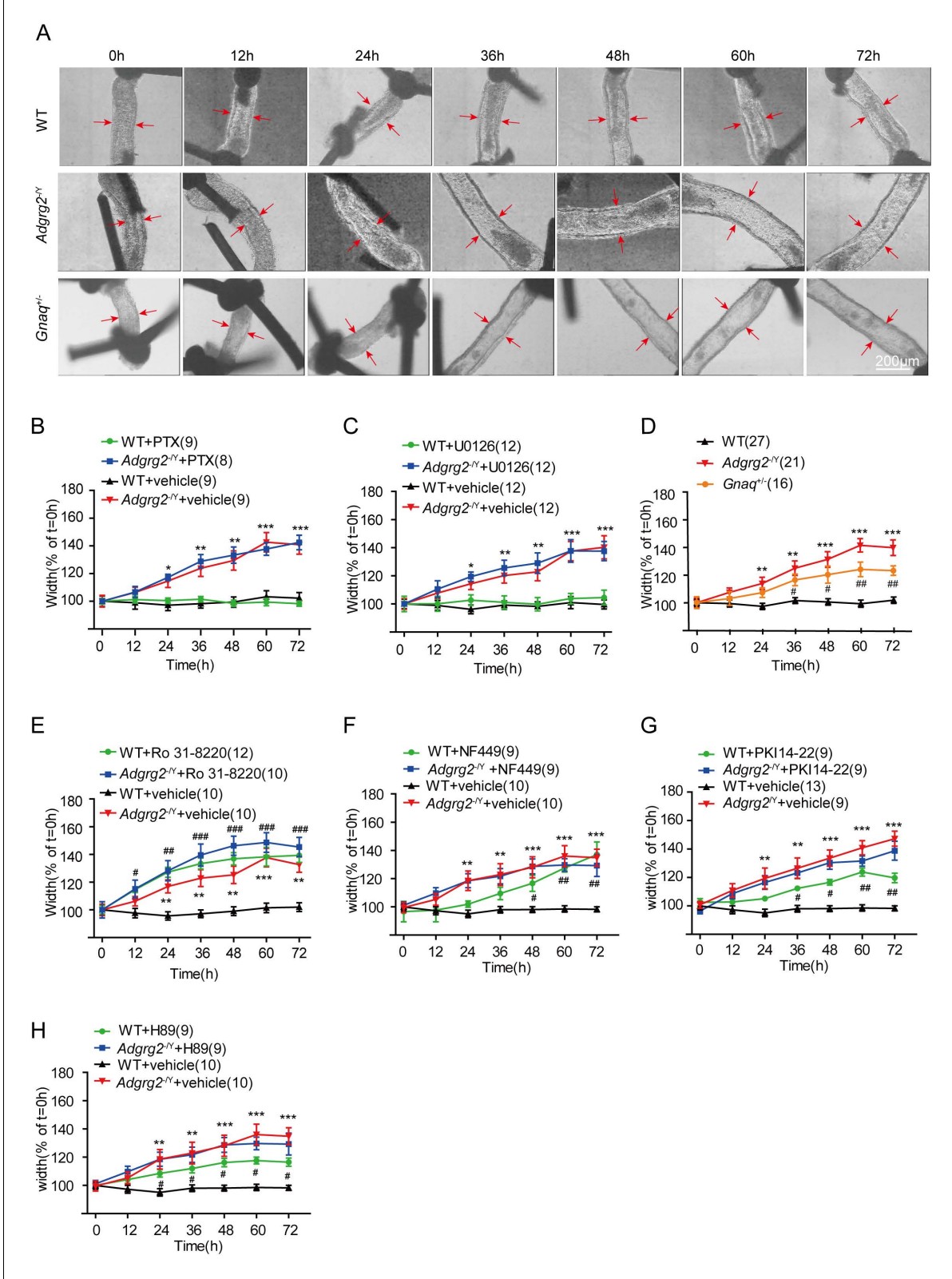

**Figure 2.** Gq activity is required for fluid reabsorption. (**A**) Images of cultured ligated efferent ductules derived from WT male mice, *Adgrg2*[-/Y] mice and *Gnaq*[+/-] male mice. Ductule segments were selected by examination of the ciliary beat, which is a marker of cell integrity. Ductule pieces from *Adgrg2*[-/Y], *Gnaq*[+/-] or WT mice were ligated, microdissected and cultured for up to 72 hr. Scale bars, 200 μm. (**B–C, E–H**) Effects of pharmacological intervention on the diameters of ligated efferent ductules derived from WT or *Adgrg2*[-/Y] mice. (**B**) PTX (100 ng/ml), a Gi inhibitory protein. WT (n = 9) or *Adgrg2*[-/Y]

*Figure 2 continued on next page*

*Figure 2 continued*

(n = 8); (**C**) U0126 (10 μM), a MEK inhibitor (ERK pathway blockade), WT (n = 12) or *Adgrg2^(-/Y)* (n = 12). (**E**) Ro 31–8220 (500 nM), a protein kinase C (PKC) inhibitor, WT (n = 12) or *Adgrg2^(-/Y)* (n = 10); (**F**) NF449 (1 μM), a Gs inhibitor, WT (n = 9) or *Adgrg2^(-/Y)* (n = 9); (**G**) PKI14-22 (300 nM), a PKA inhibitor, WT (n = 9) or *Adgrg2^(-/Y)* (n = 9); (**H**) H89 (500 nM), a non-selective PKA inhibitor, WT (n = 9) or *Adgrg2^(-/Y)* (n = 9). (**D**) Diameters of the luminal ductules derived from WT (n = 27) mice remained unchanged over 72 hr, whereas the lumens of the ductules derived from *Adgrg2^(-/Y)* (n = 21) mice and *Gnaq^(+/-)* (n = 16) mice were significantly increased, indicating fluid reabsorption dysfunction. (2B-2H) *p<0.05, **p<0.01, ***p<0.001, *Adgrg2^(-/Y)* mice or *Gnaq^(+/-)* mice were compared with WT mice. #p<0.05, ##p<0.01, ###p<0.001, treatment with selective inhibitors or stimulators was compared with control vehicles. n.s., no significant difference. At least three independent biological replicates were performed for *Figure 2B–H*.

DOI: https://doi.org/10.7554/eLife.33432.005

The following figure supplements are available for figure 2:

**Figure supplement 1.** The ADGRG2 protein knockout strategy, PCR strategy and western blot results of *Adgrg2^(-/Y)* and *Gnaq^(+/-)* mice.

DOI: https://doi.org/10.7554/eLife.33432.006

**Figure supplement 2.** Effects of Forskolin and IBMX on the diameters of ligated efferent ductules derived from WT or *Adgrg2^(-/Y)* mice.

DOI: https://doi.org/10.7554/eLife.33432.007

## The outwardly rectifying whole-cell Cl⁻ current ($I_{ADGRG2-ED}$) of ADGRG2 promoter-labeled efferent ductule cells

We then performed whole-cell Cl⁻ recording of primary ADGRG2 promoter-labeled efferent ductule cells derived from WT and *Adgrg2^(-/Y)* mice with normal Cl⁻ concentrations or by substituting Cl⁻ with gluconate (Gluc⁻) in the bath solution (*Figure 6A–E* and *Table 1*). Patch-clamp recording on ADGRG2 promoter-labeled non-ciliated cells derived from WT mice revealed a reversible whole-cell Cl⁻ current ($I_{ADGRG2-ED}$), which was significantly diminished in response to substitution of the bath Cl⁻ solution with Gluc⁻ (148.5 mM Cl⁻ was replaced by 48.5 mM Cl⁻ and 100 mM Gluc⁻) (*Figure 6A–B*). This whole-cell Cl⁻ current ($I_{ADGRG2-ED}$) was recovered once Gluc⁻ was substituted with Cl⁻ solution (*Figure 6A–B*). Further I-V analysis identified an outwardly rectifying whole-cell Cl⁻ current ($I_{ADGRG2-ED}$) of wild type mice, which was significantly reduced in response to Gluc⁻ substitution (*Figure 6C–E* and *Table 1*). The change in the reversal potential ($E_{rev}$) with Gluc⁻ replacement followed the Nernst equation (*Figure 6C* and *Table 1*). In contrast, the $I_{ADGRG2-ED}$ of *Adgrg2^(-/Y)* mice was substantially lower than the $I_{ADGRG2-ED}$ of their WT littermates, which showed no significant changes in response to substitution of the bath Cl⁻ solution with Gluc⁻ (*Figure 6A–6E* and *Table 1*). These results suggested that ADGRG2 deficiency in the efferent ductules significantly reduced the whole-cell Cl⁻ current of ADGRG2 promoter-labeled non-ciliated cells.

## CFTR mediates the whole-cell Cl⁻ current of ADGRG2 promoter-labeled efferent ductule cells

We next examined the effects of different Cl⁻ channel and transporter inhibitors on the $I_{ADGRG2-ED}$ of efferent ductule cells derived from *Adgrg2^(-/Y)* mice and their WT littermates. Although application of the ANO1 inhibitor Ani9 or the chloride-bicarbonate exchanger inhibitor DIDS exerted no significant effects on the $I_{ADGRG2-ED}$ of WT mice, the specific CFTR inhibitor CFTRinh-172 significantly reduced the $I_{ADGRG2-ED}$ current (*Figure 7A–B* and *Figure 7—figure supplement 1*). Moreover, the difference in the $I_{ADGRG2-ED}$ between *Adgrg2^(-/Y)* mice and their WT littermates was eliminated by the application of CFTRinh-172(*Figure 7A–B*). After the application of CFTRinh-172, the $I_{ADGRG2-ED}$ showed no significant response to Gluc⁻ substitution in the bath solution (*Figure 7—figure supplement 2*). Consistently, when we knocked down CFTR expression in efferent ductules (*Figure 7C*), the whole-cell Cl⁻ current ($I_{ADGRG2-ED}$) of WT mice was significantly reduced (*Figure 7D–E* and *Figure 7—figure supplement 3*). These results suggested that CFTR is essentially activated in ADGRG2 promoter-labeled efferent ductule cells, which mediate the observed outwardly rectifying whole-cell Cl⁻ current, and ADGRG2 is required for the basic activation of CFTR in these cells.

CFTR is activated by FSK and IBMX (*Lu et al., 2010*). In response to FSK and IBMX stimulation, the $I_{ADGRG2-ED}$ of both *Adgrg2^(-/Y)* and WT mice significantly increased to similar levels (*Figure 7F–G*), consistent with the western blot results, indicating that CFTR expression levels did not change in *Adgrg2^(-/Y)* mice. The results also indicated that basic CFTR activation in ADGRG2 promoter-labeled efferent ductule cells does not represent the full activation state (*Figure 7F–G*).

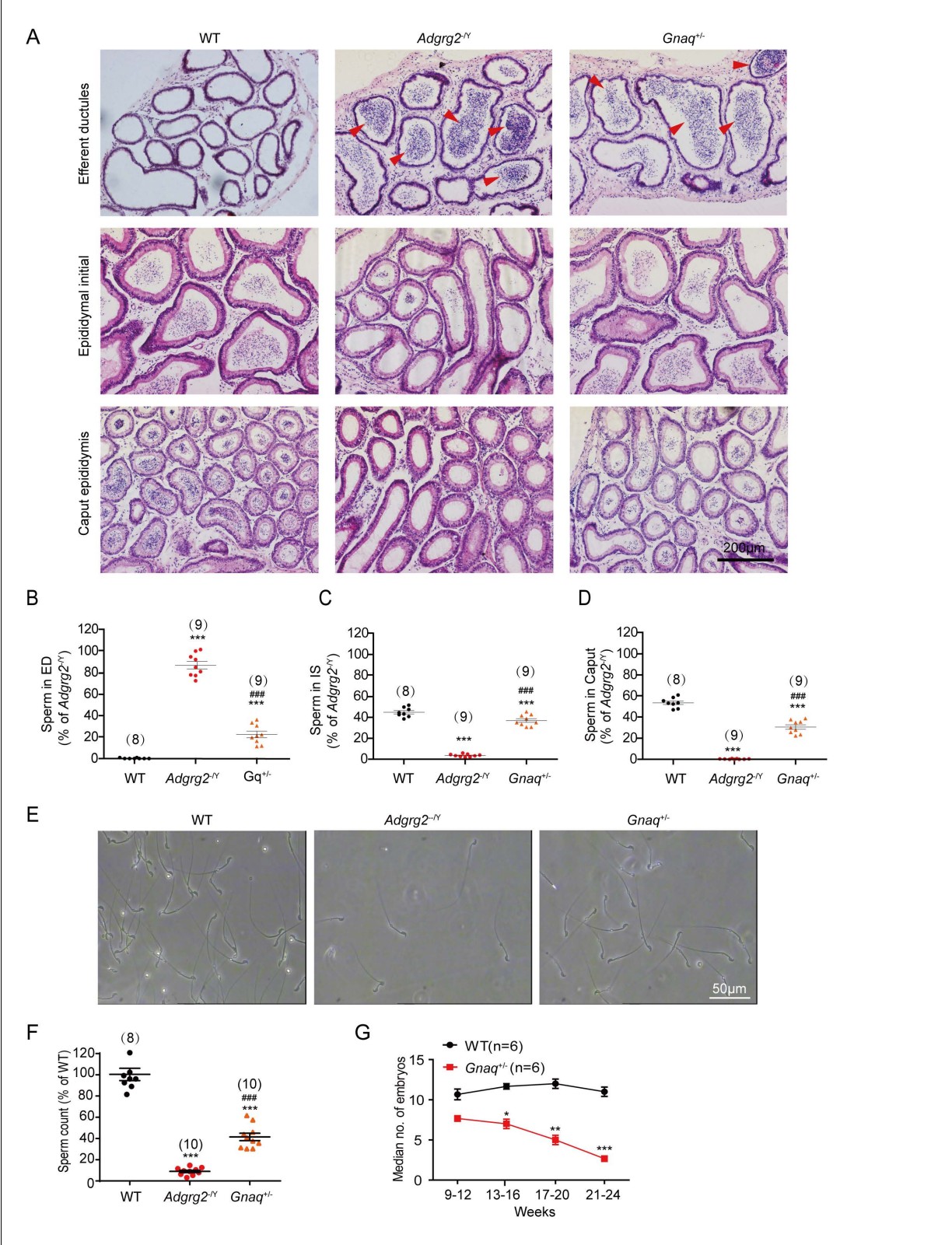

**Figure 3.** Gq expression is required for sperm transportation and male fertility. (**A**) Representative hematoxylin and eosin staining of WT, *Adgrg2-/Y* or *Gnaq+/-* mice. Scale bars, 200 μm. (**B–D**) Corresponding bar graphs demonstrating the accumulation of spermatozoa according to the hematoxylin and eosin staining of WT (n = 8), *Adgrg2-/Y* (n = 9) or *Gnaq+/-* (n = 9) mice. ED: efferent ductules; IS: epididymal initial segment; CA: caput epididymis. (**E**) Representative photographs of caudal sperm preparation from the caudal epididymis of WT, *Adgrg2-/Y* or *Gnaq+/-* mice. Scale bars, 50 μm. (**F**) Bar

*Figure 3 continued on next page*

*Figure 3 continued*

graph depicting the quantitative analysis of the number of sperm shown in (*Figure 1E*) of WT (n = 8), *Adgrg2*[-/Y] (n = 10) or *Gnaq*[+/-] (n = 10) mice. (**G**) Line graph depicting the fertility of *Gnaq*[+/-] (n = 6) and WT (n = 6) male mice at various ages, as measured by the median number of embryos. (3B-D and 3 F-G): *p<0.05, **p<0.01, ***p<0.001, *Adgrg2*[-/Y] mice or *Gnaq*[+/-] mice were compared with WT mice. #p<0.05, ##p<0.01, ###p<0.001. *Gnaq*[+/-] mice were compared with *Adgrg2*[-/Y] mice. n.s., no significant difference. At least three independent biological replicates were performed for ***Figure 3B–D and and F–G***.

DOI: https://doi.org/10.7554/eLife.33432.008

## Gq activity is required for ADGRG2/CFTR coupling in the efferent ductules

Similar to *Adgrg2*[-/Y] mice, the efferent ductules derived from *Gnaq*[+/-] mice exhibited imbalances in pH homeostasis (*Figure 8A*). We utilized *Gnaq*[+/-] mice because *Gnaq*[-/-] mice were not available due to the infertility of the *Gnaq*[+/-] mice. Consistently, we observed a significantly decreased whole-cell Cl[-] $I_{ADGRG2-ED}$ current of the ADGRG2 promoter-RFP-labeled primary non-ciliated cells in *Gnaq*[+/-] mice compared with that observed in their WT littermates (*Figure 8B–D* and *Figure 8—figure supplement 1A–B*). The application of Ro 31–8220, an inhibitor of the Gq downstream effector PKC, further inhibited the observed $I_{ADGRG2-ED}$ and showed much stronger effects than the PKA inhibitor PKI 14–22 (*Figure 8—figure supplement 1D–G*). These results indicated that the Gq-PKC pathway plays critical roles in basic CFTR activation in the efferent ductules, which controls Cl[-] and pH homeostasis for efficient fluid reabsorption.

We next investigated whether Gq activation by ADGRG2 is required for CFTR function, as both Gq and ADGRG2 are required for normal CFTR currents in the efferent ductules. In the efferent ductules, the Gq is localized in ADGRG2-expressing cells but not acetylated tubulin-labeled cells (*Figure 8E–F* and *Figure 8—figure supplement 2*). Consistently, Gq was readily detected in ADGRG2 antibody immuno-precipitated complexes, whereas Gi was not detectable, suggesting a physical interaction of ADGRG2 with Gq in the efferent ductules (*Figure 5H* and *Figure 5—figure supplement 4*). Moreover, the endogenous resting IP1 and cAMP levels of the ligated efferent ductules derived from the *Adgrg2*[-/Y] mice were significantly lower than those of their WT littermates (*Figure 8G and H*). These decreases were not caused by changes in the expression of the Gs-Adenyl-cyclase or Gq-PLC (Phospholipase C) system because Gs and Gq protein levels were similar (*Figure 8—figure supplement 3*), and the application of ATP induced similar levels of IP3 accumulation in the *Adgrg2*[-/Y] mice and their WT littermates (*Figure 8G*). Taken together, these data indicate that Gq regulates fluid reabsorption by mediating ADGRG2/CFTR coupling, and both the Gq-IP3-PKC pathway and the Gs-cAMP pathway were activated in ADGRG2 promoter-labeled efferent ductule cells.

Previous studies have shown that the activation of Angiotensin II receptor, type 2(AGTR2) increases proton secretion (*Shum et al., 2008*). We therefore stimulated the efferent ductules with different concentrations of angiotensin II and evaluated whether they rescued the fluid reabsorption dysfunction in *Adgrg2*[-/Y] mice by restoring pH homeostasis in the efferent ductules. Although applying 1 μM angiotensin II had no significant effect, administering 100 nM angiotensin II restored fluid reabsorption in the efferent ductules derived from *Adgrg2*[-/Y] mice (*Figure 4L–M*). This rescue was blocked by only the AGTR2 antagonist PD123319 (*Figure 4L*) but not by the Angiotensin II receptor, type 1(AGTR1) antagonist candesartan (*Figure 4M*). In summary, Gq and ADGRG2 regulated fluid reabsorption by maintaining pH and chloride homeostasis. The pharmacological activation of AGTR2 rescued the ADGRG2 or Gq dysfunction involved in fluid reabsorption in the efferent ductules.

## ADGRG2/CFTR complex formation mediated by β-arrestin-1 but not β-arrestin-2 is essential for fluid reabsorption in the efferent ductules

In parallel with G protein signaling, arrestins mediate important functions downstream of many GPCRs, including the connection of GPCR activation to channel functions (*Alvarez-Curto et al., 2016*; *Dong et al., 2017*; *Liu et al., 2017*; *Thomsen et al., 2016*). We therefore examined the fluid reabsorption in *Arrb1*[-/-] and *Arrb2*[-/-] knockout mice. Whereas the efferent ductules derived from *Arrb2*[-/-] knockout mice showed normal fluid reabsorption as well as pH homeostasis compared to their WT littermates, these functions of the efferent ductules derived from *Arrb1*[-/-] knockout mice

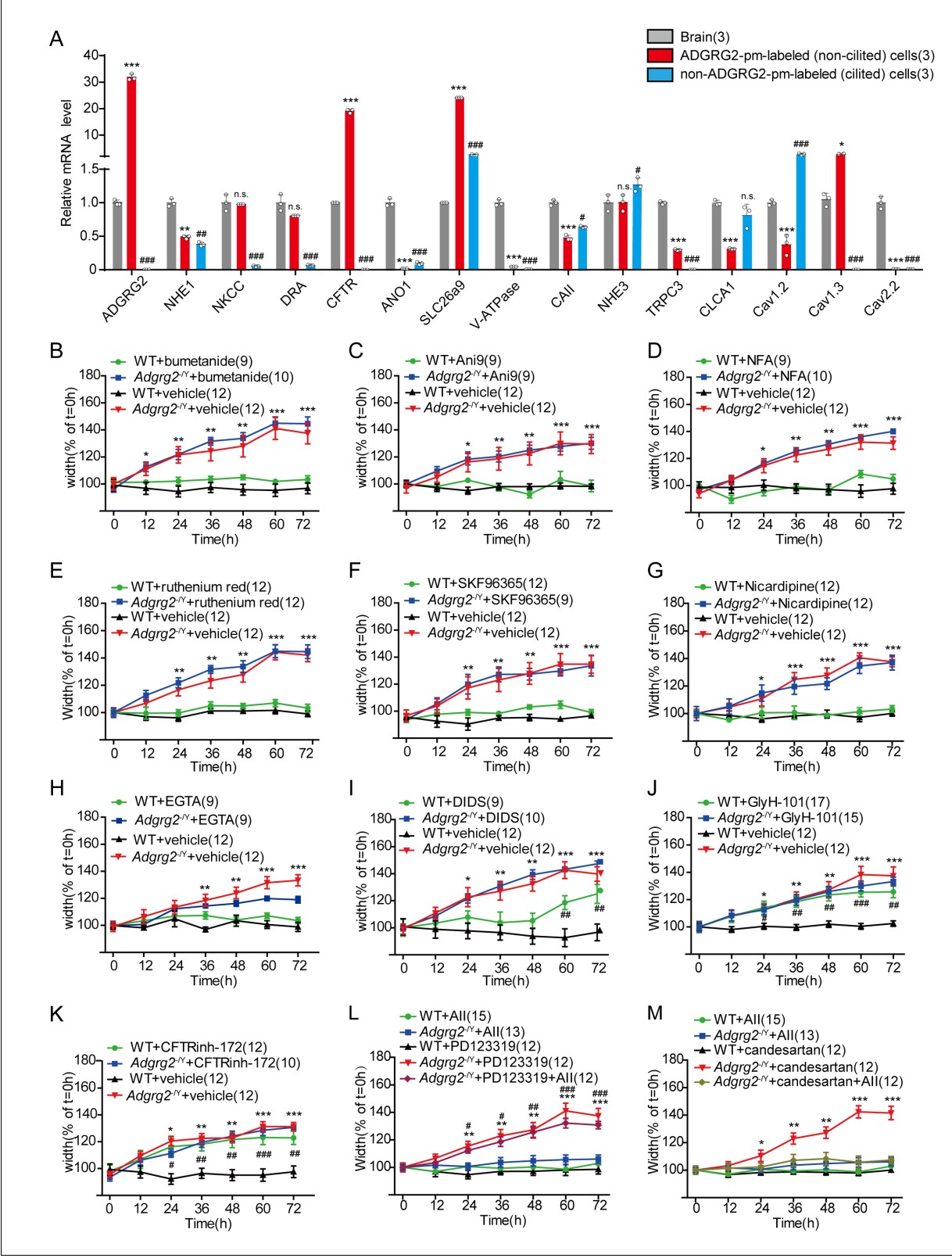

**Figure 4.** Inhibition of CFTR activity in the efferent ductules pheno-copied the activity in *Adgrg2⁻/Y* mice. (**A**) qRT-PCR analysis of the mRNA transcription profiles of potential osmotic drivers including selective ion channels and transporters in ADGRG2 promoter-labeled cells, non-ADGRG2 promoter-labeled cells and brain tissues of WT (n = 3) male mice. Expression levels were normalized to GAPDH levels. *p<0.05, **p<0.01, ***p<0.001, ADGRG2 promoter-labeled cells were compared with brain tissues. #p<0.05, ##p<0.01, ###p<0.001, non-ADGRG2 promoter-labeled cells were

*Figure 4 continued on next page*

Figure 4 continued

compared with brain tissues. (B–M) Effects of different channel blockers on the diameters of luminal ductules derived from WT or $Adgrg2^{-/Y}$ mice. (B) Bumetanide (10 µM), an NKCC blocker, WT (n = 9) or $Adgrg2^{-/Y}$ (n = 10); (C) Ani9 (150 nM), an ANO1 inhibitor, WT (n = 9) or $Adgrg2^{-/Y}$ (n = 9); (D) NFA (20 µM), a CaCC inhibitor, WT (n = 9) or $Adgrg2^{-/Y}$ (n = 10); (E) ruthenium red (10 µM), a non-specific TRP channel blocker, WT (n = 12) or $Adgrg2^{-/Y}$ (n = 12); (F) SKF96365 (10 µM), a TRPC channel inhibitor, WT (n = 12) or $Adgrg2^{-/Y}$ (n = 9); (G) nicardipine (20 µM), an L-type calcium channel blocker, WT (n = 12) or $Adgrg2^{-/Y}$ (n = 12); (H) EGTA (5 mM), an extracellular calcium chelator, WT (n = 9) or $Adgrg2^{-/Y}$ (n = 9); (I) DIDS (20 µM), a chloride-bicarbonate exchanger blocker, WT (n = 9) or $Adgrg2^{-/Y}$ (n = 10); (J) GlyH-101 (25 µM), a non-specific CFTR inhibitor, WT (n = 17) or $Adgrg2^{-/Y}$ (n = 15); (K) CFTRinh-172(10 µM), a specific CFTR inhibitor, WT (n = 12) or $Adgrg2^{-/Y}$ (n = 10). (L) Effects of angiotensin II (100 nM, an angiotensin receptor agonist) and PD123319 (1 µM, an AT2 receptor antagonist) on the diameters of luminal ductules derived from WT or $Adgrg2^{-/Y}$ mice (n ≥ 12). (M) Effects of angiotensin II (100 nM) and candesartan (1 µM, an AT1 receptor antagonist) on the diameters of luminal ductules derived from WT or $Adgrg2^{-/Y}$ mice (n ≥ 12). Application of GlyH-101 and CFTRinh-172 to ligated ductules derived from WT mice recapitulated the phenotype of the ductules derived from $Adgrg2^{-/Y}$ mice. (4A-M)*p<0.05, **p<0.01, ***p<0.001; $Adgrg2^{-/Y}$ mice compared with WT mice. #p<0.05, ##p<0.01, ###p<0.001. Treatment with selective inhibitors or stimulators was compared with control vehicles. n.s., no significant difference. At least three independent biological replicates were performed for **Figure 4A–M**.

DOI: https://doi.org/10.7554/eLife.33432.009

The following figure supplement is available for figure 4:

**Figure supplement 1.** Expression and functional analysis of potential osmotic drivers in efferent ductules.
DOI: https://doi.org/10.7554/eLife.33432.010

were significantly impaired (**Figure 9A–C** and **Figure 9—figure supplement 1**). Moreover, whereas ADGRG2 and CFTR co-localized in the apical membrane regions of the non-ciliated cells of the efferent ductules derived from $Arrb2^{-/-}$ or WT mice, they were separated in $Arrb1^{-/-}$ mice (**Figure 9D–K**). In β-arrestin-1-deficient efferent ductules, CFTR localized away from ezrin (**Figure 9F–K**), an apical membrane marker, suggesting that β-arrestin-1 is required for the correct localization of CFTR. Consistently, whereas CFTR was co-immunoprecipitated with ADGRG2 in WT and $Arrb2^{-/-}$ mice, it was not found in ADGRG2-immunoprecipitated complexes from the efferent ductules derived from $Arrb1^{-/-}$ mice, further suggesting that β-arrestin-1 is an essential component in a signaling complex encompassing ADGRG2 and CFTR in the efferent ductules (**Figures 5H** and **9L** and **Figure 9—figure supplement 2**).

We therefore used HEK293 cells to investigate the in vitro role of β-arrestins in ADGRG2/CFTR complex formation. Overexpression of β-arrestin-1 but not β-arrestin-2 promoted the interaction between ADGRG2 and CFTR (**Figure 9—figure supplement 3**), confirming the essential role of β-arrestin-1 in assembly of ADGRG2/CFTR coupling.

## Molecular determinants of ADGRG2 coupling with G protein subtypes and their contribution to the regulation of CFTR activity in vitro

ADGRG2 belongs to the adhesion GPCR group of the GPCR superfamily (**Purcell and Hall, 2018**; **Monk et al., 2015**). Whereas the endogenous ligand of ADGRG2 in the testis is unknown, several members of the same adhesion GPCR subfamily, such as VLGR1 and GPR56, showed constitutive activity via overexpression in a heterologous system (**Purcell and Hall, 2018**; **Hu et al., 2014**; **Paavola et al., 2011**). To dissect the molecular mechanism underlying ADGRG2 signaling in the modulation of CFTR functions, we overexpressed ADGRG2 and CFTR in HEK293 cells (**Figure 10—figure supplement 1**). In vitro, the overexpression of ADGRG2 causes constitutive Gs and Gq coupling activity; a stronger effect is observed with ADGRG2β (**Figure 10—figure supplements 2–5**). Whole-cell recordings were performed to examine the effects of ADGRG2 and CFTR co-expression on membrane currents by using an I-V analysis (**Figure 10A**). The co-expression of ADGRG2 and CFTR significantly increased the amplitude and slope of the current responses, which were significantly reduced by the CFTR inhibitor CFTRinh-172, compared with cells transfected with CFTR alone, indicating that CFTR channels are activated by ADGRG2 in a recombinant system (**Figure 10B–D**). Similar to primary efferent ductule cells (**Figure 7F–G**), the application of FSK and IBMX further increased the whole-cell Cl⁻ current in the presence of both ADGRG2 and CFTR, confirming that ADGRG2 increased the basal activity of CFTR but did not stimulate CFTR to a full activation state (**Figure 10B–C** and **Figure 10—figure supplement 5A**).

Importantly, increased CFTR activity induced by ADGRG2 was significantly diminished by the PKC inhibitor Ro 31-8220 (**Figure 10D** and **Figure 10—figure supplement 5B–D**). Taken together, these

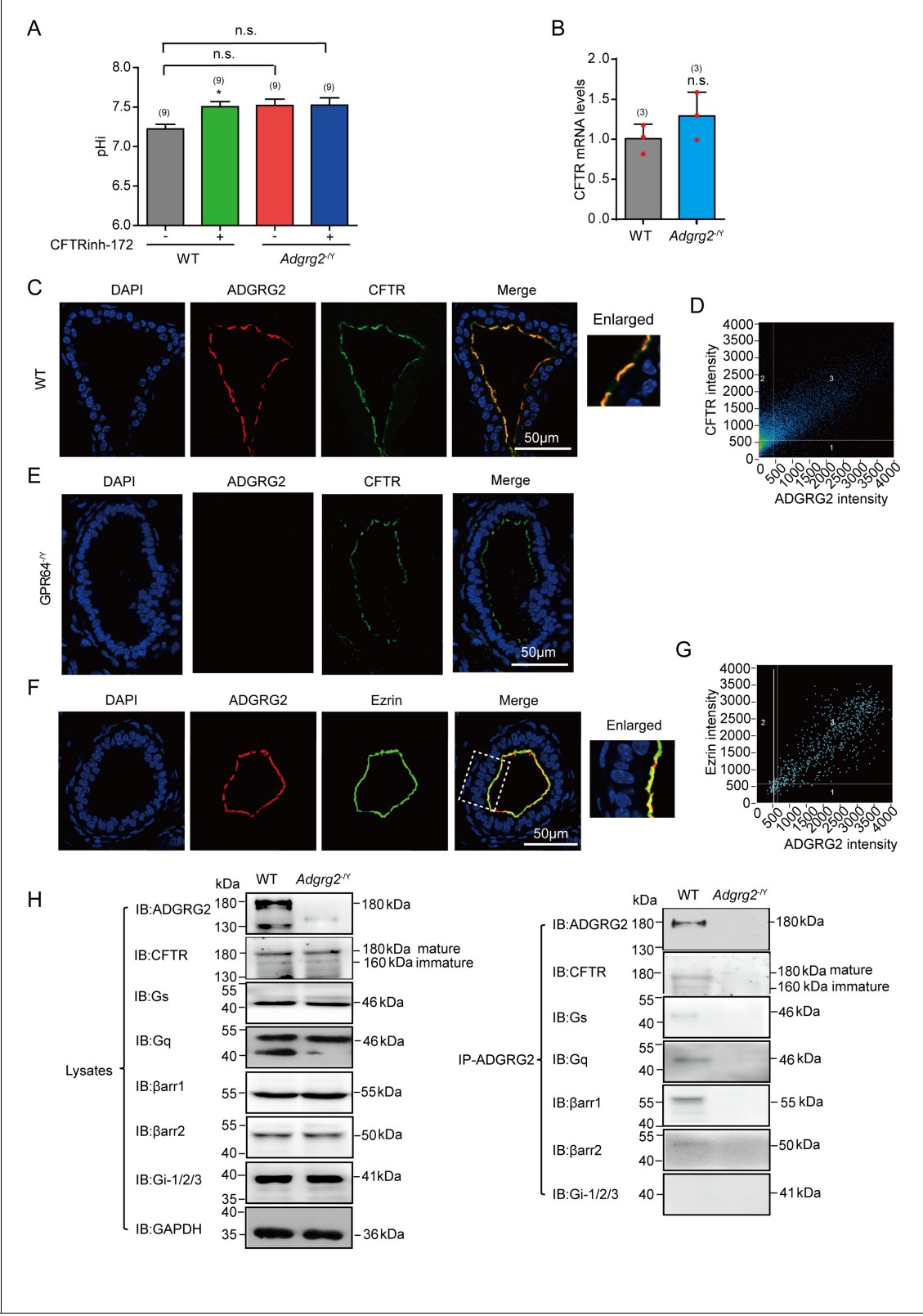

**Figure 5.** Functional coupling and co-localization of CFTR and ADGRG2 on the apical membrane in the efferent ductules. (**A**) Intracellular pH (pHi) of the ligated efferent ductules from WT (n = 9) mice and *Adgrg2⁻/Y* (n = 9) mice were measured by carboxy-SNARF (5 μM), with or without incubation with the CFTR inhibitor CFTRinh-172. (**B**) qRT-PCR analysis of CFTR levels in the efferent ductules of WT (n = 3) or *Adgrg2⁻/Y* (n = 3) mice. (**C**) Co-localization of ADGRG2 (red fluorescence) and CFTR (sc-8909, Santa Cruz, green fluorescence) in the male efferent ductules of WT mice. Scale bars, 50 μm. (**D**)

*Figure 5 continued on next page*

*Figure 5 continued*

Analysis of ADGRG2 and CFTR fluorescence intensities by Pearson's correlation analysis. The Pearson's correlation coefficient was 0.76. (**E**) Immunofluorescence staining of ADGRG2 (red fluorescence) and CFTR (sc-8909, Santa Cruz, green fluorescence) in the efferent ductules of *Adgrg2*$^{-/Y}$ mice. Scale bars, 50 μm. (**F**) Co-localization of ADGRG2 (red fluorescence) and ezrin (green fluorescence) in the male efferent ductules of WT mice. Scale bars, 50 μm. (**G**) Analysis of ADGRG2 and ezrin fluorescence intensities by Pearson's correlation analysis. The Pearson's correlation coefficient was 0.69. (**H**) ADGRG2 was immunoprecipitated with an anti-ADGRG2 antibody from the male efferent ductules of WT mice or *Adgrg2*$^{-/Y}$ mice, and co-precipitated CFTR, Gs, Gq, β-arrestin-1, β-arrestin-2 and Gi-1/2/3 levels were examined by using specific corresponding antibodies (CFTR antibody:20738–1-AP, Proteintech). (5A-5B) *p<0.05, **p<0.01, ***p<0.001, *Adgrg2*$^{-/Y}$ mice compared with WT mice. #p<0.05, ##p<0.01, ###p<0.001. Treatment with selective inhibitors or stimulators was compared with control vehicles. n.s., no significant difference. At least three independent biological replicates were performed for *Figure 5A–B*.

DOI: https://doi.org/10.7554/eLife.33432.011

The following figure supplements are available for figure 5:

**Figure supplement 1.** Representative agrose gel for the reverse transcription PCR analysis of CFTR mRNA level in efferent ductules of WT or *Adgrg2*$^{-/Y}$ mice.

DOI: https://doi.org/10.7554/eLife.33432.012

**Figure supplement 2.** pH homeostasis in the efferent ductules was impaired in *Adgrg2*$^{-/Y}$ mice.

DOI: https://doi.org/10.7554/eLife.33432.013

**Figure supplement 3.** Immunostaining experiments for CFTR location in efferent ductules.

DOI: https://doi.org/10.7554/eLife.33432.014

**Figure supplement 4.** Bar graph representation and statistical analyses of *Figure 5H*.

DOI: https://doi.org/10.7554/eLife.33432.015

data demonstrate that ADGRG2 increases CFTR Cl⁻ currents through the activation of Gq-PLC-PKC signaling.

Previous crystallographic studies have shown that the intracellular loop 2 of the β2-adrenergic receptor is important for Gs coupling, and mutations in the intracellular loop three affect G protein coupling activity by receptors (*Hu et al., 2014*; *Rasmussen et al., 2011*). We therefore selected mutations in intracellular loops 2 and 3 and examined their effects on the constitutive activity of ADGRG2 in Gs or Gq signaling, as detected by cAMP or NFAT-dual-luciferase reporter (DLR) luciferase measurements (*Figure 11A–C* and *Figure 11—figure supplement 1*) in HEK293 cells. Under the equal expression of these mutants in the cell membrane, a double mutation in the 'DRY' motif H696A/M697A of ADGRG2 eliminated coupling activity with both Gs and Gq (*Figure 11B–C* and *Figure 11—figure supplement 2*). Three mutations in intracellular loop 2, specifically Y698A and F705A, significantly impaired the Gs coupling activity of ADGRG2 but did not exert significant effects on NFAT-DLR activity (*Figure 11B–C*). However, Y708A in intracellular loop 2 and R803E/K804E in intracellular loop 3 nearly abolished the Gq coupling activity of ADGRG2 but did not have significant effects on intracellular cAMP levels compared with the WT ADGRG2. Thus, the 'DRY/HMY' motif mutant is a G-protein dysfunctional mutant for both Gs and Gq signaling, Y698A and F705A are specific Gs-defective mutants, and Y708A and R803E/K804E are specific Gq-defective mutants of ADGRG2 (*Figure 11B–C*).

The coupling of these ADGRG2 mutants to CFTR activity was then examined using the whole-cell recording technique. Voltage clamps were used to generate the I-V relationships of the CFTR currents in cells co-transfected with CFTR and ADGRG2 (*Figure 11D–F* and *Figure 11—figure supplement 3*). Interestingly, although the mutant with a specific Gs signaling defect showed decreased coupling of ADGRG2 to CFTR, the Gq-dysfunctional mutant and the H696A/M697A double Gs/Gq signaling-defective mutant did not demonstrate coupling between ADGRG2 and CFTR (*Figure 11D–F*). Taken together, these results demonstrate that specific residues in intracellular loops 2 and 3 are determinants of the G protein subtype coupling of ADGRG2. Furthermore, downstream of ADGRG2, Gq signaling is essential for CFTR activation in recombinant in vitro systems.

## Effects of the conditional expression of WT-ADGRG2 or its selective G-subtype signaling mutants on the rescue of reproductive defects in *Adgrg2*$^{-/Y}$ mice

We next examined how the molecular determinants of ADGRG2/G protein subtype interactions contribute to the function of ADGRG2 infertility in vivo. Both ADGRG2 WT and G protein subtype

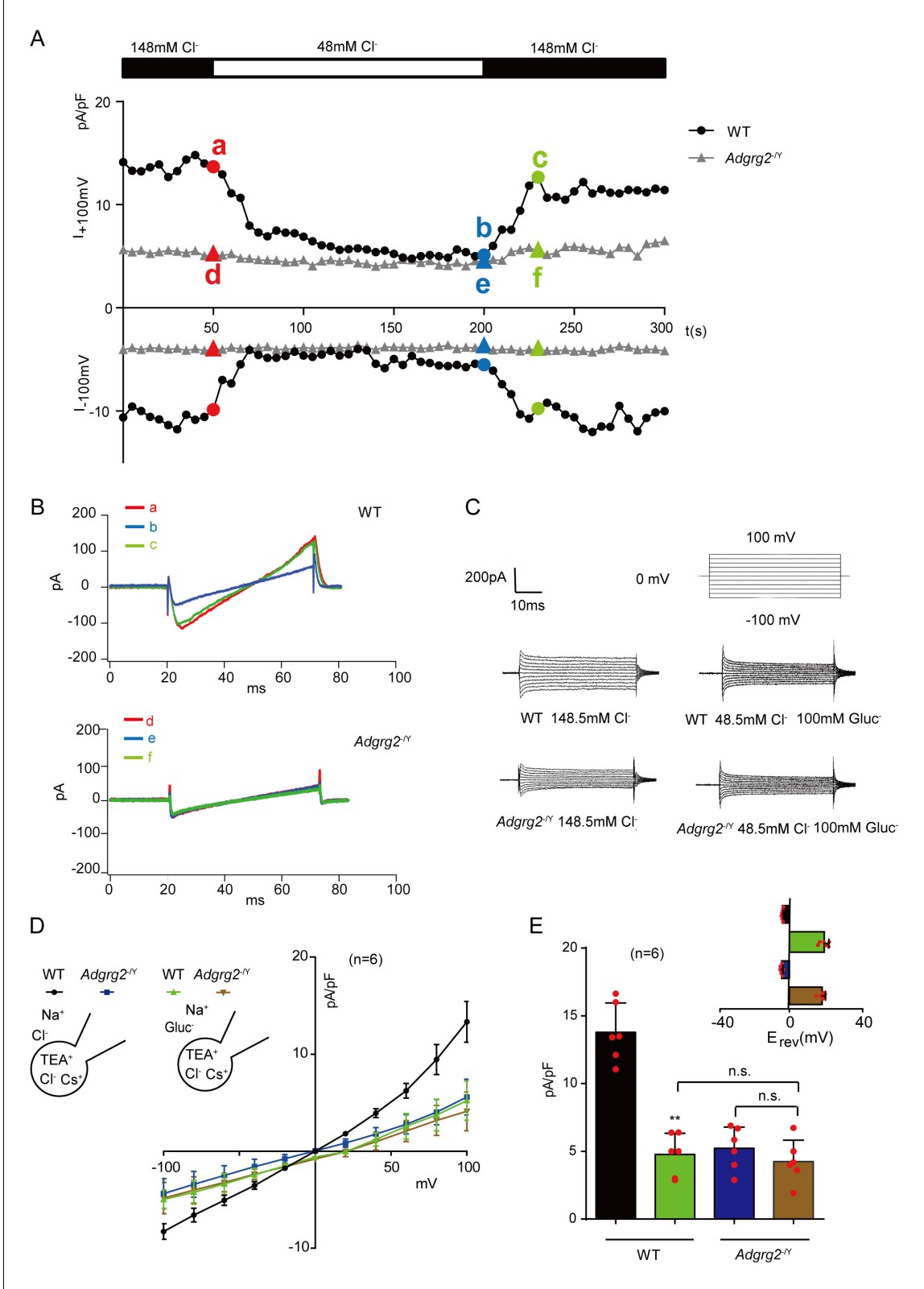

**Figure 6.** The whole-cell $Cl^-$ current recording of ADGRG2 promoter-labeled efferent ductule cells. (**A**) Time course of whole-cell $Cl^-$ current ($I_{ADGRG2-ED}$) at +100 and −100 mV in ADGRG2 promoter-labeled efferent ductule cells derived from *Adgrg2^-/Y* mice or their littermates. An 'a' or 'd' indicates the substitution of the $Cl^-$ bath solution with $Gluc^-$ (148.5 mM $Cl^-$ was replaced by 48.5 mM $Cl^-$ and 100 mM $Gluc^-$); and 'b' or 'e' indicates the substitution of the $Gluc^-$ bath solution with $Cl^-$ (148.5 mM $Cl^-$). 'a','"b' and 'c' belong to WT mice. 'd','"e' and 'f' belong to *Adgrg2^-/Y* mice. (**B**) The current-voltage

*Figure 6 continued on next page*

*Figure 6 continued*

relationship of $I_{ADGRG2-ED}$ at specific time points (from 6A) is shown. (C) The whole cell Cl$^-$ current of $I_{ADGRG2-ED}$ elicited by voltage steps between −100 mV and +100 mV in a representative ADGRG2-promoter-RFP labeled efferent ductule cells derived from the *Adgrg2$^{-/Y}$* mice and their wild type littermates. The outwardly rectifying $I_{ADGRG2-ED}$ was significantly diminished when bath Cl$^-$ was substituted for gluconate (Gluc$^-$). (D) Representative whole-cell Cl$^-$ current of ADGRG2 promoter-labeled efferent ductule cells; $I_{ADGRG2-ED}$ versus voltage (I–V) relationships in response to voltage ramps recorded with a CsCl pipette solution in *Adgrg2$^{-/Y}$* (n = 8) or WT mice (n = 8). The outwardly rectifying $I_{ADGRG2-ED}$ was significantly diminished, and its reversal potential ($E_{rev}$) shifted to the positive direction when Cl$^-$ was substituted for Gluc$^-$. (E) Average current densities (pA/pF) measured at 100 mV of (C). Inset: average Erev (±s.e.m., n = 8 for each condition). **p<0.01, $I_{ADGRG2-ED}$ in Gluc$^-$ solution was compared with $I_{ADGRG2-ED}$ in Cl$^-$ solution. ns, no significant difference. At least three independent biological replicates were performed.

DOI: https://doi.org/10.7554/eLife.33432.016

mutants were conditionally expressed in the efferent ductules via virus infection under the 1 kb ADGRG2 promoter (*Figure 12A*). Similar to ADGRG2 WT mice, exogenously introduced ADGRG2 WT and mutants specifically localized to the inner surface of the non-ciliated cells of the efferent ductules (*Figure 12—figure supplement 1A*).

The efferent ductules of *Adgrg2$^{-/Y}$* animals frequently exhibited the accumulation of obstructed spermatozoa compared with observations in WT mice (*Figure 12B*). The conditional expression of ADGRG2 in non-ciliated cells in *Adgrg2$^{-/Y}$* mice significantly reduced this obstruction, whereas the expression of G protein signaling-deficient mutants of ADGRG2, including Y698A, F705A, Y708A, RK803EE and HM696AA, significantly reduced this rescue effect (*Figure 12B–E* and *Figure 12—figure supplement 1B–C*). Specifically, conditional infection of the Gs/Gq double signaling-deficient mutant ADGRG2-HM696AA or the Gq signaling-deficient mutants Y708A and RK803EE did not result in differing levels of accumulation in the efferent ductules compared with those in *Adgrg2$^{-/Y}$* mice infected with a control virus. The Gs signaling-deficient mutants Y698A and F705A exhibited improved rescue activity compared with the Gq mutants (*Figure 12B–E* and *Figure 12—figure supplement 1B–C*).

Consistent with observations in the efferent ductules, the lumen of the initial segment and caput region in *Adgrg2$^{-/Y}$* mice showed reduced sperm numbers compared with those in WT mice (*Figure 12B,D–E* and *Figure 12—figure supplement 1C*). The exogenous introduction of WT ADGRG2 to non-ciliated cells nearly restored the appearance of sperm in the initial segment and significantly increased sperm numbers in the caput (*Figure 12B and E* and *Figure 12—figure supplement 1C*). However, introducing any of the Gs or Gq signaling-deficient mutants into *Adgrg2$^{-/Y}$* mice did not induce a significant effect on sperm number restoration in these regions (*Figure 12B and E* and *Figure 12—figure supplement 1C*).

Sperm prepared from the caudal epididymis were then examined. *Adgrg2$^{-/Y}$* mice exhibited significantly reduced sperm numbers and presented morphologically abnormal sperm compared with those of WT mice (*Figure 12C* and *Figure 12—figure supplement 1B*). Conditional expression of WT ADGRG2 in the efferent ductules restored sperm numbers in the caudal epididymis by more than half, whereas exogenous introduction of the two Gs-deficient mutants Y698A and F705A into *Adgrg2$^{-/Y}$* mice increased sperm numbers by 5–10% compared with those in *Adgrg2$^{-/Y}$* mice. Expression of the other 4 Gs-, Gq- or double-deficient mutants did not rescue the phenotype (*Figure 12C*).

To investigate whether the sperm production phenotype was related to fluid reabsorption, we isolated the efferent ductules after virus infection with the WT ADGRG2 or one of the mutants and measured the luminal area after ligation. Interestingly, conditional expression of the Gs-deficient

**Table 1.** Average reversal potential calculated at different Cl$^-$ concentrations for *Figure 6C*.
Average reversal potential($E_{rev}$) (±s.e.m., n = 8 for each condition) in *Figure 6C* and calculated Nernst potential at different Cl$^-$ concentrations. The Nernst equation was: $E_{rev}=-RT/Z$ [Ln (Cl$^-$)$_{in}$/(Cl$^-$)$_{out}$ ].

| Group | $E_{rev}$[Cl$^-$]$_o$148.5 mM(mV) | $E_{rev}$[Cl$^-$]$_o$48.5 mM(mV) |
|---|---|---|
| Nernst | −4.6 | 25.3 |
| WT | −4.0 ± 0.51 | 20.1 ± 2.52 |
| *Adgrg2$^{-/Y}$* | −4.1 ± 0.36 | 19.4 ± 2.47 |

DOI: https://doi.org/10.7554/eLife.33432.017

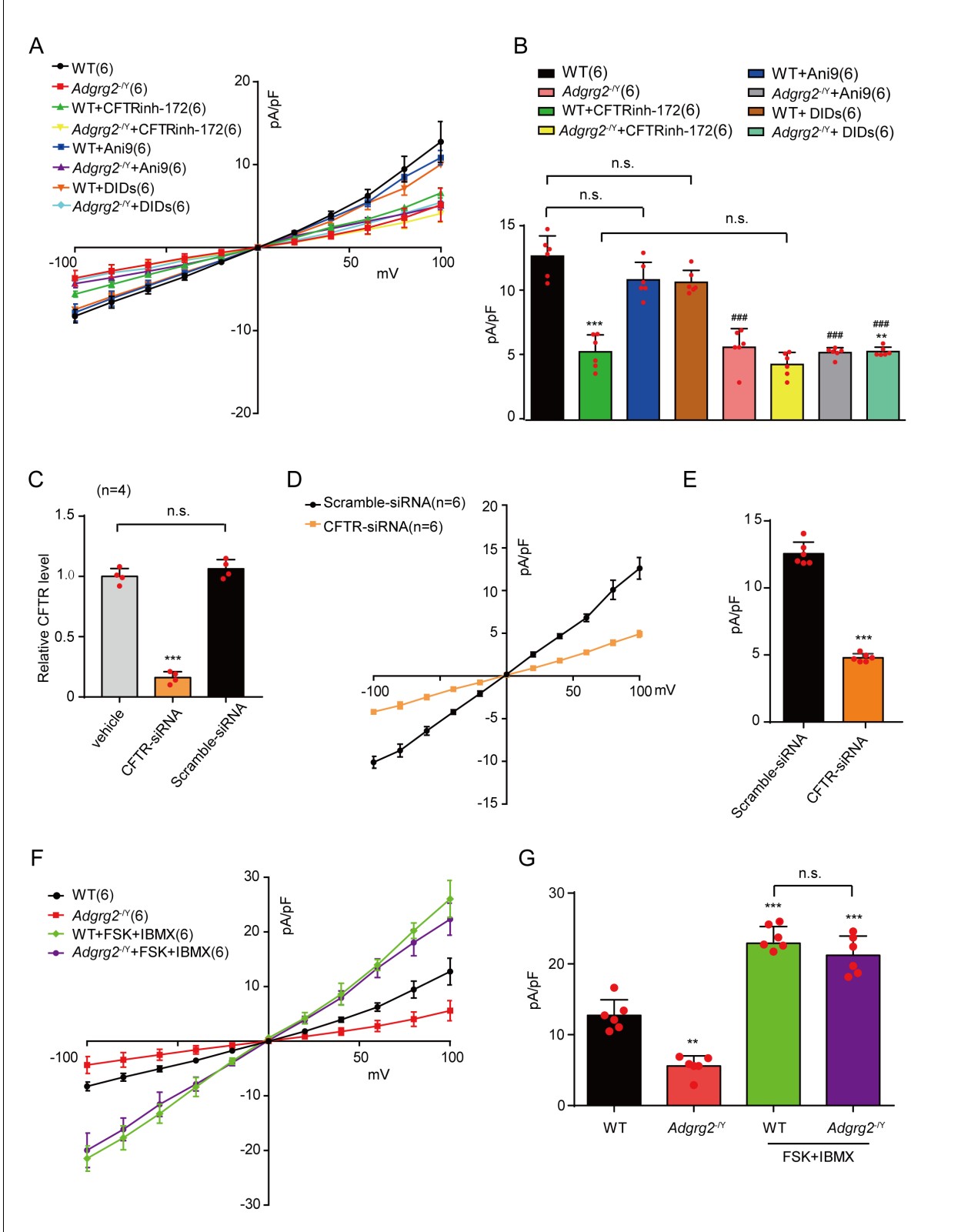

**Figure 7.** Cl⁻ currents in the non-ciliated cells of the efferent ductules through CFTR. (**A, D and F**) Corresponding I-V curves of the whole-cell Cl⁻ $I_{ADGRG2-ED}$ currents recorded in *Figure 6* and (**A, D and F**) Corresponding I-V curves of the whole-cell Cl- IADGRG2-ED currents recorded in *Figure 7— figure supplement 1(A,F)* and *Figure 7—figure supplement 3(D)*. WT (n = 6), *Adgrg2⁻/Y* (n = 6); WT +CFTRinh-172 (n = 6), *Adgrg2⁻/Y*+CFTRinh-172 (n = 6), WT +ANI9 (n = 6), *Adgrg2⁻/Y*+ANI9 (n = 6), WT +DIDS (n = 6), *Adgrg2⁻/Y*+DIDS (n = 6); WT +Control RNAi (n = 6), WT +CFTR RNAi (n = 6), *Figure 7 continued on next page*

*Figure 7 continued*

*Adgrg2$^{-/Y}$*+Control RNAi (n = 6), *Adgrg2$^{-/Y}$*+CFTR RNAi (n = 6); WT +FSK + IBMX (n = 6), *Adgrg2$^{-/Y}$*+FSK+IBMX (n = 6). (B,E and G) Corresponding bar graph depicting the average current densities (pA/pF) measured at 100 mV in (A), (D) and (F). (C) qRT-PCR analysis of CFTR levels in the efferent ductules treated with CFTR siRNA (n = 3) or control RNAi (n = 3). (B, E and G) *$p < 0.05$, **$p < 0.01$, ***$p < 0.001$, *Adgrg2$^{-/Y}$* mice compared with WT mice. #$p < 0.05$, ##$p < 0.01$, ###$p < 0.001$. Treatment with selective inhibitors, stimulators or CFTR RNAi was compared with control vehicles or control RNAi. n.s., no significant difference. At least three independent biological replicates were performed for *Figure 7B,E and G*.

DOI: https://doi.org/10.7554/eLife.33432.018

The following figure supplements are available for figure 7:

**Figure supplement 1.** Effects of different stimulators or inhibitors of osmotic drivers on I $_{ADGRG2-ED}$ Cl$^-$ currents of efferent ductule cells derived from *Adgrg2$^{-/Y}$* mice and their wild type littermates.

DOI: https://doi.org/10.7554/eLife.33432.019

**Figure supplement 2.** Effects of Cl$^-$ concentration change and CFTRinh-172 on the I$_{ADGRG2-ED}$ Cl$^-$ currents.

DOI: https://doi.org/10.7554/eLife.33432.020

**Figure supplement 3.** Effects of CFTR knocked down on the I $_{ADGRG2-ED}$ Cl$^-$ currents.

DOI: https://doi.org/10.7554/eLife.33432.021

mutations Y698A and F705A marginally reduced the inflation of the efferent ductules of *Adgrg2$^{-/Y}$* mice, whereas the Gq signaling mutants did not exert significant effects on the luminal volume (*Figure 13A–F*). This result is consistent with the effects of these mutants on sperm numbers in the caudal epididymis, thereby suggesting a direct correlation between efferent ductule reabsorption ability and mature sperm numbers (*Figures 12C* and *13A–F*). Taken together, our results demonstrate that Gq activity is required downstream of ADGRG2, and Gs function contributes to fluid reabsorption in the efferent ductules and sperm transportation.

## Discussion

Fluid reabsorption is the main function of the efferent ductules and is essential for sperm maturation; it therefore serves as a promising target for the development of new contraceptive methods for men (*Hess, 2002*). The cell surface orphan receptor ADGRG2 is an X-linked gene specifically expressed in the reproductive system, and recent studies have found that its deficiency results in the dysfunction of fluid reabsorption and male fertility. However, the mechanism by which fluid reabsorption is regulated by ADGRG2 in the efferent ductules remains unclear (*Davies et al., 2004*). ADGRG2 belongs to the adhesion GPCR (aGPCRs) subfamily, whose members are either structurally essential in specific tissues (VLGR1 participates in forming the ankle link) or critical signaling molecules in the nervous and immune systems (GPR56, CD97 and EMRs) (*Purcell and Hall, 2018*; *Sun et al., 2013*; *Sun et al., 2016*). Although the efferent ductules of *Adgrg2$^{-/Y}$* mice exhibit normal morphology, our results here have identified essential signaling roles for ADGRG2 in non-ciliated cells of the efferent ductules to maintain pH homeostasis as well as the basic CFTR outward-rectifying current, which is required for fluid reabsorption and sperm maturation. Currently, there have been no reported endogenous ADGRG2 ligands. While an unknown ADGRG2 agonist may be responsible for ADGRG2 function in the efferent ductules, it is also likely that the constitutive activity of ADGRG2 in non-ciliated cells is sufficient to maintain the basic CFTR current and pH homeostasis, which is supported by our data using both primary ADGRG2 promoter-labeled efferent ductule cells and a recombinant heterologous HEK293 system (*Figures 5–7* and *Figure 10*). Therefore, our results provide an example of the functional relevance of the constitutive activity of aGPCRs. Moreover, there are several examples indicating the constitutive activity of aGPCRs is tunable by mechanical stimulation (*Purcell and Hall, 2018*; *Petersen et al., 2015*; *Scholz et al., 2015*). As ADGRG2 was expressed in efferent ductules that were controlled by extensive tension, it will be interesting to investigate the effects of tension on ADGRG2 functions in future studies.

Downstream of GPCRs, 16 different G protein subtypes and arrestins play important roles in almost every aspect of human physiological processes (*Liu et al., 2017*; *Ning et al., 2015*; *Yang et al., 2015*, *2017*). However, the expression and function of five different G protein subtypes as well as arrestins in the efferent ductules have never been systematically investigated. Here, we have determined that the majority of G protein subtypes are expressed in the efferent ductules (*Figure 1A and D*). Gq activity is essential for male fertility by maintaining basic CFTR activity and

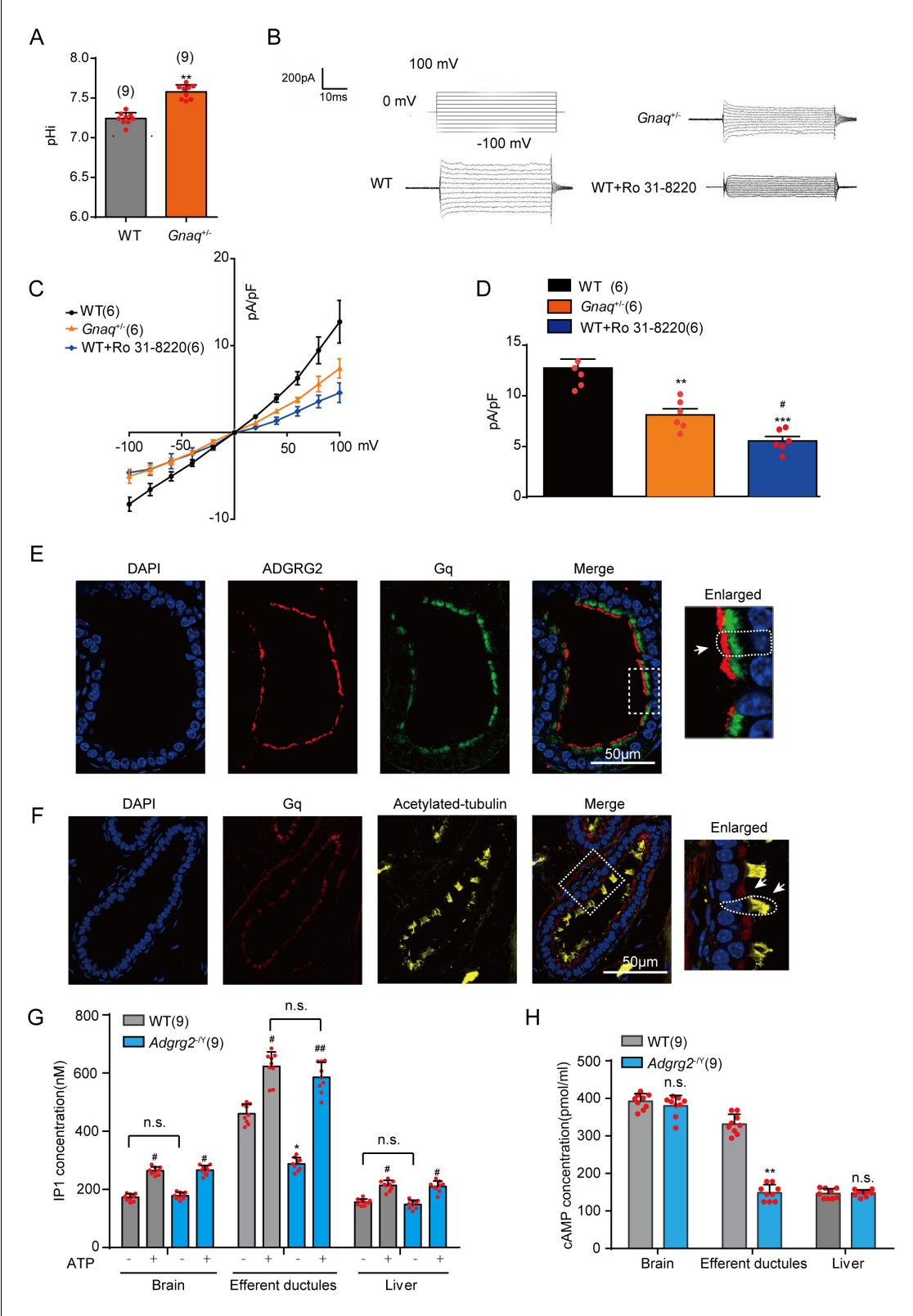

**Figure 8.** Gq activity regulated Cl⁻ current and pH homeostasis in the efferent ductules. (**A**) Intracellular pH (pHi) of the ligated efferent ductules from WT (n = 9) mice or *Gnaq*⁺/⁻ (n = 9) mice was measured by carboxy-SNARF. (**B**). The whole-cell Cl⁻ current of the I$_{ADGRG2-ED}$ elicited by voltage steps between −100 mV and +100 mV in representative ADGRG2 promoter-RFP-labeled efferent ductule cells derived from *Gnaq*⁺/⁻ mice, their WT littermates, or WT murine cells incubated with the PKC inhibitor Ro 31–8220 (500 nM). The whole-cell Cl⁻ I$_{ADGRG2-ED}$ current was recorded with a CsCl

*Figure 8 continued on next page*

Figure 8 continued

pipette solution (101 mM CsCl, 10 mM EGTA, 10 mM Hepes, 20 mM TEACl, 2 mM MgATP, 2 mM MgCl₂, 5.8 mM glucose, pH7.2, with D-mannitol compensated for osm 290) and a bath solution containing 138 mM NaCl, 4.5 mM KCl, 2 mM CaCl₂, 1 mM MgCl₂, 5 mM glucose, and 10 mM HEPES, pH 7.4 with D-mannitol compensated for osm 310. (C) Corresponding I-V curves of the whole-cell Cl- currents recorded in (B). WT (n = 6), $Gnaq^{+/-}$ (n = 6), WT +Ro 31–8220 (n = 6). (D) Corresponding bar graph of the average current densities (pA/pF) measured at 100 mV according to (C). (E) Co-localization of ADGRG2 (red) and Gq (green) in the male efferent ductules. Scale bars, 50 μm. (F) Co-localization of Gq (red) and acetylated-tubulin (yellow) in the male efferent ductules. Scale bars, 50 μm. (G) IP1 levels in the brain tissues, ligated efferent ductules, and livers of WT (n = 9) or $Adgrg2^{-/Y}$ (n = 9) mice in response to ATP (5 mM) or control vehicles, measured by ELISA. (H) cAMP concentrations in the brains, ligated efferent ductules, and livers of WT (n = 9) or $Adgrg2^{-/Y}$ (n = 9) mice were measured using ELISA. (8A,8D,8G-H) *p<0.05, **p<0.01, ***p<0.001, $Adgrg2^{-/Y}$ mice or $Gnaq^{+/-}$ mice compared with WT mice. #p<0.05, ##p<0.01, ###p<0.001, ATP- or Ro 31–8220-treated cells were compared with control vehicles. n.s., no significant difference. At least three independent biological replicates were performed for *Figure 8A,D,G and H*.

DOI: https://doi.org/10.7554/eLife.33432.022

The following figure supplements are available for figure 8:

**Figure supplement 1.** Effects of G protein signaling on the I$_{ADGRG2-ED}$ Cl⁻ currents.

DOI: https://doi.org/10.7554/eLife.33432.023

**Figure supplement 2.** Gq is localized in the ADGRG2 expressed cells, but not the acetylated-tubulin-labeled cells in efferent ductules.

DOI: https://doi.org/10.7554/eLife.33432.024

**Figure supplement 3.** The expression of ADGRG2, CFTR, Gs, Gq, β-arrestin-1, β-arrestin-2 in efferent ductules, brain and liver tissue of WT and $Adgrg2^{-/Y}$ mice.

DOI: https://doi.org/10.7554/eLife.33432.025

pH homeostasis in the efferent ductules (*Figure 14*). In particular, specific residues in intracellular loops 2 and 3 are structural determinants of the ADGRG2/Gq interaction (*Figures 11–14*), which mediates the constitutive activity of Gq-PLC-IP3 signaling in non-ciliated cells of the efferent ductules.

Notably, we found that ADGRG2 and Gq regulate fluid reabsorption in the efferent ductules via the activation of CFTR, an important ion channel whose mutation leads to cystic fibrosis (CF). One of the hallmarks of CF is infertility (*Cutting, 2015*; *Massie et al., 2014*), which has a 97–98% incidence rate in male CF patients (*Chen et al., 2012*). CFTR knockout and the application of specific CFTR inhibitors in animal models indicate that CFTR plays important roles in spermatogenesis and sperm capacitation (*Chen et al., 2012*). Here, we demonstrated the specific co-localization of ADGRG2 and CFTR in the apical membrane in the non-ciliated cells of the efferent ductules (*Figure 5* and *Figure 9*). CFTR was basically active in ADGRG2 promoter-labeled efferent ductule cells, and this activity was significantly decreased by ADGRG2 or Gq deficiency. The application of a specific CFTR inhibitor, CFTRinh-172, consistently pheno-copied the ligated efferent ductules of $Adgrg2^{-/Y}$ mice (*Figure 4K*). Further pharmacological intervention in the efferent ductules and recombinant experiments in vitro confirmed the coupling of ADGRG2 and CFTR activity through Gq. Moreover, previous studies have shown that PKC phosphorylation is required for subsequent PKA phosphorylation to fully activate CFTR (*Chappe et al., 2004*; *Jia et al., 1997*). Our study not only agreed with the observation that PKC and PKA lie downstream of Gq and Gs, respectively, but also suggested that ADGRG2-activated Gq primes the full activation of CFTR in the efferent ductules. Therefore, our results demonstrate that the physiological and functional coupling of ADGRG2 and CFTR mediated by Gq in the non-ciliated cells of the efferent ductules primes the basic activity of CFTR, which is essential for fluid reabsorption. The ADGRG2-Gq-CFTR signaling axis is important to maintain male reproductive functions (*Figure 14*).

Parallel to G protein signaling, β-arrestins are known to play important roles in almost all GPCR functions (*Cahill et al., 2017*; *Dong et al., 2017*; *Liu et al., 2017*; *Yang et al., 2017*). Knockout of β-arrestin-1 but not β-arrestin-2 abolished the co-localization of ADGRG2 and CFTR, demonstrating the essential role of β-arrestin-1 in assembling ADGRG2/CFTR/Gq signaling compartmentalization to regulate Cl⁻ and pH homeostasis during fluid reabsorption in the efferent ductules. For decades, GPCR/β-arrestin complexes were thought to play fundamental roles in the internalization and desensitization of G protein signaling. Recently, a mega complex encompassing the GPCR, G trimer proteins and β-arrestins was identified by using an in vitro reconstruction system in HEK293 cells to provide a new paradigm of GPCR signaling (*Thomsen et al., 2016*). Consistently, we identified the ability of β-arrestin-1 to facilitate ADGRG2/Gq/CFTR signaling compartmentalization, which

indicated that such a receptor/G protein/β-arrestin mega complex plays important roles in the regulation of important physiological processes, such as fluid reabsorption in the efferent ductules.

Finally, our results suggest that the inhibition of either CFTR or ADGRG2 impairs the resorptive function of the efferent ductules, which may confer a contraceptive function. Indeed, anti-spermatogenic agents, such as indazole compounds, block CFTR activity (*Chen et al., 2005*; *Gong et al., 2002*). Compared with CFTR, which is broadly expressed and has important functions in many tissues, ADGRG2 is specifically expressed in the efferent ductules and epididymis. Contraceptive compounds targeting ADGRG2 may have fewer side effects. Moreover, the dysfunction of ADGRG2 or CFTR is rescued by the activation of AGTR2 in the efferent ductules (*Shum et al., 2008*). Therefore, a specific agonist of AGTR2 should be considered for the development of therapeutic methods to treat male infertility caused by impaired ADGRG2-Gq-CFTR signaling, such as that observed in CF patients.

# Materials and methods

**Key resources table**

| Reagent type (species) or resource | Designation | Source or reference | Identifiers | Additional information |
|---|---|---|---|---|
| Chemical compound, drug | PTX | Enzo | Cat#:BML-G100 | 100 ng/ml |
| Chemical compound, drug | U0126 | Sigma | Cat#:U120 | 10 µM |
| Chemical compound, drug | Ro 31–8220 | Adooq | Cat#:A13514 | 500 nM |
| Chemical compound, drug | NF449 | Tocris | Cat#:1391 | 1 µM |
| Chemical compound, drug | PKI14-22 | Adooq | Cat#:A16031 | 300 nM |
| Chemical compound, drug | H89 | Beyotime | Cat#:S1643 | 500 nM |
| Chemical compound, drug | bumetanide | Aladdin | Cat#:B129942 | 10 µM |
| Chemical compound, drug | Ani9 | Sigma | Cat#:SML1813 | 150 nM |
| Chemical compound, drug | Niflumic acid (NFA) | Aladdin | Cat#:N129597 | 20 µM |
| Chemical compound, drug | DIDS | Sigma | Cat#:D3514 | 20 µM |
| Chemical compound, drug | GlyH-101 | Adooq | Cat#:A13723 | 10 µM |
| Chemical compound, drug | CFTRinh-172 | Adooq | Cat#:A12897 | 10 µM |
| Chemical compound, drug | EGTA | Aladdin | Cat#:E104434 | 5 mM |
| Chemical compound, drug | SKF96365 | Sigma | Cat#:S7809 | 10 µM |
| Chemical compound, drug | Ruthenium red | Sigma | Cat#:R2751 | 10 µM |
| Chemical compound, drug | Nicardipine | Sigma | Cat#:N7510 | 20 µM |
| Chemical compound, drug | LaCl3 | Sigma | Cat#:449830 | 100 µM |
| Chemical compound, drug | IBMX | Sigma | Cat#:I7018 | 100 µM |
| Chemical compound, drug | U73122 | Sigma | Cat#:U6756 | 10 µM |

*Continued on next page*

*Continued*

| Reagent type (species) or resource | Designation | Source or reference | Identifiers | Additional information |
|---|---|---|---|---|
| Chemical compound, drug | Forskolin | Beyotime | Cat#:S1612 | 10 µM |
| Chemical compound, drug | PD123319 | Adooq | Cat#:A13201 | 1 µM |
| Chemical compound, drug | Candesartan | Adooq | Cat#:A10175 | 1 µM |
| Chemical compound, drug | Amiloride | Aladdin | Cat#:A129545 | 1 mM |
| Chemical compound, drug | Acetazolamide | Medchem express | Cat#:HY-B0782 | 500 µM |
| Peptide, recombinant protein | ANGII | China Peptides | | 100 nM |
| Commercial assay or kit | Carboxy SNARF—1, acetoxymethyl ester | Invitrogen | Cat#:C-1272 | 5 µM |
| Commercial assay or kit | Lipofectamine TM2000 | Invitrogen | Cat#:11668–019 | |
| Commercial assay or kit | Collagenase I | sigma | Cat#:C0130 | |
| Commercial assay or kit | cAMP ELISA kit | R and D systems | Cat#:KGE012B | |
| Commercial assay or kit | IP1 ELISA assay | Shanghai Lanpai Biotechnology Co., Ltd | Cat#:lp034186 | |
| Commercial assay or kit | The dual-luciferase reporter assay system | Promega | Cat#:E1960 | |
| Antibody | ADGRG2 antibody (rabbit polyclonal) | Sigma | RRID:AB_1078923 | |
| Antibody | ADGRG2 antibody (rabbit polyclonal) | Sigma | RRID:AB_2722557 | |
| Antibody | ADGRG2 antibody (sheep polyclonal) | R and D systems | RRID:AB_2722556 | |
| Antibody | CFTR antibody (goat polyclonal) | Santa Cruz | RRID:AB_638427 | |
| Antibody | CFTR antibody (rabbit polyclonal) | Proteintech | RRID:AB_2722558 | |
| Antibody | Gq antibody (goat polyclonal) | Santa Cruz | RRID:AB_2279038 | |
| Antibody | Gq antibody (rabbit polyclonal) | Proteintech | RRID:AB_2111647 | |
| Antibody | Flag antibody (mouse monoclonal) | Sigma | RRID:AB_259529 | |
| Antibody | HA antibody (mouse monoclonal) | Santa Cruz | RRID:AB_627809 | |
| Antibody | GAPDH (rabbit monoclonal) | Cell Signaling | RRID:AB_10622025 | |
| Antibody | Gs antibody (rabbit polyclonal) | Proteintech | RRID:AB_2111668 | |
| Antibody | Gi antibody (mouse monoclonal) | Santa Cruz | RRID:AB_2722559 | |
| Antibody | β-arrestin-1 antibody (rabbit polyclonal) | Dr R.J. Lefkowitz | A1CT | |
| Antibody | β-arrestin-2 antibody (rabbit polyclonal) | Dr R.J. Lefkowitz | A2CT | |

*Continued on next page*

*Continued*

| Reagent type (species) or resource | Designation | Source or reference | Identifiers | Additional information |
|---|---|---|---|---|
| Antibody | ANO1 antibody (rabbit polyclonal) | Proteintech | RRID:AB_2722560 | |
| Antibody | Ezrin antibody (rabbit polyclonal) | Proteintech | RRID:AB_2722561 | |
| Antibody | Acetylated Tubulin(Lys40) Antibody(mouse monoclonal) | Proteintech | RRID:AB_2722562 | |
| Antibody | Donkey anti-sheep IgG(H + L) (secondary antibody) | Abcam | RRID:AB_2716768 | |
| Antibody | Donkey anti-rabbit IgG(H + L) (secondary antibody) | Invitrogen | RRID:AB_2534017 | |
| Antibody | Donkey anti-mouse IgG(H + L) (secondary antibody) | Invitrogen | RRID:AB_141607 | |
| Antibody | Donkey anti-goat IgG(H + L) (secondary antibody) | Invitrogen | RRID:AB_142672, RRID:AB_141788 | |
| Antibody | HRP-conjugated Affinipure Rabbit Anti-Sheep IgG(H + L) | Proteintech | RRID:AB_2722563 | |
| Antibody | HRP-conjugated Affinipure Goat Anti-Rabbit IgG(H + L) | Proteintech | RRID:AB_2722564 | |
| Antibody | HRP-conjugated Affinipure Goat Anti-Rabbit IgG(H + L) | Proteintech | RRID:AB_2722565 | |

All other chemicals or reagents were from Sigma unless otherwise specified.

## Mice

Mice were individually housed in the Shandong University on a 12:12 light: dark cycle with access to food and water ad libitum. The use of mice were approved by the animal ethics committee of Shandong university medical school (protocol LL-201502036). All animal care and experiments were reviewed and approved by the Animal Use Committee of Shandong University, School of Medicine. *Adgrg2*$^{+/-}$ mice were obtained from Dr DLL and MYL at East China Normal University, Shanghai, China. *Adgrg2*$^{-/Y}$ mice and WT mice were generated by crossing WT (C57BL/6J) males mice and *Adgrg2*$^{+/-}$ females mice. *Arrb1*$^{-/-}$ and *Arrb2*$^{-/-}$ mice were obtained from Dr RJ Lefkowitz (Duke University, Durham, NC); *Arrb1*$^{-/-}$ and WT mice were generated by crossing *Arrb1*$^{+/-}$ male mice and *Arrb1*$^{+/-}$ female mice. *Arrb2*$^{-/-}$ and WT mice mice were generated by crossing *Arrb2*$^{+/-}$ male mice and *Arrb2*$^{+/-}$ female mice. *Gnaq*$^{+/-}$ mice were obtained from Dr JL Liu at Shanghai Jiao Tong University. *Gnaq*$^{+/-}$ mice and WT mice were generated by crossing *Gnaq*$^{+/-}$ male mice and *Gnaq*$^{+/-}$ female mice. All C57BL/6J male mice were purchased from Beijing Vital River Laboratory Animal Technology.

## Genotyping the *Adgrg2*$^{-/Y}$ KO mice

Genotyping of the intercrossed mice were examined using following primers: Fcon (Forward-control): TTTCATAGCCAGTGCTCACCTG, Fwt (Forward-wild-type): CCTGTTGGCAGACCTGAAG, Fmut (Forward-mutant): CTGTTGGCAGACCTTTTGTATATC, R (Reverse-general): CTTCCTAACATGTGCCATGGC. For the wild-type *Adgrg2*$^{+/Y}$ mice, Fcon, Fwt and R primers were used to generate two PCR products (189 bp, 397 bp); and Fcon, Fmut and R primers were used to generate one PCR product (397 bp). For the mutant *Adgrg2*$^{-/Y}$, Fcon, Fwt and R primers were used to generate one PCR product (405 bp); and Fcon, Fmut and R primers were used to generate two PCR products (196 bp, 405 bp). The female mice were genotyped by the same method. The knockout of ADGRG2 in these mice was confirmed by western blotting.

## Preparation of the membrane fraction of the epididymis and efferent ductules

The membrane fraction of the epididymis or efferent ductules was prepared from pooled mouse tissues (n = 4–6). These tissues (epididymis or efferent ductules) were dounced in a glass tube within

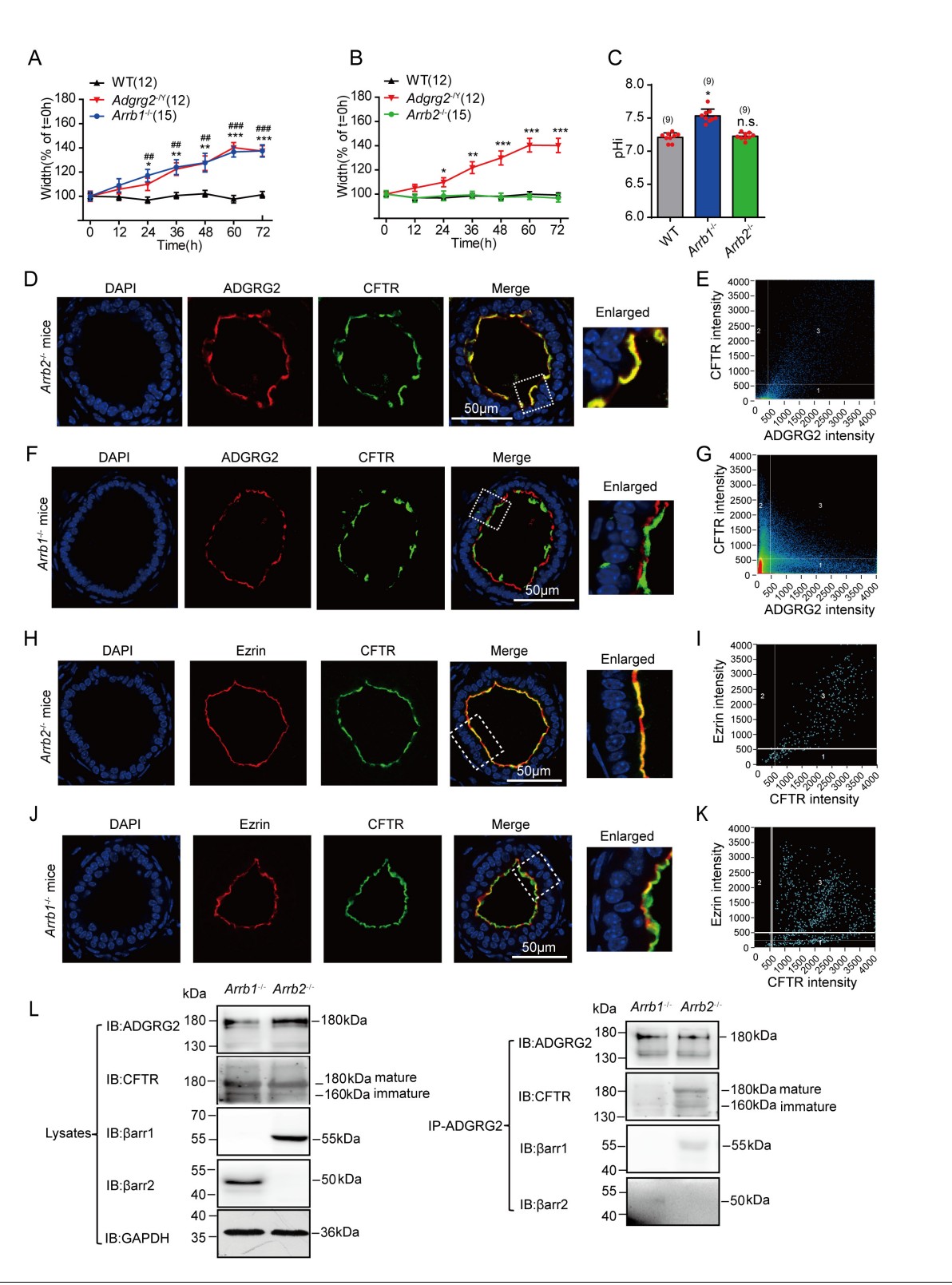

**Figure 9.** β-arrestin-1 is required for fluid reabsorption in the efferent ductules via scaffolding ADGRG2/CFTR complex formation. (**A**) Diameters of the luminal ductules derived from WT (n = 12), $Adgrg2^{-/Y}$ (n = 12) or $Arrb1^{-/-}$ (n = 15) mice. (**B**) Diameters of the luminal ductules derived from WT (n = 12), $Adgrg2^{-/Y}$ (n = 12) or $Arrb2^{-/-}$ (n = 15) mice. (**C**) Intracellular pH (pHi) of the ligated efferent ductules derived from WT (n = 9), $Arrb1^{-/-}$ (n = 9) or $Arrb2^{-/-}$ (n = 9) mice were measured by carboxy-SNARF. (**D**) Co-localization of ADGRG2 (red fluorescence) and CFTR (sc-8909, Santa Cruz, green fluorescence)

*Figure 9 continued on next page*

*Figure 9 continued*

in the male efferent ductules of *Arrb2⁻/⁻* mice. (E) Analysis of ADGRG2 and CFTR fluorescence intensities in *Arrb2⁻/⁻* mice by Pearson's correlation analysis. The Pearson's correlation coefficient was 0.62. (F) Localization of ADGRG2 (red fluorescence) and CFTR (sc-8909, Santa Cruz, green fluorescence) in the male efferent ductules of *Arrb1⁻/⁻* mice. (G) Analysis of ADGRG2 and CFTR fluorescence intensities in *Arrb1⁻/⁻* mice by Pearson's correlation analysis. The Pearson's correlation coefficient was −0.15. (H) Co-localization of ezrin (red fluorescence) and CFTR (sc-8909, Santa Cruz, green fluorescence) in the male efferent ductules of *Arrb2⁻/⁻* mice. (I) Analysis of ezrin and CFTR fluorescence intensities in *Arrb2⁻/⁻* mice by Pearson's correlation analysis. The Pearson's correlation coefficient was 0.66. (J) Co-localization of ezrin (red fluorescence) and CFTR (sc-8909, Santa Cruz, green fluorescence) in the male efferent ductules of *Arrb1⁻/⁻* mice. (K) Analysis of ezrin and CFTR fluorescence intensities in *Arrb1⁻/⁻* mice by Pearson's correlation analysis. The Pearson's correlation coefficient was −0.15. (L) ADGRG2 was immunoprecipitated by an anti-ADGRG2 antibody in the male efferent ductules of *Arrb1⁻/⁻* mice or *Arrb2⁻/⁻* mice, and co-precipitates with CFTR, β-arrestin-1, and β-arrestin-2 were examined by using specific corresponding antibodies (CFTR antibody:20738–1-AP, Proteintech). (9A-C) *$p<0.05$, **$p<0.01$, ***$p<0.001$, *Adgrg2⁻/Y* mice compared with WT mice. #$p<0.05$, ##$p<0.01$, ###$p<0.001$, *Arrb1⁻/⁻* mice or *Arrb2⁻/⁻* mice compared with WT mice. ns, no significant difference. At least three independent biological replicates were performed for *Figure 9A–C and L*.

DOI: https://doi.org/10.7554/eLife.33432.026

The following figure supplements are available for figure 9:

**Figure supplement 1.** Western blot analysis of β-arrestin1/2 expression in the efferent duct tissue.

DOI: https://doi.org/10.7554/eLife.33432.027

**Figure supplement 2.** β-arrestin-1 is an essential component in a signaling complex encompassing the ADGRG2 and CFTR in efferent ductules.

DOI: https://doi.org/10.7554/eLife.33432.028

**Figure supplement 3.** The complex formation between ADGRG2, β-arrestin-1 and CFTR in HEK293 cells.

DOI: https://doi.org/10.7554/eLife.33432.029

---

ten volumes of homogenization buffer (75 mM Tris-Cl, pH 7.4; 2 mM EDTA, and 1 mM DTT supplemented with protease inhibitor cocktail). The dounced suspension was centrifuged at 1000 rpm for 15 min to discard the unbroken tissues. The collected suspensions were then centrifuged at 17,000 rpm for 1 hr to prepare the plasma membrane fraction. For the western blot or immunoprecipitation assays, the membranes were re-suspended in lysis buffer (50 mM Tris pH 8.0; 150 mM NaCl; 10% glycerol; 0.5% NP-40; 0.5 mM EDTA; and 0.01% DDM supplemented with protease inhibitor cocktail (Roche, Basel Switzerland) for 30 min.

## Isolation and ligation of efferent ductules

The efferent ductules were microdissected into 1–1.5 mm lengths and incubated for 24 hr in M199 culture medium containing nonessential amino acids (0.1 mM), sodium pyruvate (1 mM), glutamine (4 mM), 5α-dihydrotestosterone (1 nM), 10% fetal bovine serum, penicillin (100 IU/ml), and streptomycin (100 μg/ml) at 34°C in 95% humidified air and 5% $CO_2$. The segments were then ligated on two ends to exclude the entry and exit of fluids. Digital images of the ductules were analyzed at 0, 3, 12, 24, 36, 48, 60 and 72 hr after ligation. Damaged ductal segments were discarded. A rapid ciliary beat and clear lumens were used as evaluation standards for ductile segments that had undergone ligation. Between 9 and 36 total ductal segments from at least three mice were analyzed for each group. The differences between the means were calculated by one-way or two-way ANOVA.

## Recombinant adenovirus construction (*Wang et al., 2009*)

The recombinant adenovirus carrying the RFP or ADGRG2 gene with the ADGRG2 promoter (pm-ADGRG2) from the epididymal genome was produced in our laboratory using the AdEasy system for the rapid generation of recombinant adenoviruses according to the established protocol (*Luo et al., 2007*). An adenovirus carrying green fluorescent protein (GFP) was used as a control. For the in vivo studies, a single exposure to $5 \times 10^8$ plaque-forming units (pfu) of pm-RFP or pm-ADGRG2 adenovirus was delivered to isolated efferent ductules and incubated for 24 hr to allow for sufficient infection. Epididymal efferent ductules or epididymal efferent ductule epithelium were prepared for further experiments.

## Measurement of intracellular pH (pHi) with carboxy-SNARF−1

Digital images of the ductules were analyzed at 36 hr after ligation. Intracellular pH is examined with SNARF-1, a pH-sensitive fluorophore with a pKa of about 7.5. To load SNARF-1, cultured ductules were incubated with 5 μM SNARF-1-AM (diluted from a 1 mM stock solution in DMSO) for 45 min in

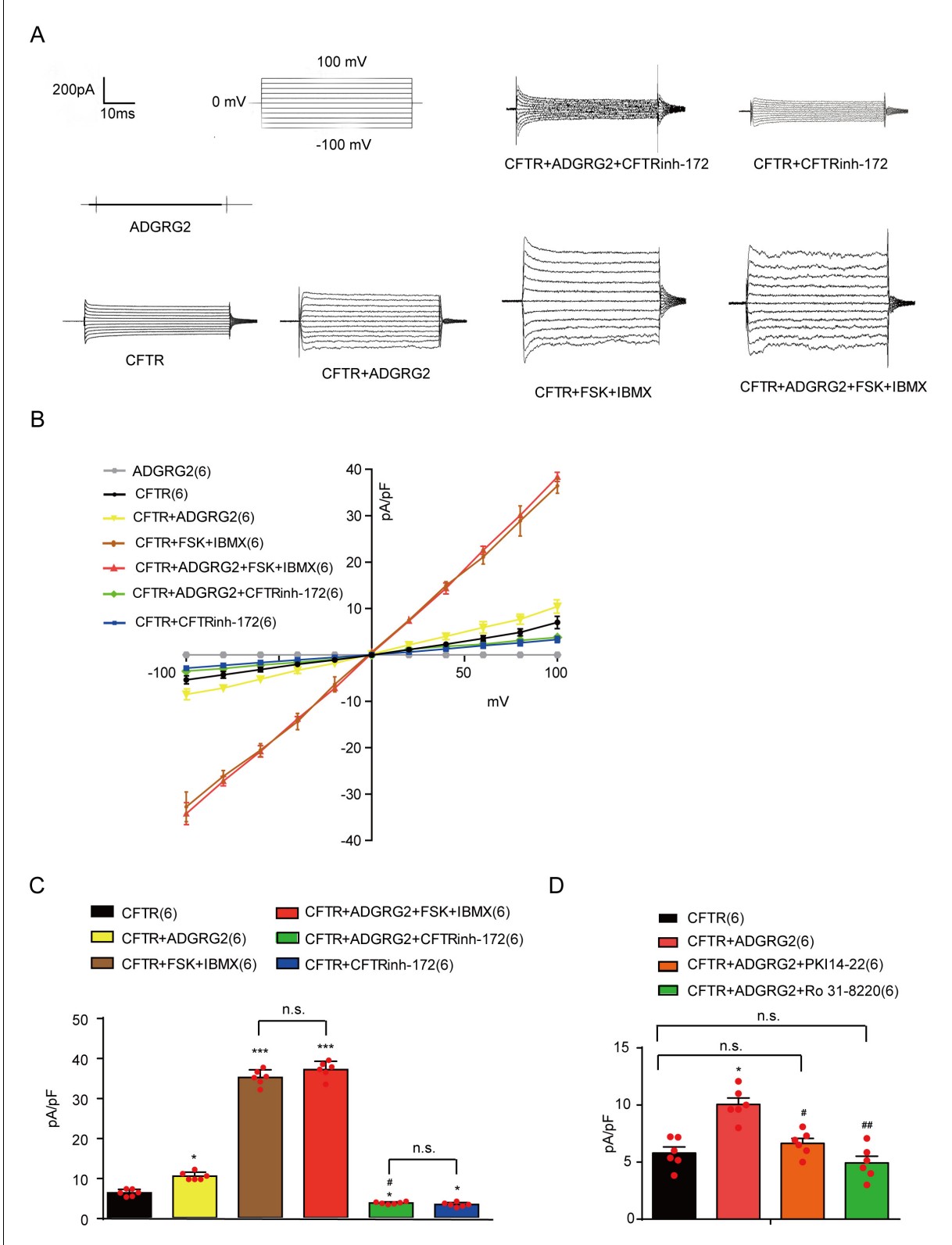

**Figure 10.** ADGRG2 upregulates CFTR Cl⁻ currents through G protein signaling. (**A**) Whole-cell Cl⁻ currents recorded with a CsCl pipette solution in HEK293 cells transfected with plasmids encoding ADGRG2 or/and CFTR, with or without CFTR inhibitor CFTRinh-172(10 μM) or its activator (FSK (10 μM)+IBMX (100 μM)). (**B**) Corresponding I-V curves of the whole-cell Cl⁻ currents recorded in (**C**). ADGRG2 (n = 6), CFTR (n = 6), CFTR + CFTRinh-172 (n = 6), CFTR + FSK + IBMX (n = 6), CFTR + ADGRG2 (n = 6), CFTR + ADGRG2+CFTRinh-172(n = 6), CFTR + ADGRG2+FSK + IBMX (n = 6). (**C and D**)

*Figure 10 continued on next page*

*Figure 10 continued*

Bar graph representation of average current densities (pA/pF) measured at 100 mV according to (**B**) and *Figure 9*; (**C and D**) Bar graph representation of average current densities (pA/pF) measured at 100 mV according to (**B**) and *Figure 10–figure supplement 5C*. (10C-10D) *p<0.05, **p<0.01, ***p<0.001, HEK293 cells transfected with CFTR compared with cells transfected with pCDNA3.1. #p<0.05, ##p<0.01, ###p<0.001, HEK293 cells transfected with ADGRG2 compared with non-ADGRG2 transfected cells. $p<0.05, $$, p<0.01, $$$, p<0.001, CFTRinh-172, FSK, NF449, U73122 or Ro 31–8220 compared with control vehicle. n.s., no significant difference. At least three independent biological replicates were performed for *Figure 10C–D*.

DOI: https://doi.org/10.7554/eLife.33432.030

The following figure supplements are available for figure 10:

**Figure supplement 1.** Co-localization analysis of ADGRG2 and CFTR in HEK293 cells.

DOI: https://doi.org/10.7554/eLife.33432.031

**Figure supplement 2.** Construction and expression of ADGRG2-full length (ADGRG2FL) and a truncated form ADGRG2β.

DOI: https://doi.org/10.7554/eLife.33432.032

**Figure supplement 3.** Overexpression of ADGRG2FL and ADGRG2β lead to constitutive increased cellular cAMP levels.

DOI: https://doi.org/10.7554/eLife.33432.033

**Figure supplement 4.** Overexpression of ADGRG2FL and ADGRG2β have constitutive Gq-NFAT signaling activities.

DOI: https://doi.org/10.7554/eLife.33432.034

**Figure supplement 5.** ADGRG2 upregulates CFTR Cl⁻ currents and Cl⁻ efflux through G protein signaling.

DOI: https://doi.org/10.7554/eLife.33432.035

culture medium at 37°C, 5% $CO_2$. The cells are washed twice with buffer containing 110 mM NaCl, 5 mM KCl, 1.25 mM $CaCl_2$, 1.0 mM $MgSO_4$, 0.5 mM $Na_2HPO_4$, 0.5 mM $KH_2PO_4$, and 20 mM HEPES, pH 7.4, then placed on the microscope stage in buffer containing 5 mM KCl, 110 mM NaCl, 1.2 mM $NaH_2PO_4$, 25 mM $NaHCO_3$, 30 mM glucose, 10 U/ml penicillin, 10 μg/ml streptomycin, and 25 mM HEPES, pH 7.30. The fluorescence was examined using an LSM 780 laser confocal fluorescence microscope (Carl Zeiss) with the excitation wavelength at 488 nm. The emissions of SNARF-1 at 590 and 635 nm were captured in the first two consecutive scans.

### Intracellular pH calibration (*Seksek et al., 1991*)

In vivo pH calibration was performed according to the method developed by Seksek et al. Briefly, after incubation with the fluorescent probe, cells were washed in a buffer containing 10 mM Hepes, 130 mM KCl, 20 mM NaCl, 1 mM $CaCl_2$, 1 mM $KH_2PO_4$, 0.5 mM $MgSO_4$, at various pH values obtained by addition of small amounts of 0.1 M solutions of KOH or HCl. The pH changes of the external buffer of the cell suspension were followed with a Tacussel Isis 20000 pH-meter. Addition of nigericin (1 pg/ml) and valinomycin (5 pM) allowed an exchange of $K^+$ for $H^+$ which resulted in a rapid equilibration of external and internal pH. The fluorescence of the probe was excited at 488 nm, then the emission of SNARF-1 at 590 and 635 nm were captured in the first two consecutive scans. The fluorescent ratio values obtained for each pH point were used for the calibration curve obtained with Prism software, from which pHi values of the samples (6.0–8.5) were determined. Determinations were performed in quintuplicate. The sensor does not have significant effects on cell viability.

The effect of bicarbonate on intracellular pH was determined by incubating ductules in culture medium containing 25 mM bicarbonate for 40 min at 37°C, and then transferring these ductules into bicarbonate-free salt solution and then the fluorescence of the SNARF-1 probe was examined (*Teti et al., 1989*). Bicarbonate-free solutions were prepared by substituting $NaHCO_3$ with Na- gluconate and equilibrating with air.

1 mM amiloride or 500 μM acetazolamide were added 100 s after the beginning of the measurement to examine the effects of acetazolamide and amiloride.

### Quantitative real-time PCR

Total RNA from the mouse efferent ductules was extracted using a standard TRIzol RNA isolation method (Invitrogen, Carlsbad, CA) as previously described (*Wang et al., 2014*). The reverse transcription and PCR experiments were performed with the Revertra Ace qPCR RT Kit (TOYOBO FSQ-101) using 0.5 μg of each sample, according to the manufacturer's protocols. The quantitative real-time PCR was conducted in the LightCycler apparatus (Bio-Rad) using the FastStart Universal SYBR

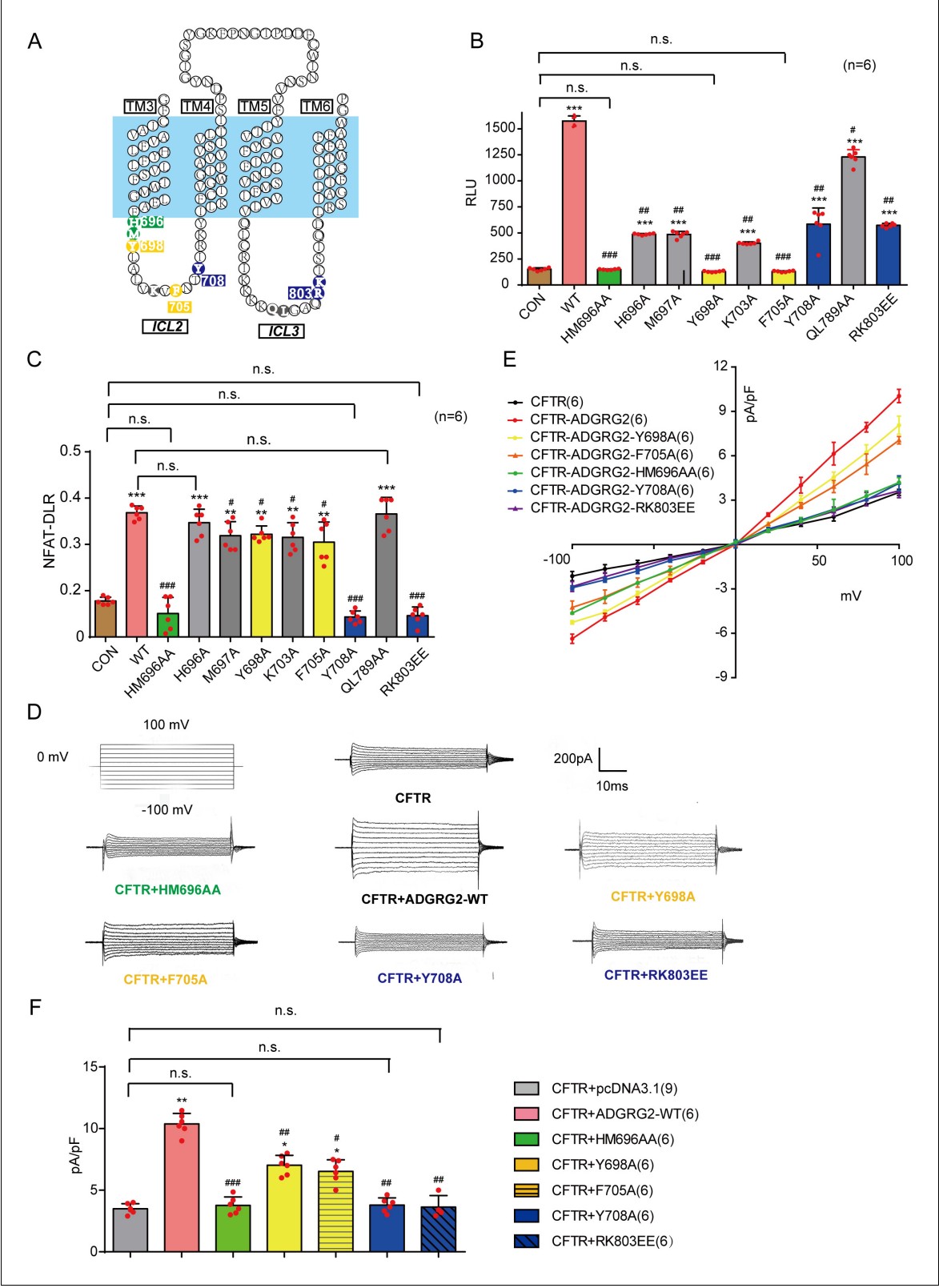

**Figure 11.** Key mutations of ADGRG2 downregulates CFTR Cl⁻ currents through G protein signaling. (A) Schematic representation of the location of the selected ADGRG2 mutants in intracellular loop 2 and loop 3 of ADGRG2. (B) Effects of the overexpression of ADGRG2 (n = 6) and its mutations (n = 6) on cAMP levels. (C) Effects of the overexpression of ADGRG2 (n = 6) and its mutations (n = 6) on NFAT-DLR activation. (D) Whole-cell Cl⁻ currents recorded with a CsCl pipette solution in HEK293 cells overexpressing CFTR, CFTR and ADGRG2-WT, CFTR and ADGRG2-HM696AA, CFTR

*Figure 11 continued on next page*

*Figure 11 continued*

and ADGRG2-Y698A, CFTR and ADGRG2-F705A, CFTR and ADGRG2-Y708A or CFTR and ADGRG2-RK803EE. (E) Corresponding I-V curves for the whole-cell Cl⁻ currents recorded in (D). (F) Bar graph representation of average current densities (pA/pF) measured at 100 mV according to (E). (11B-11C and 11F) *$p<0.05$, **$p<0.01$, ***$p<0.001$, cells transfected with ADGRG2-WT or mutants compared with the control plasmid (pCDNA3.1). #$p<0.05$, ##$p<0.01$, ###$p<0.001$, cells overexpressing ADGRG2 mutants compared with ADGRG2-WT. n.s., no significant difference. At least three independent biological replicates were performed for *Figure 11B–C,F*.

DOI: https://doi.org/10.7554/eLife.33432.036

The following figure supplements are available for figure 11:

**Figure supplement 1.** Sequence alignment of the transmembrane domains of ADGRG2 (*Homo sapiens, Mus musculus, Rattus norvegicus*), β2AR (*H. sapiens, M. musculus,* and *R. norvegicus*), and GPR126 (*H. sapiens*).
DOI: https://doi.org/10.7554/eLife.33432.037

**Figure supplement 2.** Western blot and ELISA analysis of the expression of these mutants in the cell membrane.
DOI: https://doi.org/10.7554/eLife.33432.038

**Figure supplement 3.** Corresponding bar graph of average reversal potential($E_{rev}$) (±s.e.m., n = 6 for each condition) recorded in *Figure 11D–11E* and calculated Nernst potential.
DOI: https://doi.org/10.7554/eLife.33432.039

Green Master (Roche). The qPCR protocol was as follows: 95°C for 10 min; 40 cycles of 95°C for 15 s and 60°C for 1 min; and then increasing temperatures from 65°C to 95°C at 0.1 °C/s. The mRNA level was normalized to GAPDH in the same sample and then compared with the control. All primers are listed in *Supplementary file 1* and *Supplementary file 2*.

## Immunofluorescence staining

The mice were decapitated, and the epididymis and efferent ductules were removed immediately. After dissection, the epididymis and efferent ductules were fixed in 4% paraformaldehyde by immersion overnight at 4°C. The fixed tissues were then rinsed for 4 hr at 4°C in PBS containing 10% sucrose, for 8 hr in 20% sucrose, and then overnight in 30% sucrose. The tissues were embedded in Tissue-Tek OCT compound (Sakura Fintek USA, Inc., Torrance, CA) and then mounted and frozen at −25°C. Subsequently, 8-μm-thick coronal serial sections were cut at the level of the efferent ductules and mounted on poly-D-lysine-coated slides. The slides were incubated in citrate buffer solution for antigen retrieval. Non-specific binding sites were blocked with 2.5% (wt/vol) BSA, 1% (vol/vol) donkey serum and 0.1% (vol/vol) Triton X-100 in PBS for 1 hr. After blocking, the slides were incubated in primary antibody against ADGRG2 (1:300), CFTR (1:50), Gs (1:20), Gq (1:20), ANO1(1:50), Anti-ezrin(1:50) or Anti-Acetylated Tubulin(Lys40)(1:50) at 4°C overnight. Subsequently, the slides were incubated for 1.5 hr with the secondary antibody (1:500, Invitrogen) at room temperature. For nuclear staining, the slides were incubated with DAPI (1:2000, Beyotime) for 15 min at room temperature. The immunofluorescence results were examined using a LSM 780 laser confocal fluorescence microscope (Carl Zeiss). The normal saline group was treated as the control.

## Culture of mouse epididymal efferent duct epithelium (*Leung et al., 2001*)

After opening the lower abdomen, the efferent ductules were isolated under sterile conditions to remove fat or connective tissue. The ductules were severed into small segments and then transferred to Hanks balanced salt solution (HBSS) containing 0.2% (w/v) collagenase I and 0.1% (w/v) trypsin. Subsequently, the ductules were incubated at 34°C for 1 hr with vigorous shaking (150 strokes/min) and then separated by centrifugation at 800 g for 5 min. The pellets were re-suspended in HBSS containing collagenase I 0.2% (w/v) for 30 min at 34°C with vigorous shaking. The solutions were then centrifuged again at 800 g for 5 min, and the cell pellets were re-suspended in HBSS buffer containing 0.2% (w/v) collagenase I and then subjected to repeated pipetting for 15 min. Finally, the cells were centrifuged at 800 × g again for 5 min and resuspended in M199 medium. The cell suspension was incubated at 34°C for 5–6 hr in 5% $CO_2$. The resulting fibroblasts and smooth muscle cells were attached to the bottom of the culture flask, whereas the epithelial cells were in suspension. The suspensions were collected, and the epithelial cells were seeded on culture flasks.

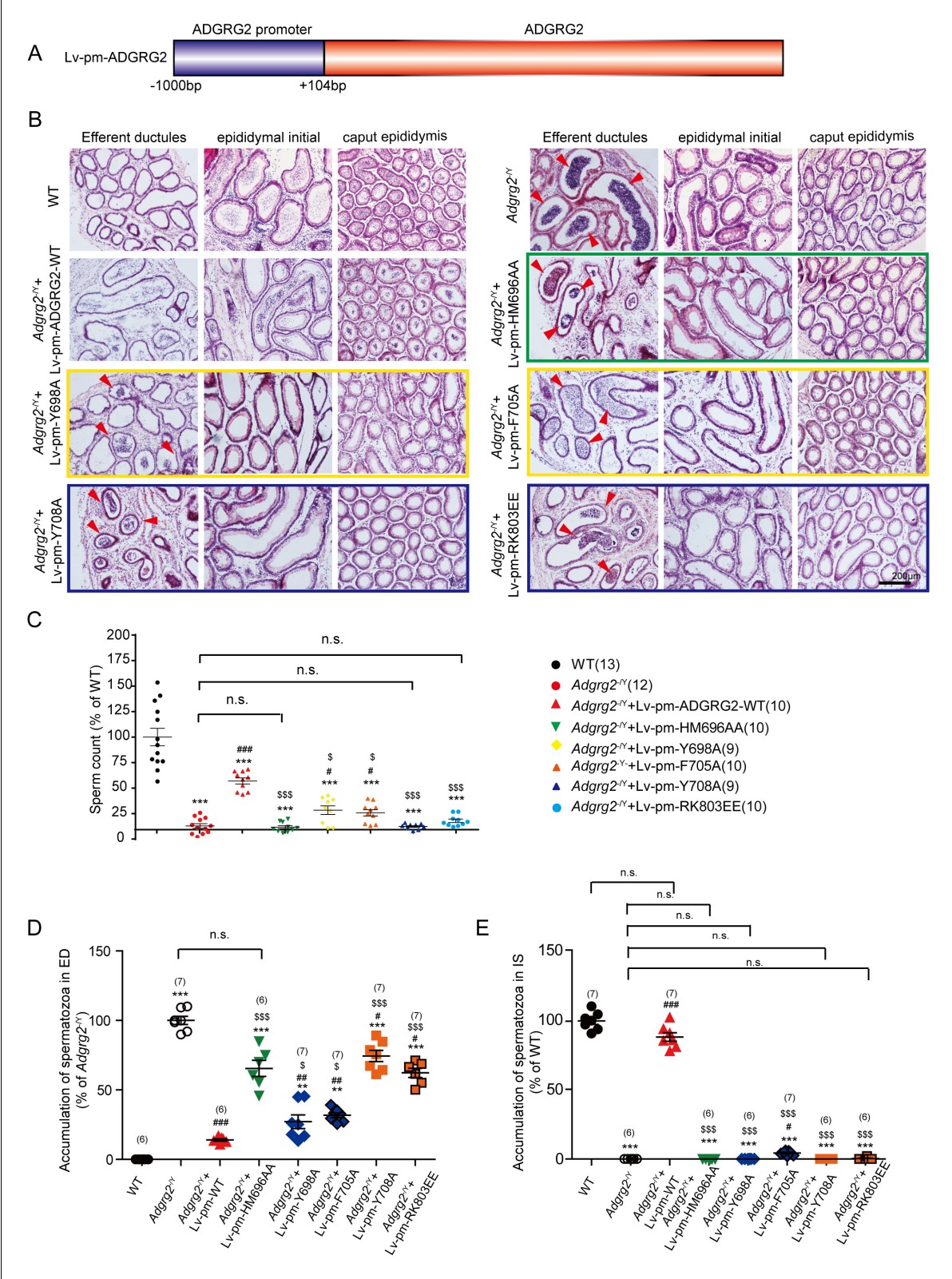

**Figure 12.** Conditional expression of ADGRG2 wild-type or its selective G-subtype signaling mutants in the efferent ductules in *Adgrg2⁻/ᵧ* mice and their effects on the morphology, sperm maturation of efferent ductules. (**A**) Schematic representation of the mouse ADGRG2 promoters used in the rescue experiment. (**B**) Representative hematoxylin-eosin staining of the WT mice, *Adgrg2⁻/ᵧ* mice or *Adgrg2⁻/ᵧ* mice infected with lentivirus encoding ADGRG2-WT or different G-subtype mutants at the efferent ductules, initial segment or caput of the epididymis. Scale bars, 200 μm. (**C**) Bar graph

*Figure 12 continued on next page*

*Figure 12 continued*

representing the quantitative analysis of the number of sperm shown in *Figure 12B* from at least four independent experiments. (D–E) The corresponding bar graph of the accumulation of spermatozoa according to the hematoxyline-eosin staining of the WT, *Adgrg2*$^{-/Y}$ mice or *Adgrg2*$^{-/Y}$ mice infected with lentivirus encoding GRP64-WT or different G subtype mutants at the efferent ductules (D) or initial segment (E) of epididymis. (C–E) *p<0.05, **p<0.01, ***p<0.001; *Adgrg2*$^{-/Y}$ mice compared with WT mice. #p<0.05, ##p<0.01, ###p<0.001; *Adgrg2*$^{-/Y}$ mice infected with the lentivirus encoding different ADGRG2 constructs compared with *Adgrg2*$^{-/Y}$ mice infected with the control lentivirus. $, p<0.05, $$$, p<0.001; *Adgrg2*$^{-/Y}$ mice infected with the lentivirus encoding different ADGRG2 constructs compared with *Adgrg2*$^{-/Y}$ mice infected with the ADGRG2-WT lentivirus. n.s., no significant difference. At least three independent biological replicates were performed for *Figure 12C–E*.

DOI: https://doi.org/10.7554/eLife.33432.040

The following figure supplement is available for figure 12:

**Figure supplement 1.** Effect of the conditional expression of ADGRG2-WT or its selective G-subtype signaling mutants on the rescue of reproductive defects in *Adgrg2*$^{-/Y}$ mice.

DOI: https://doi.org/10.7554/eLife.33432.041

## Constructs

The wild-type ADGRG2 full-length (ADGRG2FL) plasmid was obtained from Professor Xu Z. G. at Shandong University School of Life Sciences, Jinan, Shandong, China. ADGRG2 was cloned from mouse total cDNA libraries using the following primers: forward, ATTCTCGAGGATGCTTTTCTC TGGTGGG; and reverse, ATTGAATTCCATTTGCTCGATAAAGTG. The sequences were inserted into the mammalian pEGFP-N2 expression vector, and then ADGRG2FL and ADGRG2 C-terminal trunca-tions (ADGRG2β) were subcloned into the pcDNA3.1 expression vector, with the flag sequence added at the N-terminus. The ADGRG2FL mutants (HM696AA, H696A, M697A, Y698A, K703A, V704A, F705A, Y708A, QL798AA, RK803EE) were generated using a QuikChange Mutagenesis Kit (Stratagene). All of the mutations were verified by DNA sequencing. All primers are listed in *Supplementary file 3*.

## Cell culture, transfection, and western blotting

HEK293 cells were obtained from Cell Resource Center of Shanghai Institute for Biological Sciences (Chinese Academy of Sciences, Shanghai, China). The cell line was validated by STR profiling (Shang-hai Biowing Applied Biotechnology (SBWAB) Co. Ltd.) and was negative for mycoplasma as mea-sured by MycoAlert Mycoplasma Detection Kit (Lonza). HEK293 cells were maintained in Dulbecco's modified Eagle's medium (DMEM) supplemented with 10% heat-inactivated fetal bovine serum (Hyclone Thermo Scientific, Scoresby, Victoria, Australia), penicillin (100 IU/ml), and streptomycin (100 µg/ml) as previously described (*Hu et al., 2014*; *Wang et al., 2014*). For receptor or other pro-tein expression, plasmids carrying the desired genes were transfected into cells using Lipofectamine TM 2000 (Invitrogen). To monitor the protein expression levels, cells were collected 48–72 hr post-transfection with lysis buffer (50 mM Tris, pH 8.0; 150 mM NaCl; 1 mM NaF; 1% NP-40; 2 mM EDTA; Tris-HCl, pH 8.0; 10% glycerol; 0.25% sodium deoxycholate; 1 mM Na$_3$VO$_4$; 0.3 µM aprotinin; 130 µM bestatin; 1 µM leupeptin; 1 µM repstatin; and 0.5% IAA). The cell lysates were subjected to end-to-end rotation for 20 min and spun at 12,000 rpm for 20 min at 4°C. Then, an equal volume of 2 × loading buffer was added. Proteins were denatured in the loading buffer and subjected to west-ern blot analysis. The protein bands from the western blot were quantified using ImageJ software (National Institutes of Health, Bethesda MD). Each experiment was repeated at least in triplicate. A data analysis was conducted using GraphPad software.

## Co-immunoprecipitation

The efferent ductules of WT or *Adgrg2*$^{-/Y}$ mice were dissected into small pieces. The interaction between proteins is stabilized by addition of 1 ml of cross-linker buffer (D-PBS containing 10 mM HEPES and 2.5 mM DSP in 1:1 (v/v) dimethyl sulfoxide (DMSO)) as previously described(*Ning et al., 2015*; *Yang et al., 2015*). After continuous slow agitation for 30 min at room temperature, crosslink-ing was stopped by adding 25 mM Tris-HCl (pH 7.5) and incubated for another 15 min. The tissue were washed with cold PBS and then lysed in cold lysis buffer with protease inhibitors. After centrifu-gation, the supernatants were incubated with anti-ADGRG2 antibody (AF7977, R and D systems) for at least 2 hr at 4°C. Next, Protein A/G PLUS-Agarose (sc-2003, Santa Cruz) was added, and the com-plexes were incubated overnight at 4°C. The beads were washed with PBS buffer several times, and

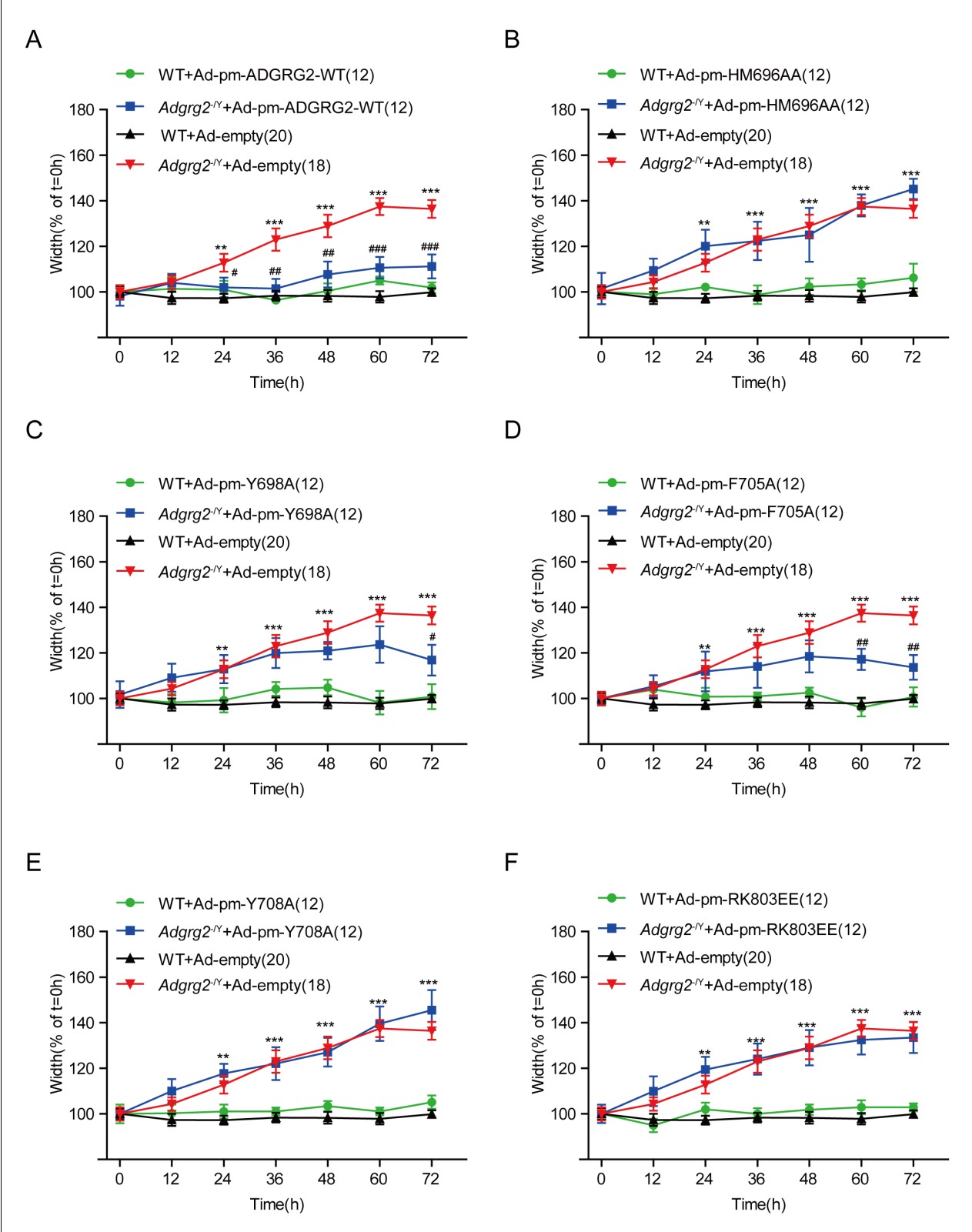

**Figure 13.** Effects of conditional expression of ADGRG2 wild-type or its selective G-subtype signaling mutants in *Adgrg2*[−/Y] mice on the fluid reabsorption of efferent ductules. (**A**) Effects of the expression of the ADGRG2-WT adenovirus on the diameter of the ligated efferent ductules derived from the WT or *Adgrg2*[−/Y] mice. (**B–F**) Effects of the expression of adenovirus encoding different ADGRG2 mutants on the diameter of the ligated efferent ductules derived from the WT (n = 12) or *Adgrg2*[−/Y] (n = 12) mice. (**A–F**) *p<0.05, **p<0.01, ***p<0.001; *Adgrg2*[−/Y] mice infected with the empty

*Figure 13 continued on next page*

*Figure 13 continued*
adenovirus compared with WT mice infected with the empty adenovirus. #p<0.05, ##p<0.01, ###p<0.001; *Adgrg2*⁻/Y mice infected with the adenovirus encoding different ADGRG2 constructs compared with *Adgrg2*⁻/Y mice infected with the control adenovirus. n.s., no significant difference.
DOI: https://doi.org/10.7554/eLife.33432.042

proteins were denatured in the SDS-PAGE loading buffer and subjected to western blot analysis with the indicated antibodies.

### Whole-cell patch-clamp recording (*Guo et al., 2014*)

The efferent ductules infected by adenovirus with the ADGRG2 promoter were isolated, and epithelial cells were purified and cultured on coverslips before the patch-clamp recording. ADGRG2-promoter labeling was achieved by observation of the RFP fluorescence with the microscope. HEK293 cells transfected with plasmids encoding CFTR together with or without the ADGRG2 wild type or its mutants were cultured on coverslips before the patch-clamp recording. Borosilicate glass-made patch pipettes (Vitrex, Modulohm A/S, Herlev, Denmark) were pulled with a micropipette puller (P-97, Sutter Instrument Co.) to a resistance of 5–7 MΩ after they were filled with pipette solution. The ionic current was recorded with a data acquisition system (DigiData 1322A, Axon Instruments) and an amplifier (Axopatch-200B, Axon Instruments, Foster City, CA). The command voltages were controlled by a computer equipped with pClamp Version nine software. For the whole cell Cl⁻ current measurement, cells were bathed in a solution of NaCl at 130 mM, KCl at 5 mM, $MgCl_2$ at 1 mM, $CaCl_2$ at 2.5 mM, and HEPES 20 mM, and D-mannitol was added to an osmolarity of 310 (pH 7.4). Pipettes were filled with a solution of 101 mM CsCl, 10 mM EGTA, 10 mM Hepes, 20 mM TEACl, 2 mM MgATP, 2 mM $MgCl_2$, 5.8 mM glucose, pH7.2, with D-mannitol compensated for osm 290. When the whole-cell giga-seal was formed, the capacitance of the cell was measured. The whole-cell current was obtained by a voltage clamp with the commanding voltage elevated from −100 mV to +100 mV in 20 mV increments (*Yu et al., 2011*). Further validation of these observed currents were Cl⁻ selective was provided by experiments in which 100 mM of the extracellular Cl⁻ was replaced by gluconate.

### cAMP ELISA

The efferent ductules were carefully microdissected under sterile conditions to remove fat or connective tissue and then were ligated on two ends to exclude the entry and exit of fluids. After 24 hr, these tissues were rinsed with PBS and homogenized with a tissue homogenizer in cold 0.1 N HCl containing 500 μM IBMX at a 1:5 ratio (w/v). The supernatants were collected after the centrifugation of the tissue lysates at 10,000 × g and then neutralized with 1 N NaOH. The supernatant was collected for the cAMP determination by ELISA according to the manufacturer's instructions.

### IP1 ELISA

The efferent ductules were ligated on two ends for 24 hr, and then were added 5 mM ATP or control vehicles to the tissues for 30 min. After half an hour, the tissues were homogenized with a tissue homogenizer in an assay buffer (10 mM HEPES, 1 mM $CaCl_2$, 0.5 mM $MgCl_2$, 4.2 mM KCl, 146 mM NaCl, 5.5 mM glucose, 50 mM LiCl, pH 7.4). The 50 mM LiCl was added to block the IP1 degradation. The lysates were centrifuged at 10,000 × g to remove insoluble components, and the supernatant was then collected for IP1 determination by ELISA (lp034186) according to the manufacturer's instructions.

### GloSensor cAMP assay

The GloSensor cAMP assay was performed as previously described (*Binkowski et al., 2009*; *Fan et al., 2008*; *Hu et al., 2014*; *Kimple et al., 2009*). HEK293 cells were transfected with the GloSensor plasmid and the desired expression plasmids (0.8 μg of total DNA) with Lipofectamine 2000 in 24-well dishes. Twenty-four hours after transfection, the cells were plated on 96-well plates at a cell density of 20,000 cells/well. The cells were maintained in DMEM for another 24 hr, washed with PBS and then incubated with 100 μl of solution containing 10% FBS, 2% (v/v) GloSensor cAMP

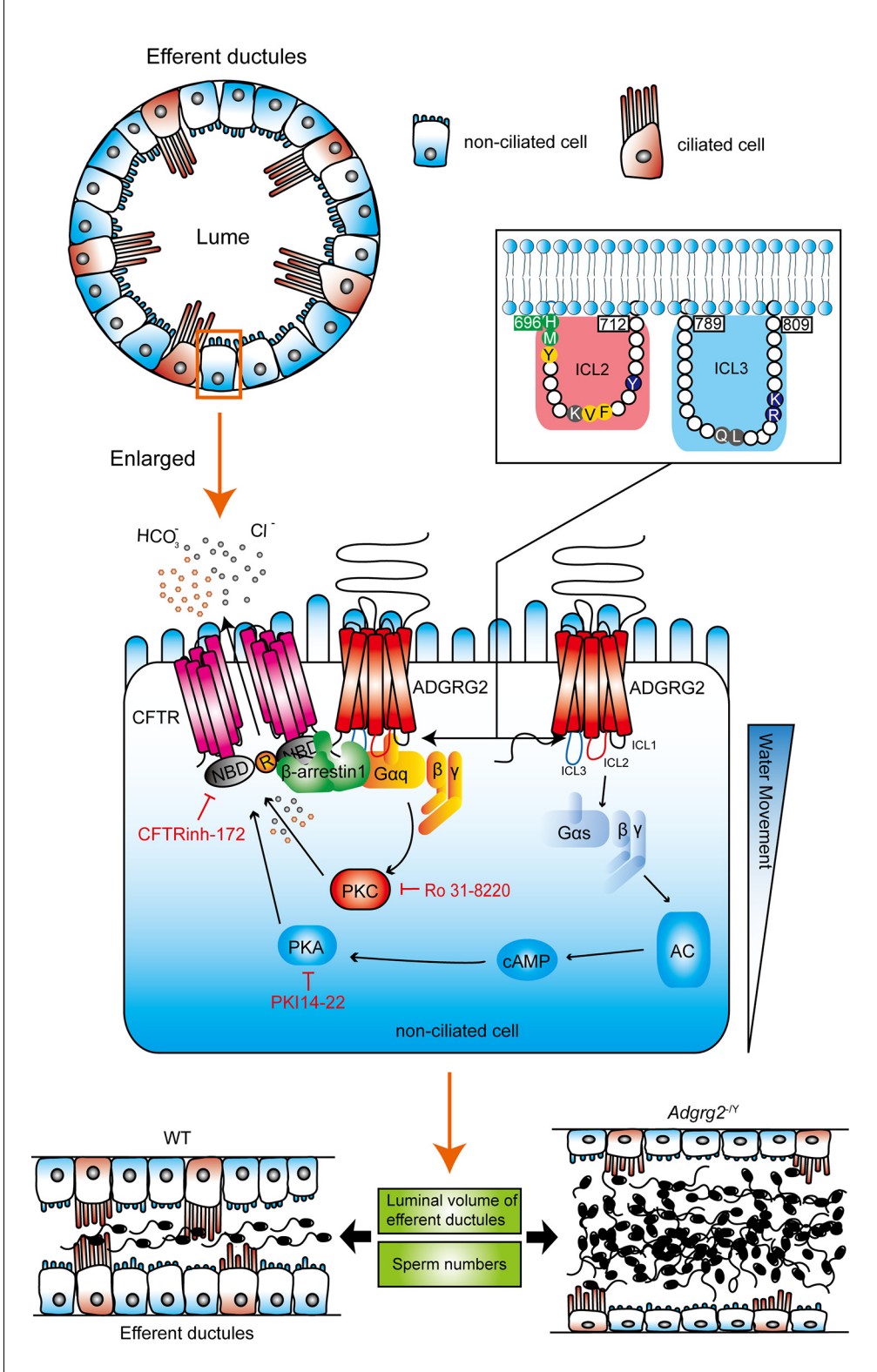

**Figure 14.** Schematic diagram depicting the GPCR signaling pathway in the regulation fluid reabsorption in the efferent ductules. The ADGRG2 and CFTR localized at cell plasma membrane, whereas Gs and Gq localize at the inner surface of non-ciliated cells. Deficiency of ADGRG2 in *Adgrg2⁻/Y* mice, reducing the Gq protein level by half in *Gnaq⁺/⁻* mice or PKC inhibitor Ro 31–8220 significantly destroyed the coupling of ADGRG2 to CFTR, thus impaired Cl⁻ and H⁺ homeostasis and fluid reabsorption of efferent ductules. Structurally, residues in intracellular loops 2 and 3 of ADGRG2 are required for the specific interactions between ADGRG2 and Gq, which are required for CFTR and ADGRG2 coupling and fluid reabsorption. In addition

*Figure 14 continued*

to G protein signaling, β-arrestin-1 is also required for fluid reabsorption in efferent ductules by scaffolding the ADGRG2 and CFTR coupling and complex formation. Therefore, a signaling complex including ADGRG2, Gq, β-arrestin-1 and CFTR that specifically localizes in non-ciliated cells is responsible for the regulation of Cl⁻ and H⁺ homeostasis and fluid reabsorption in the efferent ductules; thus, these functions are important for male fertility. Moreover, activation of the AGTR2 could rescue the H⁺ metabolic disorder caused by ADGRG2 deficiency, which restored the ability of fluid reabsorption in efferent ductules, providing a potential therapeutic strategy in treatment of male infertility caused by dysfunction of GPCR-CFTR signaling in non-ciliated cells.

DOI: https://doi.org/10.7554/eLife.33432.043

reagent and 88% $CO_2$-independent medium in each well for 2 hr. The cAMP signal was examined using a luminescence counter (Mithras LB 940).

## NFAT dual-luciferase reporter(DLR) assay (*Hu et al., 2014*)

HEK293 cells in 24-well dishes were co-transfected with plasmids encoding ADGRG2 or its mutants, pGL4.16-NFAT luciferase or pGL4.16-basic luciferase, and pRL-TK Renilla using Lipofectamine 2000. These cells were cultured for approximately 48 hr and then harvested by the addition of $1 \times$ passive lysis buffer. After incubation for 15 min at room temperature with shaking, the cell lysates were centrifuged for 10 min at 12,000 rpm at 4°C. NFAT-DLR activity was quantified by a standard luciferase reporter gene assay and then normalized to Renilla luciferase activity (Promega) as previously described (*Wang et al., 2014*). At least three independent experiments were executed for each dual-luciferase reporter (DLR) assay.

## Recombinant lentivirus construction and lentivirus injection

Recombinant lentiviruses containing the ADGRG2 gene and its mutants (HM696AA, H696A, M697A, Y698A, K703A, V704A, F705A, Y708A, QL798AA, RK803EE) under the ADGRG2 promoter were produced according to standard procedures (*Tiscornia et al., 2006*; *Ye et al., 2008*). The lentivirus titer was $1 \times 10^9$ TU/ml. Mice were anesthetized with 10% chloral hydrate and then the conditional expression of ADGRG2-WT or its selective G-subtype signaling mutants' lentivirus were microinjected into the interstitial space of the efferent ductules and the initial segment of epididymis at a multiplicity of infection of 100. After 14–21 days, the epididymis transfected with lentivirus were collected for use in further experiments.

## Histology (*Mendive et al., 2006*)

The epididymis and efferent ductules were removed and fixed overnight at 4°C in 4% paraformaldehyde and stored in 70% ethanol until further use. The tissues were dehydrated, embedded in paraffin, and then sectioned into 10 μm slices. In most cases, the whole epididymis was sectioned, and representative samples throughout the organ were mounted on slides for hematoxylin and eosin staining. Hematoxylin and eosin staining was performed according to standard procedures.

## Analysis of spermatozoa (*Davies et al., 2004*)

Spermatozoa from the caudal epididymis of the wild-type (n = 13) or *Adgrg2⁻/Y* knockout (n = 12) mice (ages between 15 and 20 weeks) were collected. The caudal region from the epididymis was open and incubated for 10 min in PBS at 34°C to allow the spermatozoa to appear. The spermatozoa were counted and analyzed by spreading the diluted homogenous suspension over a microscope slide.

## Treatment of mice efferent ductules with CFTR siRNA dicer

CFTR siRNA was designed as described before (*Ruan et al., 2012*; *Wang et al., 2006*) and chemically modified by the manufacturer (GenePharma). Sequences corresponding to the siRNA of scrambled were: sense, 5'-CUUCCUCUCU UUCUCUCCCU UGUGA-3'; and antisense, 5'- TCACA AGGGAGAGAA AGAGAGGAAG-3' or CFTR-specific siRNA-CFTR, dicer-1: sense, 5'-GUGCAAA UUCAGAGCUUUGUGGAACAG-3'; and antisense, 5'- CUGUUCCACAAA GCUCUGAAUUUGCAC-3'; CFTR-specific siRNA-CFTR, dicer-2: sense,5'-GACAACUUGUUAGUCUUCUUUCCAA-3'; and antisense, 5'- UUGGAAAGAAGACUAACAAGUUGUC-3'; CFTR-specific siRNA-CFTR, dicer-3: sense, 5'-

GAGAUUGAU GGUGUCUCAUGGAAUU-3'; and antisense, 5'-AAUUCCAUGAGACACCAUCAAUC UC-3'; For in vivo studies, 15 µg of the siRNA dissolved in 30% pluronic gel (Pluronic F-127, Sigma) solution was delivered to the mice efferent ductules immediately as previously described (*Wang et al., 2009*). After 7 days, the epididymis transfected with siRNA were collected for further experiments.

## Statistics

All the western blots were performed independently for at least three times, and the representative experimental results were shown in the main or supplementary figure. All the data are presented as the mean ±SD from at least three independent experiments. Statistical comparisons were performed using an ANOVA with GraphPad Prism5. Significant differences were accepted at p<*0.05*. The sequence alignments were performed using T-coffee.

## Additional information

### Funding

| Funder | Grant reference number | Author |
| --- | --- | --- |
| National Natural Science Foundation of China | 31470789 | Jin-Peng Sun |
| National Natural Science Foundation of China | 31611540337 | Ka Young Chung Jin-Peng Sun |
| National Natural Science Foundation of China | 81773704 | Jin-Peng Sun |
| Shandong Natural Science Fund for Distinguished Young Scholars | JQ201517 | Jin-Peng Sun |
| Shandong Provincial Natural Science Foundation | ZR2014CP007 | Dao-Lai Zhang |
| National Natural Science Foundation of China | 31671197 | Xiao Yu |
| The Program for Changjiang Scholars and Innovative Research Team in University | IRT13028 | Xiao Yu |
| National Natural Science Foundation of China | 31471102 | Xiao Yu |
| National Science Fund for Distinguished Young Scholars | 81525005 | Fan Yi |

The funders had no role in study design, data collection and interpretation, or the decision to submit the work for publication.

### Author contributions

Dao-Lai Zhang, Data curation, Software, Formal analysis, Funding acquisition, Investigation, Visualization, Methodology, Writing—original draft; Yu-Jing Sun, Data curation, Software, Formal analysis, Investigation, Visualization, Methodology, Writing—original draft; Ming-Liang Ma, Yi-jing Wang, Software, Investigation, Visualization, Methodology; Hui Lin, Software, Investigation, Methodology; Rui-Rui Li, Zong-Lai Liang, Amy Lin, Investigation, Methodology; Yuan Gao, Hui Mo, Yu-Jing Lu, Meng-Jing Li, Investigation; Zhao Yang, Formal analysis; Dong-Fang He, Supervision; Wei Kong, Ka Young Chung, Fan Yi, Jian-Yuan Li, Ying-Ying Qin, Jingxin Li, Methodology; Alex R B Thomsen, Alem W Kahsai, Zhi-Gang Xu, Data curation; Zi-Jiang Chen, Mingyao Liu, Dali Li, Resources; Xiao Yu, Conceptualization, Data curation, Formal analysis, Supervision, Funding acquisition, Validation, Methodology, Project administration; Jin-Peng Sun, Conceptualization, Data curation, Formal analysis, Supervision, Funding acquisition, Validation, Methodology, Writing—original draft, Project administration, Writing—review and editing

Author ORCIDs
Dao-Lai Zhang http://orcid.org/0000-0003-4428-8731
Amy Lin http://orcid.org/0000-0001-6723-5443
Jin-Peng Sun http://orcid.org/0000-0003-3572-1580

## Ethics

Animal experimentation: Mice were individually housed in the Shandong university on a 12:12 light: dark cycle with access to food and water ad libitum.The use of mice was approved by the animal ethics committee of Shandong university medical school (protocol LL-201502036). All animal care and experiments were reviewed and approved by the Animal Use Committee of Shandong University School of Medicine.

## Decision letter and Author response

Decision letter https://doi.org/10.7554/eLife.33432.050
Author response https://doi.org/10.7554/eLife.33432.051

# Additional files

## Supplementary files

• Supplementary file 1. Primers for the Quantitative RT-PCR (qRT-PCR) analysis of mRNA transcription profiles of G protein subtypes and β-arrestins.
DOI: https://doi.org/10.7554/eLife.33432.044

• Supplementary file 2. Primers for the Quantitative RT-PCR (qRT-PCR) analysis of mRNA transcription profiles of potential osmotic drivers including selective ion channels and transporters.
DOI: https://doi.org/10.7554/eLife.33432.045

• Supplementary file 3. Primers for the construction of ADGRG2FL mutants (HM696AA, H696A, M697A, Y698A, K703A, V704A, F705A, Y708A, QL798AA, RK803EE).
DOI: https://doi.org/10.7554/eLife.33432.046

• Transparent reporting form
DOI: https://doi.org/10.7554/eLife.33432.047

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
