## [Decision Letter]

[Editors’ note: a previous version of this study was rejected after peer review, but the authors submitted for reconsideration. The first decision letter after peer review is shown below.]

Thank you for submitting your work entitled "Gq activity and β-arrestin-1 scaffolding are required for male fertility through mediating GPR64/CFTR coupling" for consideration by *eLife*. Your article has been reviewed by three peer reviewers, and the evaluation has been overseen by a Reviewing Editor and a Senior Editor. The following individuals involved in review of your submission have agreed to reveal their identity: Peter Haggie (Reviewer #3).

Our decision has been reached after consultation between the reviewers. Based on these discussions and the individual reviews below, we regret to inform you that your work will not be considered further for publication in *eLife*.

All three reviewers thought the connection between GPR64 and CFTR was highly interesting and that mouse phenotype was also relevant in the context of efferent ductule physiology and cystic fibrosis. However, all reviewers also pointed out significant technical concerns that affect key results. In particular, the use of a relatively unspecific CFTR inhibitor that also targets Slc26a9, and concerns with the electrophysiology experiments raised questions about the primary data and the interpretation of key results. Moreover, the probes and methods used for measuring pH and Cl^-^ levels also present technical problems. These significant technical issues and the need for extensive experimental work to provide proper controls throughout make a revision of this manuscript unfeasible within a reasonable time frame.

Reviewer #1:

In this paper, Zhang et al. link GPR64-dependent activation of CFTR with male reproductive tract physiology and link in Gq/b-arrestin. Finding that GPR64 can regulate CFTR is novel and interesting. The major problem with this paper, however, is that there is too much data, yet not enough controls. My recommendation would be to focus the paper on GPR64 and CFTR story and try to keep in the data that is really needed to justify these conclusions. Then, with the space that you've saved, perform the controls that are needed and save the additional data for other papers. The mouse experiments are nicely done and believable, but you fall down on the immunohistochemistry, western blots and patch clamping due to lack of controls. This paper is highly focused on CFTR, GPR64, Gq – it'd be nice to see for example, that GPR64 does not interact/colocalize with other ion channels (e.g. Ano1 ro whatever) etc. My comments are below.

General

What is the natural ligand for GPR64 in the testes? This is not discussed. For GPR64, this might not be known, but that's ok and it should be pointed out not avoided. GPR64 belongs to a superfamily of "adhesion GPCRs". From Wikipedia, "The defining feature of adhesion GPCRs that distinguishes them from other GPCRs is their hybrid molecular structure. The extracellular region of adhesion GPCRs can be exceptionally long and contain a variety of structural domains that are known for the ability to facilitate cell and matrix interactions." This has been reviewed and might give you a clue or at least give you something to discuss.

Patch clamp. As discussed below, more needs to be done by patch clamp to ensure that this is CFTR that you are recording (more inhibitors, demonstrate that the reversal potential changes appropriately). Also, is this CFTR that's basally active? More needs to be done to define this. What happens with forskolin or inhibitors of PKA/PKC to CFTR in WT and KO cells? Why is CFTR basally active in this cell type? How is GPR64/Gq regulating CFTR? Is it affecting N or Po? Single channel records would be nice to see, as would surface biotinylation/western blots to see if this process affects CFTR trafficking.

WBs are too cropped and often lack controls to demonstrate specificity (except those with knockout mice). Also, integrated densitometry should be included for all WBs. There are also specific issues with the CFTR westerns (i.e. I think that they're inverted). Here, the reviewers should indicate which is the mature band.

Immnuo – Not enough info is presented to convince me that what you are seeing takes place in non-ciliated cells. Please provide light images and more extensive co-localization markers to verify this.

Figure 1 - Going from A to R is excessive and many images are too small. Please consider splitting up this figure into 3.

Figure 1 - Why did you only do brain and testes? Why not do other organs that expresses CFTR? This would fit better into the theme of the paper (that is expanded upon in the discussion) that CFTR is highly expressed but GPR64 gives specificity by being more highly expressed in testes.

Figure 1 - The images are too small and there is not enough information. Also, there are no controls. Also, the authors claim to that GPR64 is localized to non-ciliated cells, but I cannot tell this from the information provided.

Figure 1 - The images are too small. Scale bars are hard to read.

Figure 1- I can barely see the cells on these images.

Figure 1I - H89 is only PKA-specific at nM levels. At 10 μm it will hit many other kinases – this needs to be addressed.

IBMX is a phosphodiesterase inhibitor – This should be stated in both the Results section and discussion section rather than calling it a "cAMP motivator".

Figure 2.

In Figure 2 - The authors measure duct width as an indicator of fluid secretion. The first add a series of Ca^2+^ channel inhibitors and then a CFTR-specific inhibitor. This is a strange juxtaposition. Ca^2+^ is not present in biological fluids at a sufficiently high concentration to be an osmotic driver of fluid secretion and secretion is mediated by Cl, HCO_3_^-^, Na^+^ and K^+^. As such, a more conventional approach where inhibits of Cl, HCO_3_^-^, Na^+^ and K^+^ transport should be employed. For example, as well as GLYH101, DIDS, bumetanide, niflumic acid could be added as inhibitors of Cl^-^ transport. Similarly, amiloride (Na+) etc. should also be added. Inhibiting Ca^2+^ channels is a different question, and if they are going to go this route, they should also inhibit Orai1/STIM1, chelate Ca^2+^ etc.

Figure 2 - I suspect that something is wrong with your measurements. Inhibition of CFTR does not usually lead to changes in intracellular Cl^-^ measurements since there are many other anion channels/exchangers that can modulate intracellular Cl^-^ homeostasis. Indeed, cells from CF patients have normal intracellular Cl^-^ levels. MQAE is a non-rationmetric dye that has all the problems associated with this type of dye – for example, changes in cell size can concentrate or dilute the dye which will change fluorescence. Or, is dye loading normal in these cells? What if GPR64 affects xenobiotic pumps like MRP1 that can extrude fluorescent dyes?

Figure 2 - Are you really seeing a chronic pH change of >1? This is huge and likely incompatible with normal cellular function. See above comments about dye extrusion. Also, give the potential importance of this, where are the controls? What happens in bicarb free media? With acetazolamide or inhibition of Na/H or H/K exchange?

Figure 2 - You really should be using Pearson's correlation to measure and quantify colocalization. Controls are needed – it'd be nice to show an ion channel that does not co- colocalize. Also, showing some antibody specificity by showing secondary only.

Figure 2 - The current traces look like CFTR but a sizeable amount of current remains after addition of GLYH101. Is there leak or another background current. To prove that this is CFTR other anion channel inhibitors should be used (e.g. DIDs etc.). Also, these experiments are performed in near-symmetrical solutions. The authors should reduce the extracellular solution by 100 mM or more and show that they can get a Nernstian shift in reversal potential.

Figure 3

Figure 3 - I have the same issues with the ic Cl/pH measurements. Here, the pH difference is smaller, but WT is closer to 7.

Figure 3 - The above statement that not enough is done to.

Figure 3 - No controls (i.e. other proteins) – also they should include ciliated cell-specific markers such as anti-acetylated a tubulin which is only found in cilia. I'm not convinced by their claim that this effect doesn't occur in cilia. Also, there is no quantification.

Figure 3 – Why did they do spleen as a control here (brain was used in Figure 1). I think that a more rational approach should be used – i.e. some other CFTR-expressing and some non-CFTR expressing tissues.

Figure 3 - Please include a positive control that can alter IP levels such as ATP which will activate Gq via purinergic receptors. Since IP metabolism can vary depending on the cells needs, a common approach to meaure IP levels is to add lithium to prevent IP degradation and look for total IP levels of all species.

Figure 3 - See above but WBs are too cropped and don't have controls like cells that do not express GPR64.

Figure 4

Figure 4 - Same comments as above.

Figure 4 - Same comments as above. (needs controls and better quantification).

Figure 4 - CFTR is usually seen as Band C (mature, glycosylated) and Band B (immature, minimally glycosylated). Normally Band C is the predominant form in non-CF cells. Thus, unless your cells are CF or very different, I strongly suspect that your CFTR blots are upside down.

Figure 5

Figure 5 - Same comments as above.

5B-C - Same comments as above. (needs controls and better quantification).

Figure 5 - For the CFTR electrophys in this paper, the reversal potential is negative (unlike Figure 2 where it is 0). This is surprising given the predicted reversal potential for Cl^-^ is ~0. Why is this? Did you change your conditions relative to Figure 2?

Figure 5 - Given the length of the paper, this in my opinion is too much. This data should be removed and saved for another paper.

Figure 6

Figure 6 - Please spell out what ED, IS stand for on the figures. There's room. Also, please add arrows pointing to sperm accumulation etc.

Supplement – many of the same concerns arise. The figures are very small, immunofluorescence doesn't have adequate controls and Western blots are too cropped.

Discussion section – given all of the data presented. The discussion is very short. I would like to see more consideration of GPR64s physiological role as well as more discussion of CFTR regulation.

Reviewer #2:

Gq activity and B-arrestin1-scaffolding are required for male fertility through mediating CPR64/CFTR coupling.

Overview: This is a comprehensive study evaluating the role for a CFTR/GPR64 complex in fluid reabsorption across efferent ducts of the testis – a function that is vital for male fertility. Further the authors interrogated the regulation of this complex by Gq and B-arrestin. For this most part- these studies are robust and convincing with respect to the role of GPR64 in this function. However- the role of CFTR in this function could use additional supportive data.

Specific comments:

Figure 1 focuses on the tubule fluid transport properties of efferent ductules- studied ex-vivo. These studies clearly show that disruption of GPR64 and Gq impairs fluid reabsorption, morphology and sperm count.

On the other hand- the role for CFTR is not clear as increases and decreases in activation (via kinase activators and inhibitors respectively) leads to the same effect of luminal swelling.

Figure 2 shows that disruption of GPR64 alters the sensitivity of the ductular fluid transport to ion channel blockers (calcium transport proteins in addition of a CFTR channel blocker). They also show that chloride channel activity contributes to fluid transport. Their data falls somewhat short of proving that this activity is CFTR mediated. The regulatory properties of the conductance (i.e. regulation by PKA) was not shown and this is important. Also, while GlyH-101 is a well-known CFTR channel blocker- it is not specific and inhibits other chloride channels, including the SLC26A9 channel. The authors should include another inhibitor (i.e. CFTRinh-172) to test specificity.

Figure 4 aims to show that Β-arrestin1 contributes to fluid reabsorption in these tubes and its expression promotes co-localization of GPR64 and CFTR. However, labeling of the diagrams in Figure 4 seems somewhat confusing- does 4J show a line scan of GPR64 and CFTR localization or BArr1 and GPR64? The pattern of staining for GPR64 and CFTR looks similar but not overlapping- what compartments are the two proteins localized after BArr1 KO?

Figure 5 shows the consequence of co-expressing CFTR with GPR64 Wt or mutants bearing substitution in intracellular loops 2 and 3. This is a comprehensive set of studies supporting the role for these loops in mediating functional interaction between GRP64 and CFTR. However- an important control would include single transfections with GPR64 (no CFTR) to ensure that it is not modulating a distinct chloride channel. Western blotting to ensure expression of the each of the mutant GPR64 proteins would also be helpful to support the conclusion that there are site specific effects in the interaction- rather than reporting differences in protein abundance.

Reviewer #3:

The manuscript under review considers interaction between GPR64, CFTR and arrestins in non-cilliated cells of the male reproductive system. In general, the content of this study is interesting and relevant. However, a number of significant technical concerns with the presented study raise doubts about key results.

Major points:

Several studies have shown that GlyH-101 is not entirely specific for CFTR, for instance at 50 microM GlyH-101 inhibits SLC26A9. A panel of CFTR inhibitors, including CFTRinh172 should have been considered for studies presented in Figure 2. In addition, qPCR or similar should be employed to unambiguously determine that SLC26A9 is not a relevant player in cellular system under consideration.

In terms of the intracellular measurements of [Cl^-^], MQAE is essentially completely insensitive to [Cl^-^] above 100 mM, so it is hard to see how intracellular [Cl^-^] could be determined to be ~140 mM. In this regard, it is notable that the calibration curve for MQAE versus [Cl^-^] presented in the supplementary data is only extended to ~80 mM [Cl^-^] (i.e., well below the reported value). As such, it is difficult to have any confidence in the presented values of [Cl^-^].

In terms of the GlyH-101 studies presented in Figure 2, do driving forces predict that CFTR inhibition would mediate accumulation of cytoplasmic [Cl^-^]? What are the consequences of such inappropriately high intracellular [Cl^-^]? What happens to the concentration of cations, and the membrane potential of cells with such non-physiological levels of [Cl^-^]?

In terms of the pH measurements presented in Figure 2, sensitivity of BCECF to determine pH above ~pH 7.8 is limited. In addition, no calibration curve is presented – it is critical to demonstrate that the employed method would accurately determine the reported pKa for the fluorescent probe employed, i.e., BCECF, to have confidence that a pH of 8.4 could be accurately determined. The presented methods are unclear, for instance a 25 mM bicarbonate solution would require gassing with 5% CO2, but this is not mentioned. The description of what was measured in the Results was vague, inner solution of efferent ductules does not imply that cytoplasmic pH was determined.

In terms of the patch clamp analysis presented in Figure 2, prior studies by Muanprasat and colleagues have demonstrated that GlyH-101 inhibition of CFTR is strongly dependent on membrane potential – as would be expected for a charged molecule with a pore occluding mechanism of action. As such, GlyH-101 inhibition alters CFTR current-voltage curves from being linear to showing inward rectification. This is apparently not observed in the data presented in Figure 2 (where I-V curve remains linear). There is a concentration dependence of this phenomenon, however, I was unable to find information about how much GlyH-101 was used in the Legend or Material and Methods section for the presented data. In general, most studies used 25 microM GlyH-101. If this concentration was used in Figure 2, then inward rectification of CFTR I-V relationship would definitely be anticipated. Consideration of submaximal concentrations of GlyH-101, per Muanprasat and colleagues, should be considered to provide confidence that CFTR currents are really being observed in the reported data. In addition, delivery of PKA (in the pipette) is typically used for excised patch data, as such, experimental data should be presented for whole cell recordings with consideration of an alternative CFTR stimulant such as forskolin. In addition, for patch clamp data presented in Figure 3, pharmacological validation that currents are CFTR -dependent should be presented.

For the data presented in Figure 5, does stimulation of CFTR with an alternative agonist, such as forskolin, mediate similar cytoplasmic chloride concentration reduction? Molecular details of how GPR64 is activated have been elucidated and are considered by the authors in the supplemental data, for instance in regard to GPR64beta elevating of cAMP. Does the carboxy-terminal fragment of GPR64 also reduce cytoplasmic [Cl^-^]?

In Figure 2, the mere co-localization of two proteins to a membrane determined imaged by confocal microscopy does not indicate or suggest that a complex with functional coupling exists. It indicates that two proteins are targeted to the same membrane. This same concern is relevant for data presented in Figure 3, Figure 4, and Figure 5.

For co-IP experiments shown in Figure 4, there is insufficient explanation (in Materials and methods section, Results section, Legend etc.) to comprehend what is being done. By elimination, I assume the anti-HA blot was against arrestins, but, this is not detailed (for instance, I cannot find details of HA-tagged arrestin constructs).

[Editors’ note: what now follows is the decision letter after the authors submitted for further consideration.]

Thank you for resubmitting your work entitled "Gq activity- and β-arrestin-1 scaffolding-mediated GPR64/CFTR coupling are required for male fertility" for further consideration at *eLife*. Your revised article has been evaluated by Didier Stainier (Senior editor), a Reviewing editor, and three reviewers.

The manuscript has been improved but there are some remaining problematic issues that need to be addressed, as outlined below:

1) The intracellular Cl^-^ measurements remain problematic due to the lack of a radiometric method. These measurements should be either replaced by suitable radiometric or electrophysiological measurements or removed from the manuscript.

2) Western blots need controls and markers.

3) The CFTR currents are extremely small. The i/v shift with gluconate is too small and not typical for CFTR. Original whole cell overlay currents or continuous recordings should be shown. What is the proof in addition to CFTRinh172 that the authors truly measured CFTR currents?

4) Figure 4 suggests a GPR64-dependent expression of CFTR-mRNA. How does this fit to the expression data shown in Figure 5/I and to the data shown in Figure 6?

5) Figure 10: In what cells were these data obtained? No control for expression of the various GPR64 mutants is provided.

---

## [Author Response]

[Editors’ note: the author responses to the first round of peer review follow.]

Reviewer #1:In this paper, Zhang et al. link GPR64-dependent activation of CFTR with male reproductive tract physiology and link in Gq/b-arrestin. Finding that GPR64 can regulate CFTR is novel and interesting.

Thank you for your positive comments.

The major problem with this paper, however, is that there is too much data, yet not enough controls. My recommendation would be to focus the paper on GPR64 and CFTR story and try to keep in the data that is really needed to justify these conclusions. Then, with the space that you've saved, perform the controls that are needed and save the additional data for other papers. The mouse experiments are nicely done and believable, but you fall down on the immunohistochemistry, western blots and patch clamping due to lack of controls. This paper is highly focused on CFTR, GPR64, Gq – it'd be nice to see for example, that GPR64 does not interact/colocalize with other ion channels (e.g. Ano1 ro whatever) etc. My comments are below.Thank you for your helpful suggestions, which have improved the flow of our story and made the manuscript more compelling by including appropriate controls. We have added additional controls, including immunohistochemistry, Western blots and patch clamp experiments. For example, we have used another ion channel, Ano1, as a negative control for the co-localization of GPR64 with CFTR (Figure 1—figure supplement 1 and Figure 5—figure supplement 4). We have also reorganized the data to clarify the results. Please see the following point-by-point responses.GeneralWhat is the natural ligand for GPR64 in the testes? This is not discussed. For GPR64, this might not be known, but that's ok and it should be pointed out not avoided. GPR64 belongs to a superfamily of "adhesion GPCRs". From Wikipedia, "The defining feature of adhesion GPCRs that distinguishes them from other GPCRs is their hybrid molecular structure. The extracellular region of adhesion GPCRs can be exceptionally long and contain a variety of structural domains that are known for the ability to facilitate cell and matrix interactions." This has been reviewed and might give you a clue or at least give you something to discuss.

Thank you for your helpful suggestion. We have now included an introduction of GPR64 as an adhesion GPCR member in subsection “Molecular determinants of GPR64 coupling with G protein subtypes and their contribution to the regulation of CFTR activity in vitro”, and a discussion of the potential GPR64 activation mechanism in the Discussion section in the revised manuscript.

Patch clamp. As discussed below, more needs to be done by patch clamp to ensure that this is CFTR that you are recording (more inhibitors, demonstrate that the reversal potential changes appropriately). Also, is this CFTR that's basally active? More needs to be done to define this. What happens with forskolin or inhibitors of PKA/PKC to CFTR in WT and KO cells? Why is CFTR basally active in this cell type? How is GPR64/Gq regulating CFTR? Is it affecting N or Po? Single channel records would be nice to see, as would surface biotinylation/western blots to see if this process affects CFTR trafficking.

Thank you for your helpful comments regarding the working mechanism of CFTR in the efferent ductules. We have performed substantial new experiments, and a detailed response follows:

1) We have performed new electrophysiology whole-cell Cl^-^ recordings of primary GPR64 promoter-labeled efferent ductule cells derived from wild-type and GPR64^-/Y^ mice at normal Cl^-^ concentrations or with the substitution of Cl^-^ by gluconate (Gluc^-^) in the bath solution, with or without different agonists or pharmacological inhibitors, including forskolin, the PKC inhibitor Ro 31-8220, the PKA inhibitor PKI 14-22, the ANO1 inhibitor Ani9 (Namkung, 2016), the chloride-bicarbonate exchanger inhibitor DIDS and the specific CFTR inhibitor CFTR-Inh-172 (Figure 6, Figure 7, Figure 6—figure supplement 1 to Figure 6—figure supplement 3, and Figure 7—figure supplement 1).

In GPR64 promoter-labeled wild-type efferent ductule cells, we observed an outwardly-rectifying whole-cell Cl^-^ current (I_GPR64-ED_), which was significantly diminished in response to substitution of the bath Cl^-^ solution with Gluc^-^ (148.5 mM Cl^-^ was replaced with 48.5 mM Cl^-^ and 100 mM Gluc^-^) (Figure 6 and Figure 6—figure supplement 1). The change in the reversal potential (E_rev_) followed the Nernst equation (Figure 6 and Figure 6—figure supplement 1). However, the whole-cell Cl^-^ current of the GPR64 promoter-labeled efferent ductule cells derived from GPR64^-/Y^ mice was significantly lower than that in cells derived from their wild-type littermates, which showed only a modest difference in response to substitution of the bath Cl^-^ solution with Gluc^-^ (Figure 6). These results suggested that GPR64 deficiency in the efferent ductules significantly reduced the whole-cell Cl^-^ current of GPR64 promoter-labeled non-ciliated cells.

2) Our Western blot data showed that the ablation of GPR64 had no effect on CFTR protein expression levels (Figure 5 and Figure 5—figure supplement 5). To examine whether CFTR is basically activated in GPR64 promoter-labeled cells, we utilized the CFTR-specific inhibitor CFTR-Inh-172, the CFTR agonist FSK^+^IBMX (Lu et al., 2010) and inhibitors for other Cl^-^ transporters, including ANI9 and DIDS. The whole-cell Cl^-^ current of GPR64 promoter-labeled wild-type efferent ductule cells was significantly reduced by the application of the CFTR inhibitor CFTR-Inh-172 but not by the ANO1 inhibitor ANI9 or the chloride-bicarbonate exchanger inhibitor DIDS, suggesting that CFTR is likely the main mediator of the Cl^-^ current of GPR64 promoter-labeled efferent ductule cells (Figure 6 and Figure 6—figure supplement 3). Moreover, the CFTR agonist FSK^+^IBMX induced similar Cl^-^ currents in cells derived from both wild-type and GPR64^-/Y^ mice (Figure 6 and Figure 6—figure supplement 2). Moreover, the application of CFTR-Inh-172 to GPR64 promoter-labeled cells derived from wild-type mice decreased the I_GPR64-ED_ to a level closer to the I_GPR64-ED_ of cells derived from GPR64^-/Y^ mice (Figure 6). These results suggested that CFTR in GPR64 promoter-labeled wild-type efferent ductule cells was generally active, and this activity was dependent on the expression of GPR64 in these cells.

3) To define the mechanism underlying CFTR activity in the efferent ductules, we performed a series of cell biology experiments. The GPR64 promoter-labeled efferent ductule cells derived from GPR64^-/Y^ cells had much lower IP1 and cAMP levels compared with cells derived from wild-type cells (Figure 7 in the revised manuscript). These increases potentially cause the basal activity of CFTR, which is activated by PKA and PKC downstream of IP1 and cAMP (Chappe et al., 2003; Guggino and Stanton, 2006), and our data showed the constitutive Gq and Gs activity of GPR64 following its expression in HEK293 cells, which promoted CFTR activity in a heterologous system (Figure 9, Figure 10 and Figure 9—figure supplement 1 to Figure 9—figure supplement 6). The inhibitory effects of the PKC inhibitor Ro 31-8220 and the PKA inhibitor PKI 14–22 in the efferent ductules further supported this hypothesis (Figure 7 and Figure 7—figure supplement 1). The increased IP1 and cAMP levels were attributed to the constitutive activity of GPR64, a property shared by many adhesion GPCR members (Gupte et al., 2012; Paavola and Hall, 2012). Currently, single channel recording of GPR64-labeled efferent ductules is technologically challenging as we are unable to decrease the noise, which is essential for a good signal-to-noise ratio. We are still optimizing conditions for these experiments and may include these results in a future manuscript.

4) To monitor CFTR trafficking in the efferent ductule cells, we performed co-immunostaining experiments. In both GPR64^-/Y^ and wild-type mice, CFTR colocalized with ezrin, an apical membrane marker for the efferent ductules (Figure 8—figure supplement 2), suggesting that GPR64 deficiency does not affect CFTR trafficking. However, in β-arrestin-1-deficient mice, CFTR was localized away from both GPR64 and ezrin (Figure 8 and Figure 8), suggesting that its proper localization in the apical membrane is dependent on β-arrestin-1 but not on β-arrestin-2 or GPR64.

WBs are too cropped and often lack controls to demonstrate specificity (except those with knockout mice). Also, integrated densitometry should be included for all WBs. There are also specific issues with the CFTR westerns (ie I think that they're inverted). Here, the reviewers should indicate which is the mature band.

Thank you for these helpful suggestions. We have included larger gel images and statistical analysis of the integrated densitometry (Figure 8—figure supplement 3, Figure 9—figure supplement 2 to Figure 9—figure supplement 4, the Figure 5, the Figure 8, the Figure 7—figure supplement 3 and the Figure 10—figure supplement 2). The mature band of CFTR is indicated in Figure 5, Figure 8, Figure 7—figure supplement 3 and Figure 8—figure supplement 3 of the revised manuscript.

Immnuo – not enough info is presented to convince me that what you are seeing takes place in non-ciliated cells. Please provide light images and more extensive co-localization markers to verify this.

Thank you for these helpful suggestions. We have performed new immunofluorescence experiments using α-acetylated tubulin as a marker for the ciliated cells in the efferent ductules. As shown in Figure 1 and Figure 1—figure supplement 1 in the revised manuscript, the results indicated that GPR64 localized in the non-ciliated cells.

Figure 1 - Going from A to R is excessive and many images are too small. Please consider splitting up this figure into 3.

Thank you for your suggestion. We have divided the contents of the previous Figure 1 into three figures in the revised manuscript (Figure 1–Figure 3).

Figure 1. Why did you only do brain and testes? Why not do other organs that expresses CFTR? This would fit better into the theme of the paper (that is expanded upon in the discussion) that CFTR is highly expressed but GPR64 gives specificity by being more highly expressed in testes.

Thank you for your helpful consideration. GPR64 is a G protein coupled receptor with selective cell subtype expression in the efferent ductules (Figure 1). Therefore, we initially labeled GPR64-expressing cells in the efferent ductules and examined the immediate downstream effectors of G protein subtype expression in Figure 1 (Figure 1 in our original manuscript). Later, when we characterized the mechanism underlying GPR64-mediated suppression of fluid reabsorption dysfunction in the efferent ductules, according to the reviewer’s suggestion, we performed new experiments with liver and brain tissues. As shown in the revised Figure 4 and Figure 4—figure supplement 1, we examined the expression of important channels and transporters related to fluid reabsorption in both brain and liver tissues. The liver demonstrated high expression of CFTR but little GPR64 expression.

Figure 1 - The images are too small and there is not enough information. Also, there are no controls. Also, the authors claim to that GPR64 is localized to non-ciliated cells, but I cannot tell this from the information provided.

Thank you for your helpful suggestions. We have enlarged Figure 1 so that it is more visible. We have performed new immunofluorescence experiments using α-acetylated tubulin as a marker for ciliated cells in the efferent ductules. As shown in Figure 1 and Figure 1—figure supplement 1 in the revised manuscript, GPR64 is localized in the non-ciliated cells of the efferent ducts.

Figure 1 – the images are too small. Scale bars are hard to read.

Thank you for your helpful suggestion. We have enlarged the figure (new Figure 2 in the revised manuscript) accordingly.

Figure 1 can barely see the cells on these images.

Thank you for your helpful suggestion. We have enlarged the figure (new Figure 3 in the revised manuscript) accordingly.

Figure 1I - H89 is only PKA-specific at nM levels. At 10 μm it will hit many other kinases – this needs to be addressed.

Thank you for these helpful suggestions. We have changed the H89 concentration to 500 nM, and we also used other drugs, such as PKI14-22 (a specific PKA inhibitor) and NF449 (a Gs inhibitor), to characterize the Gs-PKA pathway in fluid reabsorption in the efferent ductules (for example, new Figure 2 in the revised manuscript).

IBMX is a phosphodiesterase inhibitor – this should be stated in both the results (p5) and discussion rather than calling it a "cAMP motivator".

Thank you for your suggestions. We have clarified that IBMX is a phosphodiesterase inhibitor in the figure legend and results (line 146 in the revised manuscript).

Figure 2.In Figure 2, the authors measure duct width as an indicator of fluid secretion. The first add a series of Ca^2+^ channel inhibitors and then a CFTR-specific inhibitor. This is a strange juxtaposition. Ca^2+^ is not present in biological fluids at a sufficiently high concentration to be an osmotic driver of fluid secretion and secretion is mediated by Cl, HCO_3_^-^, Na^+^ and K^+^. As such, a more conventional approach where inhibits of Cl, HCO_3_^-^, Na^+^ and K^+^ transport should be employed. For example, as well as GLYH101, DIDS, bumetanide, niflumic acid could be added as inhibitors of Cl^-^ transport. Similarly, amiloride (Na+) etc should also be added. Inhibiting Ca^2+^ channels is a different question, and if they are going to go this route, they should also inhibit Orai1/STIM1, chelate Ca^2+^ etc.

Thank you for your very helpful suggestions to improve our paper. We have incorporated all of the drugs you have suggested in new experiments to confirm our results.

We performed the following new experiments: first, we examined the expression of different chloride channels as well as HCO_3_^-^, Na^+^ and K^+^ transporters in GPR64-expressing efferent ductule cells and compared them with non-GPR64 promoter-labeled cells (Figure 4 in the revised manuscript). We found that CFTR, NHE1, NHE3, NKCC, DRA, SLC26a9, CAII, CLCA1, TRPC3, Cav1.2 and Cav1.3 are expressed in GPR64 promoter-labeled cells, whereas ANO1, V-ATPase and Cav2.2 demonstrated relatively lower expression levels (Figure 4).

We next used bumetanide to block the Na-K-Cl cotransporter NKCC, nifumic acid (NFA) to block the calcium-dependent chloride channel CaCC, and ANI9 to block anoctamin-1 (ANO1) (Figure 4). None of these inhibitors had significant effects on fluid reabsorption in the efferent ductules in ligation experiments. Similarly, chelating calcium with EGTA or the administration of TRP channel blockers had no significant effects (Figure 4). The application of 4,4'-diisothiocyano-2,2'-stilbenedisulfonic acid (DIDS) to block chloride-bicarbonate exchange induced small effects only after 60 hours (Figure 4). The application of amiloride to block sodium/hydrogen antiporter NHE1 activity resulted in different phenotypes compared to those of GPR64^-/Y^ mice (Figure 4—figure supplement 1). In contrast, blocking CFTR activity with either GlyH-101 or CFTR-Inh-172 exerted significant effects on fluid reabsorption in the efferent ductules and pheno-copied GPR64 ^-/Y^ mice (Figure 4). Collectively, these results indicated that CFTR plays an important role in fluid reabsorption and is a potential downstream effector of GPR64 in the efferent ductules.

Figure 2 suspect that something is wrong with your measurements. Inhibition of CFTR does not usually lead to changes in intracellular Cl^-^ measurements since there are many other anion channels/exchangers that can modulate intracellular Cl^-^ homeostasis. Indeed, cells from CF patients have normal intracellular Cl^-^ levels. MQAE is a non-rationmetric dye that has all the problems associated with this type of dye – for example, changes in cell size can concentrate or dilute the dye which will change fluorescence. Or, is dye loading normal in these cells? What if GPR64 affects xenobiotic pumps like MRP1 that can extrude fluorescent dyes?

Thank you for your helpful suggestions. We have changed our method for intracellular Cl^-^ measurement by implementing the use of specific GFP variants in the YFP family (mCIY-8M) whose fluorescence is affected by halide concentrations in the cell (Figure 5—figure supplement 1; Figure 5 and Figure 7 in the revised manuscript).

Figure 2 - Are you really seeing a chronic pH change of >1? This is huge and likely incompatible with normal cellular function. See above comments about dye extrusion. Also, give the potential importance of this, where are the controls? What happens in bicarb free media? With acetazolamide or inhibition of Na/H or H/K exchange?

Thank you for your very helpful comments. We have performed new experiments with another pH indicator, 5'(and 6')-carboxy-10-dimethylamino-3-hydroxy-spiro[7H-benzo[c]xanthene-7,1'(3H)-isobe nzofuran]-3'-one (carboxy SNARF-1) (Figure 5—figure supplement 3). Newly acquired data is shown in Figure 5, Figure 7 and Figure 8 in the revised manuscript. We did see a chronic pH change, which was below 1, which agreed well with your prediction. The pH imbalance in GPR64^-/Y^ mice was rescued by bicarb-free media or application of the carbonic anhydrase inhibitor acetazolamide (Figure 5—figure supplement 3). Application of the Na/H exchanger inhibitor amiloride is confounding, as it decreased the pH but caused acute fluid reabsorption dysfunction (Figure 4—figure supplement 1). The role of the Na/H exchanger in fluid reabsorption may be the focus of a future study.

Figure 2 - You really should be using Pearson's correlation to measure and quantify colocalization. Controls are needed – it'd be nice to show an ion channel that does not co- colocalize. Also, showing some antibody specificity by showing secondary only.

Thank you for your helpful suggestions. We changed our co-localization measurement method to Pearson’s correlation analysis to better explain the results that we acquired (Figure 5, Figure 5, Figure 8, Figure 8, Figure 8 and Figure 8; Figure 8—figure supplement 2–2B and Figure 9—figure supplement 1). We included another ion channel, Ano1, as a negative control (Figure 5—figure supplement 4). We also demonstrated the specificity of our secondary antibody, which is shown in Figure 1—figure supplement 1–1B.

Figure 2 - The current traces look like CFTR but a sizeable amount of current remains after addition of GLYH101. Is there leak or another background current. To prove that this is CFTR other anion channel inhibitors should be used (e.g. DIDs etc.). Also, these experiments are performed in near-symmetrical solutions. The authors should reduce the extracellular solution by 100 mM or more and show that they can get a Nernstian shift in reversal potential.

Thank you for your helpful suggestions. We have examined the effects of the

Ano1 inhibitor Ani9, the non-selective chloride-bicarbonate exchanger DIDS and the CFTR-specific inhibitor CFTR-Inh-172 on the electrophysiological recording of GPR64 promoter-labeled efferent ductule cells (Figure 6 and Figure 6—figure supplement 2 to Figure 6—figure supplement 3). The results indicated that only the blockade of CFTR activity significantly inhibited the I_GPR64-ED_. Considering the sizable current in wild-type mice after application of the CFTR-specific inhibitor CFTR-Inh-172, a plausible explanation is that the Cl^-^ current is mediated by neither CFTR nor GPR64 in these efferent ductule cells. When we performed these experiments by substituting Cl^-^ in the bath buffer with Gluc^-^ (148.5 mM Cl^-^ was replaced by 48.5 mM Cl^-^ and 100 mM Gluc^-^) (Figure 6), accompanied by the inhibitory effects of CFTR inh-172, we still observed a decrease in the current and a change in the reversal potential (E_rev_) that followed the Nernst equation (Figure 6—figure supplement 3C). However, these currents did not contribute to fluid reabsorption dysfunction in response to GPR64 deficiency, as there were no significant differences between GPR64^-/Y^ mice and their wild-type littermates in terms of the I_GPR64-ED_ after the application of CFTR-Inh-172 (Figure 6).

Figure 3
Figure 3- I have the same issues with the ic Cl/pH measurements. Here, the pH difference is smaller, but WT is closer to 7. Figure 3. The above statement that not enough is done to.

Thank you for your helpful suggestions. We have used alternative intracellular Cl/pH level sensors (mCIY-8M and the carboxy SNARF-1) for superior measurement (Figure 5 and Figure 5—figure supplement 1 and Figure 5—figure supplement 3).

Figure 3 No controls (i.e. other proteins) – also they should include ciliated cell-specific markers such as anti-acetylated a tubulin which is only found in cilia. I'm not convinced by their claim that this effect doesn't occur in cilia. Also, there is no quantification.

Thank you for your helpful comments. We used an acetylated α-tubulin antibody to label the ciliated cells (Figure 1 and Figure 1—figure supplement 1). Co-localization experiments suggested that Gq is mainly expressed in non-ciliated cells (Figure 7 in the revised manuscript), and Gq and GPR64 are colocalized in the same cells (Figure 7 in the revised manuscript). The quantification is shown in Figure 7—figure supplement 2 in the revised manuscript

Figure 3 – why did they do spleen as a control here (brain was used in Figure 1). I think that a more rational approach should be used – i.e. some other CFTR-expressing and some non-CFTR expressing tissues.

Thank you for your suggestions. We have included the liver (which has higher CFTR expression) and brain (relative lower CFTR expression) as controls for the efferent ductules in our revised manuscript (Figure 7 and Figure 7—figure supplement 3).

Figure 3. Please include a positive control that can alter IP levels such as ATP which will activate Gq via purinergic receptors. Since IP metabolism can vary depending on the cells needs, a common approach to measure IP levels is to add lithium to prevent IP degradation and look for total IP levels of all species.

Thank you for your suggestions. We have included ATP stimulation in Figure 7 in the revised manuscript, and all IP levels were assayed with lithium incubation in our assays accordingly (see Materials and methods section).

Figure 3. See above but WBs are too cropped and don't have controls like cells that do not express GPR64.

We have included new WB results in Figure 8—figure supplement 3, Figure 9—figure supplement 2 and Figure 9—figure supplement 3 to demonstrate GPR64 expression.

Figure 4
Figure 4 - Same comments as above.Figure 4 - Same comments as above. (needs controls and better quantification).Figure 4. CFTR is usually seen as Band C (mature, glycosylated) and Band B (immature, minimally glycosylated). Normally Band C is the predominant form in non-CF cells. Thus, unless your cells are CF or very different, I strongly suspect that your CFTR blots are upside down.

Thank you for your helpful suggestions. We have used new sensors to measure the Cl^-^ concentration and pH (Figure 6 and Figure 5—figure supplement 1 and Figure 5—figure supplement 3). The interactions of GPR64 with β-arrestin1/β-arrestin2 or GPR64 with CFTR were confirmed by co-immunoprecipitation experiments (Figure 5 and Figure 5—figure supplement 5). The co-localization of GPR64 with CFTR was confirmed by immunostaining and analyzed by Pearson’s correlation analysis (Figure 5). The localization of CFTR/GPR64 with the apical membrane or microvilli was assayed by co-immunostaining with the marker ezrin (Figure 5, Figure 8—figure supplement 2 and Figure 5—figure supplement 4).

As indicated correctly by the reviewer, we incorrectly included the CFTR blots upside down in our previous manuscript. We have corrected this accordingly and have indicated the mature CFTR band in Figure 5 and Figure 8 in the revised manuscript.

Figure 5
Figure 5 - Same comments as above.5B–C - Same comments as above. (needs controls and better quantification).Figure 5. For the CFTR electrophys in this paper, the reversal potential is negative (unlike Figure 2 where it is 0). This is surprising given the predicted reversal potential for Cl^-^ is ~0. Why is this? Did you change your conditions relative to Figure 2?

Thank you for your helpful suggestions. For co-localization, we have included a negative control (secondary antibody only in Figure 1—figure supplement 1 in the revised manuscript). We used an alternative chloride sensor in Figure 9 (Figure 5 in the original manuscript) for more accurate measurements. We performed new sets of electrophysiological recording experiments, and the reverse potentials of these measurements, which are summarized in Figure 9—figure supplement 6, almost completely followed Nernst function predictions

Figure 5 - Given the length of the paper, this in my opinion is too much. This data should be removed and saved for another paper.

Thank you for your helpful suggestions. These data are helpful for providing direct evidence of the coupling of GPR64 to Gs and Gq as well as how they function in fluid reabsorption in the efferent ductules.

Figure 6.Figure 6 - Please spell out what ED, IS stand for on the figures. There's room. Also, please add arrows pointing to sperm accumulation etc.

Thank you for your helpful suggestions. We have included the full names of the structures and tissues used in the revised manuscript (Figure 11). We have also added arrows pointing to sperm accumulation.

Supplement – many of the same concerns arise. The figures are very small, immunofluorescence doesn't have adequate controls and Western blots are too cropped.

Thank you for your helpful suggestions. We have included larger immunostained images and reduced cropping of the WB results in the supplemental data accordingly.

Discussion section – given all of the data presented. The discussion is very short. I would like to see more consideration of GPR64s physiological role as well as more discussion of CFTR regulation.

Thank you for your helpful suggestions. We have included a discussion of the physiological roles of GPR64 and CFTR regulation in the Discussion section.

Reviewer #2:Gq activity and B-arrestin1-scaffolding are required for male fertility through mediating CPR64/CFTR coupling.Overview: This is a comprehensive study evaluating the role for a CFTR/GPR64 complex in fluid reabsorption across efferent ducts of the testis – a function that is vital for male fertility. Further the authors interrogated the regulation of this complex by Gq and B-arrestin. For this most part- these studies are robust and convincing with respect to the role of GPR64 in this function. However- the role of CFTR in this function could use additional supportive data.

Thank you for your positive comments.

Specific comments:Figure 1 focuses on the tubule fluid transport properties of efferent ductules- studied ex-vivo. These studies clearly show that disruption of GPR64 and Gq impairs fluid reabsorption, morphology and sperm count.On the other hand- the role for CFTR is not clear as increases and decreases in activation (via kinase activators and inhibitors respectively) leads to the same effect of luminal swelling.

Thank you for your helpful comments. Our data have clearly identified an essential role for the Gq-PKC pathway in fluid reabsorption and the whole-cell Cl^-^ current of the efferent ductules; however, Gs-PKA signaling is confounded. We therefore only drew conclusions regarding the role of Gq-PKC signaling in fluid reabsorption of the efferent ductules in the current manuscript.

According to our current data, we have demonstrated that the Gq protein is mainly expressed in GPR64-expressing cells, whereas the Gs protein is expressed in both GPR64-expressing and non-GPR64-expressing efferent ductule cells, as shown in Author response image 1.

**Author response image 1. respfig1:** Expression and localization of Gs in different types of efferent ductule cells. (**A**) The Gs was expressed in both GPR64-expressed cells and non-GPR64-expressed cells revealed by co-immunostaining. (**B**) The Gs expression in GPR64-promoter labeled cells and non GPR64-promoter labeled cells were examined by qRT-PCR after FACS. The results indicate that the similar mRNA level of Gs were detected in GPR64-promoter labeled cells and non GPR64-promoter labeled cells..

The Gs-PKA signaling in GPR64-labeled cells facilitated the analysis of CFTR function in maintaining Cl^-^ homeostasis, as supported by the inhibitory effects of the PKA inhibitor PKI14-22 and the Gs inhibitor NF-449 (Figure 2). However, activation of Gs-PKA signaling in non-GPR64-pm-labeled cells may have also impaired fluid reabsorption but in a different manner via an unknown mechanism (Figure 2—figure supplement 2). Therefore, distinct functions of Gs-PKA signaling in different efferent ductule cells may account for the confounding observations. Both acute activation of the Gs-cAMP pathway by forskolin/IBMX and the inhibition of Gs by NF449 impaired fluid reabsorption, suggesting a delicate function for Gs in the efferent ductules. The function of Gs in the efferent ductules may be investigated in future studies, but this is not the focus of our current manuscript.

Figure 2: Shows that disruption of GPR64 alters the sensitivity of the ductular fluid transport to ion channel blockers (calcium transport proteins in addition of a CFTR channel blocker). They also show that chloride channel activity contributes to fluid transport. Their data falls somewhat short of proving that this activity is CFTR mediated. The regulatory properties of the conductance (i.e. regulation by PKA) was not shown and this is important. Also, while GlyH-101 is a well-known CFTR channel blocker- it is not specific and inhibits other chloride channels, including the SLC26A9 channel. The authors should include another inhibitor (i.e. CFTRinh-172) to test specificity.

Thank you for your helpful suggestions. We have performed new experiments to clarify that these effects, shown in Figure 4 (previous Figure 2), are CFTR-mediated. All GlyH-101 data were confirmed by the CFTR-specific inhibitor CFTR-Inh-172 and were compared with the data obtained with other Cl^-^ transporter inhibitors (Figure 4, Figure 4; Figure 6 and Figure 6—figure supplement 2). We used the specific CFTR inhibitor CFTR-Inh-172, the PKA inhibitor PKI14-22 and the cAMP activator Forskolin for electrophysiological measurements of our primary efferent ductule cells (Figure 6 and Figure 6—figure supplement 1 to Figure 6—figure supplement 3). We have also included the CFTR inhibitor CFTR-Inh-172 in the chloride measurements in Figure 5 in the revised manuscript (previous Figure 2).

Figure 4 aims to show that Β-arrestin1 contributes to fluid reabsorption in these tubes and its expression promotes co-localization of GPR64 and CFTR. However, labeling of the diagrams in Figure 4 seems somewhat confusing- does 4J show a line scan of GPR64 and CFTR localization or BArr1 and GPR64? The pattern of staining for GPR64 and CFTR looks similar but not overlapping- what compartments are the two proteins localized after BArr1 KO?

Thank you for your very helpful comments. Compared to other receptors, β-arrestins are abundant proteins and their co-localization with receptors in resting states are not easy to be assayed, due to the quality of the antibody. We therefore used co-immunoprecipitation experiments of efferent ductules to examine the interaction of GPR64 with arrestins (new Figure 5 in the revised manuscript). The β-arrestin-1 was co-immunoprecipitated with GPR64 in the efferent ductules derived from wild-type mice, but not GPR64^-/Y^ mice (new Figure 5 and span class="jrnlFigRef" data-citation-string=" F5-S5 ">Figure 5—figure supplement 5 in the revised manuscript). Moreover, in the wild-type efferent ductules, both the GPR64 and CFTR are co-localized with Ezrin on the apical membrane (Figure 5 and Figure 8—figure supplement 2). In the β-arrestin-1^-/-^ mice, while the GPR64 still localized at the apical membrane, the CFTR was mislocalized relative to GPR64 and Ezrin in the efferent ductules (Figure 8). The disruption of the interaction of GPR64 with CFTR in the β-arrestin-1^-/-^ mice was also confirmed by co-immunoprecipitation results (Figure 8 in the revised manuscript).

Figure 5 shows the consequence of co-expressing CFTR with GPR64 Wt or mutants bearing substitution in intracellular loops 2 and 3. This is a comprehensive set of studies supporting the role for these loops in mediating functional interaction between GRP64 and CFTR. However- an important control would include single transfections with GPR64 (no CFTR) to ensure that it is not modulating a distinct chloride channel. Western blotting to ensure expression of the each of the mutant GPR64 proteins would also be helpful to support the conclusion that there are site specific effects in the interaction- rather than reporting differences in protein abundance.Thank you for your helpful suggestions. We have included new data for GPR64 transfected alone in in vitro electrophysiological experiments and chloride concentration measurements in the revised manuscript (Figure 9). We have also included the expression levels of different GPR64 mutations in Figure 10—figure supplement 2.Reviewer #3:The manuscript under review considers interaction between GPR64, CFTR and arrestins in non-cilliated cells of the male reproductive system. In general, the content of this study is interesting and relevant. However, a number of significant technical concerns with the presented study raise doubts about key results.

Thank you for your positive comments.

Major points:Several studies have shown that GlyH-101 is not entirely specific for CFTR, for instance at 50 microM GlyH-101 inhibits SLC26A9. A panel of CFTR inhibitors, including CFTRinh172 should have been considered for studies presented in Figure 2. In addition, qPCR or similar should be employed to unambiguously determine that SLC26A9 is not a relevant player in cellular system under consideration.

Thank you for your helpful suggestions. We have performed new experiments using CFTR-Inh-172 to specifically block CFTR in all studies related to CFTR functions (Figure 4, Figure 5, Figure 6, Figure 9 and Figure 9 in the revised manuscript). We have performed qRT-PCR experiments, and the results indicated that SLC26A9 is highly expressed in GPR64-promoter-labeled cells (Figure 4 in the revised manuscript). However, DIDS, an inhibitor known to inhibit SLC26A9, had little effect on the inhibition of fluid reabsorption in the efferent ductules (Figure 4), whereas the CFTR inhibitor CFTR-Inh-172 exerted prominent effects (Figure 4). Further electrophysiological experiments support this conclusion (Figure 6). Therefore, our results suggested that CFTR is the major player downstream of GPR64 for the regulation of fluid reabsorption in the efferent ductules. SLC26A9 may also contribute but potentially plays a more minor role. We have included these considerations in our Discussion section in the revised manuscript.

In terms of the intracellular measurements of [Cl^-^], MQAE is essentially completely insensitive to [Cl^-^] above 100 mM, so it is hard to see how intracellular [Cl^-^] could be determined to be ~140 mM. In this regard, it is notable that the calibration curve for MQAE versus [Cl^-^] presented in the supplementary data is only extended to ~80 mM [Cl^-^] (i.e., well below the reported value). As such, it is difficult to have any confidence in the presented values of [Cl^-^].

Thank you for your helpful suggestions. We have measured Cl^-^ concentrations using a new sensor, mCIY-8M (Figure 5—figure supplement 1). We have compared mCIY-H148Q and newly reported 8-mutations of mCIY, which show slightly better signals and calibration curves. We therefore used mCIY-8M to measure Cl^-^ concentration changes (Jayaraman et al., 2000; Zhong et al., 2014). The new experimental results are presented in revised Figure 5, Figure 7 and Figure 9).

In terms of the GlyH-101 studies presented in Figure 2, do driving forces predict that CFTR inhibition would mediate accumulation of cytoplasmic [Cl^-^]? What are the consequences of such inappropriately high intracellular [Cl^-^]? What happens to the concentration of cations, and the membrane potential of cells with such non-physiological levels of [Cl^-^]?

Thank you for your helpful comments. With the new Cl^-^ indicator mCIY-8M, the Cl^-^ concentration was more accurately measured and was much lower than previously measured. The changes in Cl^-^ and pH homeostasis may account for the fluid reabsorption dysfunction, and we did observe changes in the efferent ductule volume in ligation experiments (Figure 2 and Figure 4). Moreover, ATGR2 activation, which reportedly restores pH homeostasis, rescued the fluid reabsorption dysfunction (Figure 4). These results support the critical roles of Cl^-^ and pH homeostasis in fluid reabsorption in the efferent ductules.

In terms of the pH measurements presented in Figure 2, sensitivity of BCECF to determine pH above ~pH 7.8 is limited. In addition, no calibration curve is presented – it is critical to demonstrate that the employed method would accurately determine the reported pKa for the fluorescent probe employed, i.e., BCECF, to have confidence that a pH of 8.4 could be accurately determined. The presented methods are unclear, for instance a 25 mM bicarbonate solution would require gassing with 5% CO2, but this is not mentioned. The description of what was measured in the Results was vague, inner solution of efferent ductules does not imply that cytoplasmic pH was determined.

Thank you for your helpful comments. We have used a new pH sensor, carboxy SNARF-1, to determine the pH under different conditions presented in the manuscript (Figure 5—figure supplement 3) (Buckler and Vaughanjones, 1990; Thornell et al., 2017). A calibration curve is presented in Figure 5—figure supplement 3. For the 25 mM bicarbonate solution, we did perform gassing with 5% CO_2_.

We have revised the Materials and methods section accordingly.

In terms of the patch clamp analysis presented in Figure 2, prior studies by Muanprasat and colleagues have demonstrated that GlyH-101 inhibition of CFTR is strongly dependent on membrane potential – as would be expected for a charged molecule with a pore occluding mechanism of action. As such, GlyH-101 inhibition alters CFTR current-voltage curves from being linear to showing inward rectification. This is apparently not observed in the data presented in Figure 2 (where I-V curve remains linear). There is a concentration dependence of this phenomenon, however, I was unable to find information about how much GlyH-101 was used in the Legend or Material and Methods section for the presented data. In general, most studies used 25 microM GlyH-101. If this concentration was used in Figure 2, then inward rectification of CFTR I-V relationship would definitely be anticipated. Consideration of submaximal concentrations of GlyH-101, per Muanprasat and colleagues, should be considered to provide confidence that CFTR currents are really being observed in the reported data. In addition, delivery of PKA (in the pipette) is typically used for excised patch data, as such, experimental data should be presented for whole cell recordings with consideration of an alternative CFTR stimulant such as forskolin. In addition, for patch clamp data presented in Figure 3, pharmacological validation that currents are CFTR -dependent should be presented.

Thank you for your helpful comments. In the GPR64 promoter-labeled wild-type efferent ductule cells, we observed an outwardly-rectifying whole-cell Cl^-^ current (I_GPR64-ED_), which was significantly diminished in response to substitution of the bath Cl^-^ solution with Gluc^-^ (148.5 mM Cl^-^ was replaced by 48.5 mM Cl^-^ and 100 mM Gluc^-^) (Figure 6). The change in the reversal potential (E_rev_) followed the Nernst equation (Figure 6 and Figure 6—figure supplement 1 to Figure 6—figure supplement 3). However, the whole-cell Cl^-^ current of the GPR64 promoter-labeled efferent ductule cells derived from GPR64^-/Y^ mice was significantly lower than that of cells derived from their wild-type littermates, which showed a slight difference in response to substitution of the bath Cl^-^ solution with Gluc^-^ (Figure 6). These results suggested that GPR64 deficiency in the efferent ductules significantly reduced the whole-cell Cl^-^ current of GPR64 promoter-labeled non-ciliated cells. Moreover, we characterized the electrophysiological measurements by applying the specific CFTR inhibitor CFTR-Inh-172 and forskolin in primary GPR64 promoter-labeled efferent ductule cells derived from wild-type and GPR64^-/Y^ mice (new Figure 6). In response to CFTR-Inh-172 inhibition, we did observe an inwardly-rectifying I_GPR64-ED_ current (green color in Figure 6) instead of the original outwardly-rectifying current (Figure 6). The forskolin+IBMX data are shown in Figure 6. We added new data to previous Figure 3 to include the PKC inhibitor Ro 31-8220 (new Figure 7 in the revised manuscript). Application of these selective pharmacological blockers supported the notion that the observed whole-cell Cl^-^ current of GPR64 promoter-labeled efferent ductule cells was mediated by CFTR.

For the data presented in Figure 5, does stimulation of CFTR with an alternative agonist, such as forskolin, mediate similar cytoplasmic chloride concentration reduction? Molecular details of how GPR64 is activated have been elucidated and are considered by the authors in the supplemental data, for instance in regard to GPR64beta elevating of cAMP. Does the carboxy-terminal fragment of GPR64 also reduce cytoplasmic [Cl^-^]?

Thank you for your helpful comments. We have measured the Cl^-^ by overexpressing CFTR and stimulating with forskolin (FSK) (Figure 9). The results indicated that the application of FSK significantly lowered the intracellular Cl^-^ in HEK293 cells transfected with CFTR. The results obtained with GPR64-C-terminal (GPR64β) co-expression caused more significant Cl^-^ reduction in HEK293 cells compared with that in CFTR/GPR64 full-length co-transfected cells (Figure 9—figure supplement 5).

In Figure 2, the mere co-localization of two proteins to a membrane determined imaged by confocal microscopy does not indicate or suggest that a complex with functional coupling exists. It indicates that two proteins are targeted to the same membrane. This same concern is relevant for data presented in Figure 3, Figure 4, and Figure 5.

Thank you for your helpful comments. In addition to co-localization, we also performed co-immunoprecipitation experiments. The results are shown in Figure 5, Figure 5—figure supplement 5, Figure 8 and Figure 8—figure supplement 2 in the revised manuscript.

For co-IP experiments shown in Figure 4, there is insufficient explanation (in Methods, Results, Legend etc.) to comprehend what is being done. By elimination, I assume the anti-HA blot was against arrestins, but, this is not detailed (for instance, I cannot find details of HA-tagged arrestin constructs).

Thank you for these helpful suggestions. We have revised the labeling of each lane in the new Figure 8—figure supplement 3 (previous Figure 4) and the corresponding figure legends accordingly.

[Editors' note: the author responses to the re-review follow.]

The manuscript has been improved but there are some remaining problematic issues that need to be addressed, as outlined below:1) The intracellular Cl^-^ measurements remain problematic due to the lack of a radiometric method. These measurements should be either replaced by suitable radiometric or electrophysiological measurements or removed from the manuscript.

We thank the reviewer for these helpful suggestions. The Cl^-^ measurements and the corresponding statements were removed from the current manuscript (Figure 5, Figure 7, Figure 8, Figure 9, Figure 5—figure supplement 1 and Figure 9—figure supplement 5 in the previous version of the manuscript).

2) Western blots need controls and markers.

Thank you for these helpful suggestions. The controls and markers of Figure 5, Figure 9, Figure 2—figure supplement 1, Figure 8—figure supplement 3, Figure 9—figure supplement 1, Figure 9—figure supplement 3, Figure 10—figure supplement 2–4 and Figure 11—figure supplement 2 in the revised manuscript were added accordingly

3) The CFTR currents are extremely small. The i/v shift with gluconate is too small and not typical for CFTR. Original whole cell overlay currents or continuous recordings should be shown. What is the proof in addition to CFTRinh172 that the authors truly measured CFTR currents?

We thank the reviewer for these helpful suggestions. A continuous recording of the whole-cell Cl^-^ current in response to Cl^-^ concentration changes is shown in Figure 6. In addition to the pharmacological intervention by the application of CFTRinh172, we knocked down the CFTR expression in efferent ductules using si-RNA; this knockdown was verified by qRT-PCR (Figure 7). The genetically reduced CFTR expression significantly reduced the Cl^-^ current measured in GPR64 promoter-labeled cells compared to that in control si-RNA-treated efferent ductule cells (Figure 7). Both the pharmacological intervention and genetic approach supported that CFTR mediated the observed Cl^-^ currents. Additionally, by checking more detail in the published literature, we found that the magnitude of CFTR current is sperm is similar to our observation in efferent ductules (Figueiras-Fierro et al., 2013).

4) Figure 4 suggests a GPR64-dependent expression of CFTR-mRNA. How does this fit to the expression data shown in Figure 5/I and to the data shown in Figure 6?

We thank the reviewer for this comment. The qRT-PCR result in Figure 4 indicates that both CFTR and GPR64 mRNA are highly enriched in GPR64 promoter-labeled efferent ductules. Two potential mechanisms may explain this observation. (1) The first mechanism is that CFTR expression is dependent on GPR64 expression. However, in GPR64 knockout mice, the CFTR expression level did not change significantly (Figure 5 in the revised manuscript). Furthermore, when we compared the GPR64 promoter-labeled efferent ductule cells derived from the GPR64^-/Y^ mice with their wild-type littermates, the CFTR mRNA level did not change significantly despite the significant difference in the GPR64 mRNA expression level (Figure 4—figure supplement 1 in the revised manuscript, new data). Therefore, CFTR expression is not dependent on GPR64 expression. (2) The second mechanism is that both GPR64 and CFTR were specifically expressed in non-ciliated cells in efferent ductules. GPR64 and CFTR may share a similar transcriptional or epigenetic regulatory mechanism. We added these considerations to the revised manuscript (subsection “ADGRG2 and CFTR coupling in the efferent ductules and its function in fluid reabsorption”).

In addition, we used GPR64-promoter-RFP to label the non-ciliated cells in efferent ductules, which, compared to ciliated cells (non-GRP64-RFP-labeled cells), bear distinct morphological properties (Figure 1); we also changed the label in Figure 4 to clarify the description.

5) Figure 10: In what cells were these data obtained? No control for expression of the various GPR64 mutants is provided.

We thank the reviewer for his helpful suggestion. The experiments of Figure 10 (Figure 11 in the revised manuscript) were performed using HEK293 cells. We have added corresponding statement in subsection “Molecular determinants of ADGRG2coupling with G protein subtypes and their contribution to the regulation of CFTR activity in vitro” in the revised manuscript. The expression control was added in Figure 11—figure supplement 2 accordingly.